# Tree-Wasserstein Distance for High Dimensional Data with a Latent Feature Hierarchy

**Ya-Wei Eileen Lin**[1]     **Ronald R. Coifman**[2]     **Gal Mishne**[3]     **Ronen Talmon**[1]

[1]Viterbi Faculty of Electrical and Computer Engineering, Technion
[2]Department of Mathematics, Yale University
[3]Halicioğlu Data Science Institute, University of California San Diego

## Abstract

Finding meaningful distances between high-dimensional data samples is an important scientific task. To this end, we propose a new tree-Wasserstein distance (TWD) for high-dimensional data with two key aspects. First, our TWD is specifically designed for data with a latent feature hierarchy, i.e., the features lie in a hierarchical space, in contrast to the usual focus on embedding samples in hyperbolic space. Second, while the conventional use of TWD is to speed up the computation of the Wasserstein distance, we use its inherent tree as a means to learn the latent feature hierarchy. The key idea of our method is to embed the features into a multi-scale hyperbolic space using diffusion geometry and then present a new tree decoding method by establishing analogies between the hyperbolic embedding and trees. We show that our TWD computed based on data observations provably recovers the TWD defined with the latent feature hierarchy and that its computation is efficient and scalable. We showcase the usefulness of the proposed TWD in applications to word-document and single-cell RNA-sequencing datasets, demonstrating its advantages over existing TWDs and methods based on pre-trained models.

## 1 Introduction

High-dimensional datasets with a latent feature hierarchy (Fefferman et al., 2016) are prevalent across various applications, involving data, e.g., genomic sequences (Tanay & Regev, 2017), text and images (Krishna et al., 2017), and movie user ratings (Bennett et al., 2007). In this setting, the assumption is that there is an unknown underlying hierarchical structure to the data features, which is not directly observable. For example, in word-document data, where samples are documents and features are words, the hierarchy underlying features (words) is often assumed because words are naturally organized as nodes in a tree, each connected to others through various hierarchical linguistic relationships (Borge-Holthoefer & Arenas, 2010). A key component of meaningful analyses of such data is finding distances between data samples that take into account the underlying hierarchical structure of features. As opposed to relying on generic metrics, e.g., the Euclidean or correlation distances, such meaningful distances serve as a fundamental building block in a broad range of downstream tasks, e.g., classification and clustering, often leading to significant performance improvement.

Recently, hyperbolic geometry (Ratcliffe et al., 1994) has gained prominence in hierarchical representation learning (Chamberlain et al., 2017; Nickel & Kiela, 2017) because the lengths of geodesic paths in hyperbolic spaces grow exponentially with the radius (Sarkar, 2011), a property that naturally mirrors the exponential growth of the number of nodes in hierarchical structures as the depth increases. Methods using hyperbolic geometry typically focus on finding a hyperbolic embedding of the samples, relying on a (partially) known graph, whose nodes represent the samples (Sala et al., 2018). However, considering such a *known* hierarchical structure of the *samples* is fundamentally different than the problem we consider here, where we aim to find meaningful distances between data samples that incorporate the *latent* hierarchical structure of the *features*.

In this paper, we introduce a new tree-Wasserstein distance (TWD) (Indyk & Thaper, 2003) for this purpose, where we model samples as distributions supported on a latent hierarchical structure. We propose a two-step approach. In the first step, we embed features into continuous hyperbolic spaces (Bowditch, 2007) utilizing diffusion geometry (Coifman & Lafon, 2006) to approximate the hidden

hierarchical metric underlying features. In the second step, we construct a tree that represents the latent feature hierarchy in a bottom-up manner from the hyperbolic embedding. The tree construction is based on a novel notion of a continuous analog of the lowest common ancestor (LCA) in trees defined in hyperbolic spaces by associating shortest paths on trees with hyperbolic geodesic distances. Utilizing the decoded feature tree, we introduce a new TWD for high-dimensional data with a latent feature hierarchy, based on diffusion geometry and hyperbolic geometry. We remark that in Lin et al. (2023), integrating hyperbolic and diffusion geometries was shown to be more effective in recovering latent hierarchies than existing hyperbolic embedding methods (Nickel & Kiela, 2017) (see App. E.3.4). However, the work in Lin et al. (2023) considers implicitly latent hierarchies of the samples, whereas here, we explicitly reveal the latent hierarchical structure of the features through a tree, which is then utilized for finding a meaningful TWD between the samples (see App. F.4). Using a TWD entails two useful attributes. First, the Wasserstein distance (Monge, 1781; Kantorovich, 1942) enables us to integrate feature relations into a meaningful distance between samples. Second, we use a tree to represent the latent feature hierarchy, learn this tree based on the data, and then, naturally incorporate it in the proposed TWD. Notably, this use of a tree significantly differs from the conventional use of trees in TWD for speeding up the computation of Wasserstein distance based on the Euclidean ground metric.

From a theoretical standpoint, we show that our TWD provably recovers the Wasserstein distance based on the ground metric induced by the latent feature hierarchy. From a practical standpoint, we show that the proposed TWD is computationally efficient and scalable, making it suitable for large datasets. We showcase the efficacy of our method on an illustrative synthetic example, word-document datasets, and single-cell RNA-sequencing datasets. Specifically, we show that our TWD leads to superior classification performance compared to existing TWD-based methods (Indyk & Thaper, 2003; Backurs et al., 2020; Le et al., 2019; Yamada et al., 2022) and two application-specific methods based on pre-trained models (WMD (Kusner et al., 2015) and GMD (Bellazzi et al., 2021)).

Our main contributions are as follows. (i) We present a new TWD for data with a hidden feature hierarchy, which provably recovers the Wasserstein distance with the true latent feature tree and can be efficiently computed. (ii) We present a novel data-driven tree decoding method based on a new definition of the LCA counterpart in high-dimensional hyperbolic spaces. (iii) We demonstrate the empirical advantages of our TWD over existing TWDs and pre-trained Wasserstein distance methods.

## 2   RELATED WORK

**Wasserstein Distance.** The Wasserstein distance (Villani, 2009) is a powerful tool for transforming the relations between features into a distance between probability distributions (samples from the probability simplex). It can be computed by solving the optimal transport (OT) problem (Monge, 1781; Kantorovich, 1942) with linear programming (Peyré et al., 2019) in cubic complexity. Various approaches, such as adding entropic regularization (Cuturi, 2013), using wavelet functions (Shirdhonkar & Jacobs, 2008; Gavish et al., 2010), applying Kantorovich-Rubinstein dual (Leeb & Coifman, 2016; Arjovsky et al., 2017), and computing the average Wasserstein distance across one-dimensional projections (Rabin et al., 2012), have been developed to reduce the computational complexity.

**Tree-Wasserstein Distance.** Another popular alternative to reduce the computational complexity of the Wasserstein distance is the tree-Wasserstein distance (TWD) (Indyk & Thaper, 2003), which can be computed in linear time, enabling efficient comparisons of large datasets. Existing TWD methods (Leeb, 2018; Sato et al., 2020; Takezawa et al., 2021; Yamada et al., 2022) attempt to approximate the Wasserstein distance based on the Euclidean ground metric with a tree metric by constructing a tree using the standard metric in the Euclidean ambient space (see App. B for more details). Our approach has a different goal. We assume that the features lie in a latent hierarchical space, infer a tree that represents this space, and compute a TWD based on the inferred tree to define a meaningful distance between the samples, incorporating the *latent feature hierarchy*.

**Hyperbolic Representation Learning.** While effective in computing embeddings and distances for hierarchical data samples, current hyperbolic representation learning methods (Chamberlain et al., 2017; Nickel & Kiela, 2017; 2018) are usually given a hierarchical graph, e.g., a tree, and do not include decoding a tree from observational data. One exception is the work in Chami et al. (2020), which introduces a method for decoding a tree in a two-dimensional Poincaré disk for hierarchical clustering (Murtagh & Contreras, 2012), by minimizing the continuous Dasgupta's cost (Dasgupta,

2016). However, this approach does not approximate the TWD between samples with a latent feature hierarchy because it does not recover the tree underlying the features.

# 3 PRELIMINARIES

**Notation.** For $m \in \mathbb{N}$, we denote $[m] = \{1, \ldots, m\}$. Let $\mathbf{X} \in \mathbb{R}^{n \times m}$ be a data matrix with $n$ rows (samples) and $m$ columns (features). We denote by $\mathbf{X}_{i,:}$ the $i$-th row (sample) and $\mathbf{X}_{:,j}$ the $j$-th column (feature). Let $d \simeq d'$ denote the bilipschitz equivalence between metrics $d$ and $d'$ defined on the set $\mathcal{Y}$. That is, there exist constants $c, C > 0$ s.t. $cd(y_1, y_2) \leq d'(y_1, y_2) \leq Cd(y_1, y_2) \, \forall y_1, y_2 \in \mathcal{Y}$.

**Tree-Wasserstein Distance.** Consider a tree $T = (V, E, \mathbf{A})$ with $m$ leaves rooted at node 1 w.l.o.g., where $V$ is the vertex set, $E$ is the edge set and $\mathbf{A} \in \mathbb{R}^{|V| \times |V|}$ contains edge weights. The tree metric $d_T$ between two nodes is the length of the shortest path on $T$. Let $\Gamma_T(v)$ be the set of nodes in the subtree of $T$ rooted at $v \in V$. For any $u \in V$, there exists a unique node $v$, which is the parent of $u$. We denote by $\omega_u = d_T(u, v)$ the length of the edge between any node $u$ and its parent $v$ (see App. A.4 for additional background on trees). The TWD (Indyk & Thaper, 2003; Evans & Matsen, 2012; Le et al., 2019) between two distributions $\boldsymbol{\mu}_1, \boldsymbol{\mu}_2 \in \mathbb{R}^m$ supported on the tree $T$ is defined by

$$\mathrm{TW}(\boldsymbol{\mu}_1, \boldsymbol{\mu}_2, T) = \sum_{v \in V} \omega_v \left| \sum_{u \in \Gamma_T(v)} (\boldsymbol{\mu}_1(u) - \boldsymbol{\mu}_2(u)) \right|. \tag{1}$$

**Diffusion Geometry.** Consider a set of points $\{\mathbf{a}_j \in \mathbb{R}^n\}_{j=1}^m$ assumed to lie on a manifold embedded in $\mathbb{R}^n$. Let $\mathbf{Q} \in \mathbb{R}^{m \times m}$ be an affinity matrix, given by $\mathbf{Q}_{jj'} = \exp\left(-d^2(j, j')/\epsilon\right)$, where $d(j, j')$ is some suitable distance between $\mathbf{a}_j$ and $\mathbf{a}_{j'}$, and $\epsilon > 0$ is a scale parameter, often chosen as the median pairwise distance among data points scaled by a constant factor (Ding & Wu, 2020) (see App. D for hyperparameter tuning). Following Coifman & Lafon (2006), we consider the density-normalized affinity matrix $\widehat{\mathbf{Q}} = \mathbf{D}^{-1} \mathbf{Q} \mathbf{D}^{-1}$ to address the effects of non-uniform data sampling, where $\mathbf{D}$ is diagonal with entries $\mathbf{D}_{jj} = \sum_{j'} \mathbf{Q}_{jj'}$, and the diffusion operator $\mathbf{P}$, defined by

$$\mathbf{P} = \widehat{\mathbf{Q}} \widehat{\mathbf{D}}^{-1}, \text{ where } \widehat{\mathbf{D}}_{jj} = \sum_{j'} \widehat{\mathbf{Q}}_{jj'}. \tag{2}$$

The matrix $\widehat{\mathbf{Q}}$ can be viewed as edge weights of a graph $G = ([m], E, \widehat{\mathbf{Q}})$. Since the diffusion operator $\mathbf{P}$ is column-stochastic, it can be viewed as a transition probability matrix of a Markov chain on the graph $G$. Therefore, $\mathbf{P}$ can propagate densities between nodes. Specifically, the vector $\mathbf{p}_j^t = \mathbf{P}^t \mathbf{e}_j \in \mathbb{R}^m$ is the propagated density after diffusion time $t \in \mathbb{R}$ of a density concentrated at node $j$, where $\mathbf{e}_j \in \mathbb{R}^m$ is the indicator vector of the $j$-th node, $\mathbf{p}_j^t(i) \geq 0$ and $\left\| \mathbf{p}_j^t \right\|_1 = 1$.

**Poincaré Half-Space Model of Hyperbolic Geometry.** Hyperbolic geometry is a Riemannian geometry with constant negative curvature $-1$ (Beardon, 2012). We utilize the $m$-dimensional Poincaré half-space model $\mathbb{H}^m$, defined by $\mathbb{H}^m = \{\mathbf{x} \in \mathbb{R}^m | \mathbf{x}(m) > 0\}$ with the Riemannian metric tensor $ds^2 = \left(d\mathbf{x}^2(1) + d\mathbf{x}^2(2) + \ldots + d\mathbf{x}^2(m)\right)/\mathbf{x}^2(m)$. The Riemannian distance for any two hyperbolic point $\mathbf{x}, \mathbf{y} \in \mathbb{H}^m$ is defined by $d_{\mathbb{H}^m}(\mathbf{x}, \mathbf{y}) = 2 \sinh^{-1}(\|\mathbf{x} - \mathbf{y}\|_2 / (2\sqrt{\mathbf{x}(m)\mathbf{y}(m)}))$.

**Rooted Binary Tree.** A rooted and balanced binary tree $B$ with $m$ leaves is a tree whose root node has degree 2, $m$ leaves have degree 1, and $m - 2$ internal nodes have degree 3. For any two leaves $j, j'$, let $j \vee j'$ and $\Gamma_B(j \vee j')$ denote their lowest common ancestor (LCA) and the set of leaves in the subtree of $B$ rooted at $j \vee j'$, respectively. Given any three leaves $j_1, j_2, j_3$, the LCA relation $[j_2 \vee j_3 \sim j_1]_B$ holds if $\Gamma_B(j_2 \vee j_3) \subset \Gamma_B(j_1 \vee j_2 \vee j_3)$ (Wang & Wang, 2018), as shown in Fig. 1(a). For binary trees, only one of $[j_1 \vee j_2 \sim j_3]_B$, $[j_1 \vee j_3 \sim j_2]_B$, or $[j_3 \vee j_2 \sim j_1]_B$ can hold.

# 4 PROPOSED METHOD

## 4.1 PROBLEM FORMULATION

Consider a data matrix $\mathbf{X} \in \mathbb{R}^{n \times m}$, whose $n$ rows consist of samples and $m$ columns consist of features. Suppose the features $\{\mathbf{X}_{:,1}, \ldots, \mathbf{X}_{:,m}\} \subseteq \mathcal{H} \subset \mathbb{R}^n$ lie on an underlying manifold $\mathcal{H}$, which is a complete and simply connected Riemannian manifold with negative curvature embedded in a high-dimensional ambient space $\mathbb{R}^n$ with geodesic distance $d_{\mathcal{H}}$. We view $(\mathcal{H}, d_{\mathcal{H}})$ as a hidden hierarchical

metric space and consider a weighted tree $T = ([m], E, \mathbf{A}, \mathbf{X}^\top)$ as its discrete approximation in the following sense. The tree node $j \in [m]$ is associated with the $j$-th feature, $E$ is the edge set, and $\mathbf{A} \in \mathbb{R}^{m \times m}$ is the weight matrix of the tree edges, defined such that the tree distance $d_T(j, j')$ between two nodes $j$ and $j'$, i.e., the length of the shortest path on $T$, coincides with the geodesic distance $d_{\mathcal{H}}(j, j')$ between the features $j$ and $j'$. Therefore, we refer to the hidden tree distance $d_T$ as the ground truth distance. Note that the assumption of an underlying manifold embedded in an ambient space $\mathbb{R}^n$ is widely used in studies of data manifolds and manifold learning (Shnitzer et al., 2022; Katz et al., 2020; Tong et al., 2021; Huguet et al., 2022; Kapuśniak et al., 2024), where the manifold typically underlies *samples* and graphs are viewed as discretizations of the underlying Riemannian manifold. In contrast, we consider the manifold underlying *features*. Specifically, we focus on a specification of the manifold to a hierarchical space $\mathcal{H}$ of features, and accordingly, a specification of the graph to a tree $T$. Our aim is to construct a tree $B$ from the data $\mathbf{X}$, such that the TWD between samples using our learned tree $B$ is (bilipschitz) equivalent to the Wasserstein distance using the ground truth latent tree distance $d_T$. To allow this construction of $B$, we assume that local affinities between (node) features $\{\mathbf{X}_{:,j}\}_{j=1}^m$ that are close in $\mathbb{R}^n$ embody information on the hidden hierarchical structure (see Sec. 4.4 for a formal description of this assumption).

Following a large body of work (Villani, 2009; Cuturi, 2013; Courty et al., 2017; Solomon et al., 2015; Yurochkin et al., 2019; Alvarez-Melis & Jaakkola, 2018), we focus on samples $\mathbf{X}_{i,:} \in \mathbb{R}_+^m$ with positive entries that can be normalized into discrete histograms $\mathbf{x}_i = \mathbf{X}_{i,:}/\|\mathbf{X}_{i,:}\|_1$, which is natural when the distribution over features is more important than the total mass. However, we note that our method is also applicable to data normalized into a histogram using other techniques.

## 4.2 HYPERBOLIC DIFFUSION EMBEDDING

In our approach, we first construct a hyperbolic representation following Lin et al. (2023). We briefly describe the steps of this construction below. The feature diffusion operator $\mathbf{P} \in \mathbb{R}^{m \times m}$ is built according to Eq. (2), where $d(j, j')$ is an initial distance between the features $j$ and $j'$ embedded in the ambient space $\mathbb{R}^n$. In Sec. 5, we compute the initial distance using the cosine similarity in the ambient space, i.e., $d(j, j') = 1 - \frac{\mathbf{X}_{:,j} \cdot \mathbf{X}_{:,j'}}{\|\mathbf{X}_{:,j}\|_2 \|\mathbf{X}_{:,j'}\|_2}$, where $\cdot$ is the dot product. We refer to App. D for further discussion on the initial distance metric, the scale parameter $\epsilon$, and the kernel type. Then, we construct the multi-scale diffusion densities $\boldsymbol{\phi}_j^k = \mathbf{P}^{2^{-k}} \mathbf{e}_j \in \mathbb{R}^m$ for each feature $j \in [m]$, considering a series of diffusion time steps on a dyadic grid $t = 2^{-k}$ for $k \in \mathbb{Z}_0^+$. These densities give rise to a multi-scale view of the features on the dyadic grid. We then embed each feature $j$ at the $k$-th scale by $(j, k) \mapsto \mathbf{z}_j^k = [(\boldsymbol{\psi}_j^k)^\top, 2^{k/2-2}]^\top \in \mathbb{H}^{m+1}$ based on the exponential growth of the Poincaré half-space, where $\boldsymbol{\psi}_j^k = \sqrt{\boldsymbol{\phi}_j^k} \in \mathbb{R}^m$ is the element-wise square root of the diffusion density $\boldsymbol{\phi}_j^k$. The multi-scale embedding of the corresponding diffusion operators in hyperbolic space (Lin et al., 2023) results in distances that are exponentially scaled. This scaling aligns with the structure of a tree distance, effectively establishing a natural hierarchical relation between different diffusion timescales. While it is well-established that diffusion geometry effectively recovers the underlying manifold (Coifman & Lafon, 2006), recent research by Lin et al. (2023) builds on this by showing that propagated densities, when applied with diffusion times on dyadic grids, can reveal hidden hierarchical metric. This method effectively captures local information from exponentially expanding neighborhoods around each point. We note that other hyperbolic embeddings (e.g., Chamberlain et al. (2017); Nickel & Kiela (2017)) could potentially be considered. However, these methods usually assume a (partially) known graph $T$ instead of high-dimensional observational data $\mathbf{X}$, and therefore, we do not opt to use them (see App. E.3.4 for more details).

## 4.3 HYPERBOLIC DIFFUSION LCA

To reveal the latent hierarchy of the features based on the continuous hyperbolic embedding, we first establish the analogy between geodesic paths in hyperbolic space $\mathbb{H}^{m+1}$ and paths on trees. We define the following hyperbolic lowest common ancestor (LCA).

**Definition 1** (Hyperbolic LCA). *The hyperbolic LCA of any two points* $\mathbf{z}_j^k, \mathbf{z}_{j'}^k \in \mathbb{H}^{m+1}$ *at the $k$-th scale is defined by their Fréchet mean (Fréchet, 1948) as* $\mathbf{z}_j^k \vee \mathbf{z}_{j'}^k :=$ $\arg\min_{\mathbf{z} \in \mathbb{H}^{m+1}} \left( d_{\mathbb{H}^{m+1}}^2(\mathbf{z}, \mathbf{z}_j^k) + d_{\mathbb{H}^{m+1}}^2(\mathbf{z}, \mathbf{z}_{j'}^k) \right)$.

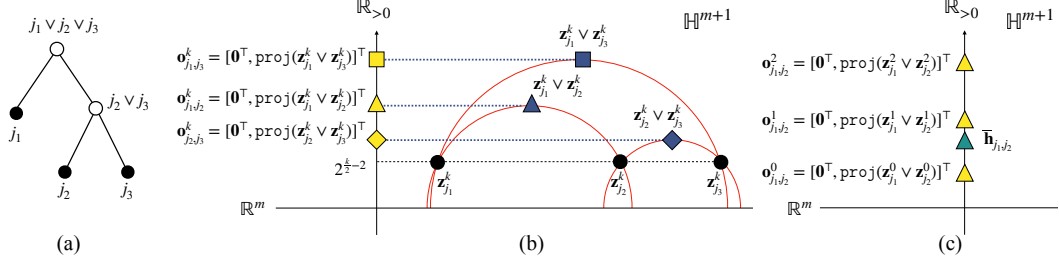

Figure 1: (a) The LCA relation of $j_1, j_2, j_3$ on tree $B$. (b) The hyperbolic LCA relation $[j_2 \vee j_3 \sim j_1]_{\mathbb{H}^{m+1}}^k$ for $\mathbf{z}_{j_1}^k, \mathbf{z}_{j_2}^k, \mathbf{z}_{j_3}^k \in \mathbb{H}^{m+1}$ along with their geodesic paths (red semi-circles). Blue points indicate the hyperbolic LCAs, and yellow points represent their orthogonal projections. The value $\texttt{proj}(\cdot)$ signifies the parent-child relation. (c) The HD-LCA $\overline{\mathbf{h}}_{j_1,j_2}$ defined as the Riemannian mean of the orthogonal projections $\{\mathbf{o}_{j_1,j_2}^k\}_{k=0}^{K_c}$, incorporating the multi-scale hyperbolic LCAs.

**Proposition 1.** *The hyperbolic LCA $\mathbf{z}_j^k \vee \mathbf{z}_{j'}^k$ in Def. 1 has a closed-form solution, given by $\mathbf{z}_j^k \vee \mathbf{z}_{j'}^k = \left[\frac{1}{2}\left(\boldsymbol{\psi}_j^k + \boldsymbol{\psi}_{j'}^k\right)^\top, \texttt{proj}(\mathbf{z}_j^k \vee \mathbf{z}_{j'}^k)\right]^\top$, where $\texttt{proj}(\mathbf{z}_j^k \vee \mathbf{z}_{j'}^k) = \left\|\left[\frac{1}{2}\left(\boldsymbol{\psi}_j^k - \boldsymbol{\psi}_{j'}^k\right)^\top, 2^{k/2-2}\right]^\top\right\|_2$.*

The proof of Prop. 1 is in App. C. Note that $\texttt{proj}(\mathbf{z}_j^k \vee \mathbf{z}_{j'}^k)$ is the orthogonal projection of the hyperbolic LCA onto the $(m+1)$-axis $\mathbb{R}_{>0}$ in $\mathbb{H}^{m+1}$, which is the "radius" of the geodesic connecting $\mathbf{z}_j^k$ and $\mathbf{z}_{j'}^k$, as depicted in Fig. 1(b). We assert that this orthogonal projection can be used to identify the hierarchical relation at the $k$-th scale. This argument is made formal in the following.

**Definition 2** (Hyperbolic LCA Relation). *Given any three points $\mathbf{z}_{j_1}^k, \mathbf{z}_{j_2}^k, \mathbf{z}_{j_3}^k \in \mathbb{H}^{m+1}$ at the $k$-th scale, the relation $[j_2 \vee j_3 \sim j_1]_{\mathbb{H}^{m+1}}^k$ holds if $\texttt{proj}(\mathbf{z}_{j_2}^k \vee \mathbf{z}_{j_3}^k) = \min\{\texttt{proj}(\mathbf{z}_{j_1}^k \vee \mathbf{z}_{j_2}^k), \texttt{proj}(\mathbf{z}_{j_1}^k \vee \mathbf{z}_{j_3}^k), \texttt{proj}(\mathbf{z}_{j_2}^k \vee \mathbf{z}_{j_3}^k)\}$.*

**Proposition 2.** *For any $k_1 \leq k_2$, $2^{-(k_2-k_1)} \simeq d_{\mathbb{H}^{m+1}}(\mathbf{z}_j^{k_2} \vee \mathbf{z}_{j'}^{k_2}, \mathbf{z}_j^{k_2})/d_{\mathbb{H}^{m+1}}(\mathbf{z}_j^{k_1} \vee \mathbf{z}_{j'}^{k_1}, \mathbf{z}_j^{k_1})$.*

The proof of Prop. 2 is in App. C. Prop. 2 implies that the hyperbolic LCAs at different scales exhibit exponential growths, which is analogous to the exponential growth of the distance between parent and child nodes in trees.

We now define the Hyperbolic Diffusion LCA (HD-LCA) that incorporates the multi-scale hyperbolic LCAs by jointly considering the dyadic diffusion times $\{2^{-k}\}_{k=0}^{K_c}$, where $K_c \in \mathbb{Z}_0^+$.

**Definition 3** (HD-LCA). *The HD-LCA of features $j, j' \in [m]$ with $K_c$ scales is defined by the Fréchet mean $\overline{\mathbf{h}}_{j,j'} := \underset{\mathbf{h} \in \mathbb{H}^{m+1}}{\arg\min} \sum_{k=0}^{K_c} d_{\mathbb{H}^{m+1}}^2(\mathbf{h}, \mathbf{o}_{j,j'}^k)$, where $\mathbf{o}_{j,j'}^k = [\mathbf{0}^\top, \texttt{proj}(\mathbf{z}_j^k \vee \mathbf{z}_{j'}^k)]^\top \in \mathbb{H}^{m+1}$.*

**Proposition 3.** *The HD-LCA $\overline{\mathbf{h}}_{j,j'}$ in Def. 3 has a closed-form solution, given by $\overline{\mathbf{h}}_{j,j'} = [0, \ldots, 0, a_{j,j'}]^\top$, where $a_{j,j'} = \sqrt[(K_c+1)]{\texttt{proj}(\mathbf{z}_j^0 \vee \mathbf{z}_{j'}^0) \cdots \texttt{proj}(\mathbf{z}_j^{K_c} \vee \mathbf{z}_{j'}^{K_c})}$.*

The proof of Prop. 3 is in App. C. Note that the HD-LCA in Def. 3 involves two applications of the geodesic paths in $\mathbb{H}^{m+1}$: (i) to establish the hyperbolic LCA at each scale $k$ by connecting two points on the same plane in $\mathbb{R}^{m+1}$, and (ii) to merge these LCAs across multiple levels along the $(m+1)$-axis $\mathbb{R}_{>0}$ in $\mathbb{H}^{m+1}$, as in Fig. 1(c).

Incorporating the multi-scale of the hyperbolic LCAs arranged on a dyadic grid via the HD-LCA facilitates the recovery of the underlying tree relations. A similar approach was presented in Lin et al. (2023), where it was shown that the distance between the embedding $j \mapsto [(\mathbf{z}_j^0)^\top, \ldots, (\mathbf{z}_j^{K_c})^\top]^\top \in \mathcal{M} = \mathbb{H}^{m+1} \times \ldots \times \mathbb{H}^{m+1}$ in a product manifold of Poincaré half-spaces, given by $d_{\mathcal{M}}(j,j') = \sum_{k=0}^{K_c} 2 \sinh^{-1}(2^{-k/2+1} \|\mathbf{z}_j^k - \mathbf{z}_{j'}^k\|_2)$, recovers $d_T(j,j')$. However, (Lin et al., 2023) considers a latent hierarchical structure of the samples. It is fundamentally different than the problem we consider in Sec. 4.1, as it does not learn the latent hierarchy of the features explicitly by constructing a tree, and then incorporate it in a meaningful distance between the samples as we do here (see App. F.4 for more details). As shown in App. F.4, this distance is less effective for document and cell classifications.

Note that the closed-form value $a_{j,j'}$ in Prop. 3 can be interpreted as the Riemannian average of the multi-scale hyperbolic LCA relations in Def. 1, resulting in the following HD-LCA relation.

**Definition 4** (HD-LCA Relation). *Given any three features $j_1, j_2, j_3 \in [m]$, the relation $[j_2 \vee j_3 \sim j_1]_{\mathcal{M}}$ holds if $a_{j_2,j_3} = \min\{a_{j_1,j_2}, a_{j_1,j_3}, a_{j_2,j_3}\}$.*

---

**Algorithm 1** HD Binary Tree Decoding

**Input:** Embedding $\{\{\mathbf{z}_1^k\}, \ldots, \{\mathbf{z}_m^k\}\}_{k=0}^{K_c}$
**Output:** $B = (\widetilde{V}, \widetilde{E}, \widehat{\mathbf{A}})$ with $m$ leaf nodes

**function** HD_BT($\{\{\mathbf{z}_1^k\}, \ldots, \{\mathbf{z}_m^k\}\}_{k=0}^{K_c}$)
  $B \leftarrow \texttt{leaves}(\{j\} : j \in [m])$
  **for** $j, j' \in [m]$ **do**
    **for** $k \in \{0, 1, \ldots, K_c\}$ **do**
      $\mathbf{o}_{j,j'}^k \leftarrow [\mathbf{0}^\top, \texttt{proj}(\mathbf{z}_j^k, \mathbf{z}_{j'}^k)]^\top$
    $\overline{\mathbf{h}}_{j,j'} \leftarrow \underset{\mathbf{h} \in \mathbb{H}^{m+1}}{\arg\min} \sum_{k=0}^{K_c} d_{\mathbb{H}^{m+1}}^2(\mathbf{h}, \mathbf{o}_{j,j'}^k)$
  $\mathcal{S} = \{(j,j') | \text{ sorted by } \overline{\mathbf{h}}_{j,j'}(m+1)\}$
  **for** $(j, j') \in \mathcal{S}$ **do**
    **if** $j$ and $j'$ are not in the same subtree
      $\mathcal{I}_j \leftarrow$ internal node for node $j$
      $\mathcal{I}_{j'} \leftarrow$ internal node for node $j'$
      add an internal node for $\mathcal{I}_j$ and $\mathcal{I}_{j'}$
      assign the geodesic edge weight
  **return** $B$

**Algorithm 2** TWD with a latent feature hierarchy

**Input:** Data matrix $\mathbf{X} \in \mathbb{R}^{n \times m}$, feature diffusion operator $\mathbf{P}$, and maximal scale $K_c$
**Output:** TWD $\mathbf{W} \in \mathbb{R}^{n \times n}$

**function** TWD_latent_HieFeature($\mathbf{X}, K_c$)
  $\mathbf{U}\mathbf{\Lambda}\mathbf{V}^\top = \texttt{eig}(\mathbf{P})$
  **for** $k \in \{0, 1, \ldots, K_c\}$ **do**
    $\boldsymbol{\rho}_k \leftarrow \mathbf{\Lambda}^{2^{-k}}$
    **for** $j \in [m]$ **do**
      $\mathbf{z}_j^k \leftarrow \left[\sqrt{(\mathbf{U}\boldsymbol{\rho}_k\mathbf{V}^\top)}\mathbf{e}_j, 2^{k/2-2}\right]^\top$
  $B \leftarrow$ HD_BT($\{\{\mathbf{z}_1^k\}, \ldots, \{\mathbf{z}_m^k\}\}_{k=0}^{K_c}$)
  **for** $i, i' \in [n]$ **do**
    $$\mathbf{W}_{ii'} \leftarrow \sum_{v \in \widetilde{V}} \alpha_v \left| \sum_{u \in \Gamma_B(v)} (\mathbf{x}_i(u) - \mathbf{x}_{i'}(u)) \right|$$
  **return** $\mathbf{W}$

---

### 4.4 BINARY TREE CONSTRUCTION

Equipped with the definitions and properties above, we now present the proposed tree construction in hyperbolic spaces. As any tree can be transformed into a binary tree (Bowditch, 2007; Cormen et al., 2022), we suggest to construct a rooted binary tree $B$ through an iterative merging process. We begin by assigning the features to the leaf nodes because it was shown that the leaves alone contain adequate information to fully reconstruct the tree (Sarkar, 2011) (see App. F.2 for more details). This is not only efficient but also aligns with established TWD (Indyk & Thaper, 2003; Backurs et al., 2020; Le et al., 2019). Then, pairs of features are merged iteratively by their HD-LCA (Def. 3). Specifically, the pair of features that exhibits the highest similarity in the HD-LCA (i.e., the smallest value of $a_{j,j'}$ in Prop. 3) is merged. Note that only one of the HD-LCA relations $[j_2 \vee j_3 \sim j_1]_{\mathcal{M}}$, $[j_1 \vee j_3 \sim j_2]_{\mathcal{M}}$, or $[j_1 \vee j_2 \sim j_3]_{\mathcal{M}}$ holds in $B$. Once the nodes are merged, the weight of the edge is assigned with the induced $\ell_1$ distance between the respective embeddings. This iterative process constructs the binary tree $B$ in a bottom-up manner, similar to single linkage clustering (Gower & Ross, 1969), with the measure of similarity determined by their HD-LCAs and edge weights assigned with the geodesic distance. We summarize the proposed HD binary tree decoding in Alg. 1.

We note that the distance in the embedding (Lin et al., 2023) could be used in existing graph-based algorithms (Alon et al., 1995) or linkage methods (Murtagh & Contreras, 2012) instead of the HD-LCA in the binary tree decoding (Alg. 1). However, these generic algorithms, while providing upper bounds on the distortion of the constructed tree distances, do not build a geometrically meaningful tree when given a tree metric (Sonthalia & Gilbert, 2020), because they do not exploit the hyperbolic geometry of the embedded space. Indeed, as empirically shown in App. E, constructing a feature tree using existing TWDs, graph-based, or linkage approaches with embedding distance is less effective. This may suggest that the information encoded in the multi-scale hyperbolic LCAs is critical for revealing the latent feature tree. In addition, we argue that having a closed-form solution to the basic element of the tree construction HD-LCA in Prop. 3 is an important advantage because it directly enables us to present an effective and interpretable algorithm (Alg. 1) for recovering the underlying tree relationships from the learned hyperbolic embedding.

**Proposition 4.** *Let $r$ be the root node of $B$ obtained in Alg. 1 and $\mathbf{z}_r$ be the corresponding embedding in $\mathcal{M}$. The HD-LCA depth is equivalent to the Gromov product (Gromov, 1987), i.e., $d_{\mathcal{M}}(\mathbf{z}_r, \overline{\mathbf{h}}_{j,j'}) \simeq \langle j, j' \rangle_r$, where $\langle j, j' \rangle_r = \frac{1}{2}(d_B(j, r) + d_B(j', r) - d_B(j, j'))$.*

The proof of Prop. 4 is in App. C. Prop. 4 shows that the HD-LCA depth in the continuous embedding space is the tree depth in the decoded tree $B$.

**Theorem 1.** *For sufficiently large $K_c$ and $m$, there exist some constants $C_1, C_2 > 0$ such that $C_1 d_T(j_1, j_2) \leq d_B(j_1, j_2) \leq C_2 d_T(j_1, j_2)$ for $j_1, j_2 \in [m]$, where $d_B$ is the decoded tree metric $d_B$ in Alg. 1, $T$ is the ground truth latent tree assumed from Sec. 4.1, and $d_T$ is the hidden tree metric between features defined as the length of the shortest path on $T$ (i.e., the geodesic distances).*

The proof of Thm. 1 is in App. C. The validity of Thm. 1 relies on the assumption that the local Gaussian affinities between features allow us to reveal the hidden hierarchy. Formally, in the proof of Thm. 1, we assume that the discrete diffusion operator $\mathbf{P}$ constructed based on the local Gaussian affinities between features in the ambient space $\mathbb{R}^n$, in the limit of $m \to \infty$ and $\epsilon \to 0$, converges pointwise to the heat kernel of the underlying hierarchical metric space $(\mathcal{H}, d_{\mathcal{H}})$. While $T$ is hidden, the diffusion operator $\mathbf{P}$ is accessible and used to construct another tree, $B$, such that, the tree distances of the inferred tree $d_B$ and the hidden tree $d_T$ are equivalent. This result is stated in Thm. 1. In addition, our HD-LCA is a continuous LCA *in the product manifold of $(m + 1)$-dimensional Poincaré half-spaces* that recovers hidden tree relations. It is closely related to the hyperbolic LCA in Chami et al. (2020), where a continuous LCA in a *single* 2D Poincaré disk was proposed for hierarchical clustering, and a tree was constructed by minimizing the Dasgupta's cost (Dasgupta, 2016). However, their method does not recover the latent feature hierarchy, as shown in Sec. 5.

### 4.5 TREE-WASSERSTEIN DISTANCE FOR HIGH-DIMENSIONAL DATA WITH A LATENT FEATURE HIERARCHY

Finally, with the decoded tree $B$ at hand, we define a new TWD between high-dimensional samples with a latent feature hierarchy as follows.

**Definition 5** (TWD for High-Dimensional Data with a Latent Feature Hierarchy.). *The TWD for high-dimensional samples with a latent feature hierarchy is defined as*

$$\text{TW}(\mathbf{x}_i, \mathbf{x}_{i'}, B) := \sum_{v \in \widetilde{V}} \alpha_v \left| \sum_{u \in \Gamma_B(v)} (\mathbf{x}_i(u) - \mathbf{x}_{i'}(u)) \right|, \tag{3}$$

*where $B$ is the decoded tree in Alg. 1, $\alpha_v = d_B(v, v')$, and node $v'$ is the parent of node $v$.*

**Theorem 2.** *For sufficiently large $K_c$ and $m$, there exist some constant $\widetilde{C}_1, \widetilde{C}_2 > 0$ such that $\widetilde{C}_1 \text{TW}(\mathbf{x}_i, \mathbf{x}_{i'}, T) \leq \text{TW}(\mathbf{x}_i, \mathbf{x}_{i'}, B) \leq \widetilde{C}_2 \text{TW}(\mathbf{x}_i, \mathbf{x}_{i'}, T)$ for all samples $i, i' \in [n]$.*

The proof of Thm. 2 is in App. C. Thm. 2 shows that our TWD in Eq. (3) recovers the TWD associated with the latent feature tree $T$ without access to $T$. This could be viewed as an unsupervised Wasserstein metric learning between data samples with the latent feature hierarchy. In addition, our intermediate result in Thm. 1 is noteworthy, as it represents a form of unsupervised ground metric learning (Cuturi & Avis, 2014) derived from the latent tree underlying the features.

We summarize the computation of our method. Notably, the input to the algorithm is the observational data matrix $\mathbf{X}$ alone (i.e., the tree node attributes), and no prior knowledge about the latent feature tree graph $T$ is given. First, we build the feature diffusion operator $\mathbf{P}$ by Eq. (2). Then, for each scale $k$, we embed the features into $\mathbb{H}^{m+1}$ using the $k$-th diffusion densities. Collecting the embedding of all the features from all the scales, we decode a binary tree by Alg. 1, based on the HD-LCA in Def. 3. Finally, we compute the proposed TWD as in Eq. (3). We present the pseudo-code in Alg. 2.

Seemingly, the need to build a distance matrix and apply the eigendecomposition to the resulting kernel limits the applicability to a small number of features. However, using (Shen & Wu, 2022), the proposed TWD can be computed in $O(m^{1.2})$ compared to the naïve implementation that requires $O(mn^3 + m^3 \log m)$ (see App. D). Indeed, in Sec. 5, we show applications with thousands of features.

One limitation of our TWD in Eq. (3) is the restriction to samples from the probability simplex. This requirement, which stems from the definitions of OT framework (Monge, 1781; Kantorovich, 1942) and the Wasserstein and TWDs (Villani, 2009; Cuturi, 2013; Indyk & Thaper, 2003; Backurs et al., 2020; Le et al., 2019; Takezawa et al., 2021; Yamada et al., 2022), restricts its applicability. However, a large body of work (Villani, 2009; Cuturi, 2013; Courty et al., 2017; Solomon et al., 2015; Yurochkin et al., 2019; Alvarez-Melis & Jaakkola, 2018) illustrates that this setting is relevant in a

broad range of applications. In addition, this limitation does not affect the tree decoding in Alg. 1, as it can recover the latent feature hierarchy from $\mathbb{R}^n$, which is in line with existing tree construction methods (Alon et al., 1995; Murtagh & Contreras, 2012). Note that the problem of finding distances between samples with feature hierarchy and possibly with negative entries has been addressed in Gavish et al. (2010); Ankenman (2014); Mishne et al. (2016; 2017) using different approaches.

A notable property of our TWD in Eq. (3) is that it can be interpreted using the tree-sliced Wasserstein distance (TSWD) (Le et al., 2019). Computing the TSWD consists of constructing multiple random trees of different depths from $\mathbf{X}$. Then, the TSWD is given by the average of the TWDs corresponding to these trees. Here, considering only a single scale $k$ in Alg. 1 (as detailed in App. F.1) yields the decoded tree $B^k$. Similarly to the TSWD, we show that our TWD can be recast as the sum of the TWDs corresponding to $B^k$ at multiple scales $k = 0, \ldots, K_c$.

**Proposition 5.** *The TWD in Eq. (3) is bilipschitz equivalent to the sum of the multi-scale hyperbolic TWDs* $\mathtt{TW}(\mathbf{x}_i, \mathbf{x}_{i'}, B^k)$ *for* $k = 0, 1, \ldots, K_c$, *i.e.,* $\mathtt{TW}(\mathbf{x}_i, \mathbf{x}_{i'}, B) \simeq \sum_{k=0}^{K_c} \mathtt{TW}(\mathbf{x}_i, \mathbf{x}_{i'}, B^k)$.

The proof of Prop. 5 is in App. C. Prop. 5 implies that our TWD captures the (tree-sliced) information in the multi-scale trees $B^k$ efficiently as it circumvents their explicit construction.

We conclude with a few remarks. First, it is important to note that our primary objective is not to recover the exact hidden $T$, but to effectively approximate the TWD $\mathtt{TW}(\mathbf{x}_i, \mathbf{x}_{i'}, T)$ from the data matrix $\mathbf{X}$ (i.e., the node attributes from the hidden $T$), which we achieve by the proposed TWD $\mathtt{TW}(\mathbf{x}_i, \mathbf{x}_{i'}, B)$ in Eq. (3) as stated in Thm. 2. Second, our TWD can also be used to approximate the OT distance in linear time. For example, suppose we are interested in computing the OT distance based on a given ground tree metric between the features. While the complexity of this computation is typically $O(m^3 \log m)$ (Pele & Werman, 2009; Bonneel et al., 2011), we can approximate it by the proposed TWD in linear time, as we demonstrate in the runtime analysis in App. E. Third, our method applies to general applications with or without latent hierarchical structure (see App. E.4). Finally, our TWD is differentiable, as detailed in App. F.3, in contrast to existing TWD methods. This differentiability equips the proposed TWD with a continuous gradient, significantly enhancing its applicability in gradient-based optimization, a feature that is not typically found in TWD baselines.

## 5 EXPERIMENTAL RESULTS

We apply the proposed method to a synthetic example and to word-document and single-cell RNA-sequencing (scRNA-seq) data. We use Word2vec (Mikolov et al., 2013) as word embedding vectors for the word-document data and Gene2vec (Du et al., 2019) as gene embedding vectors in scRNA-seq experiments. The implementation details are in App. D Additional experiments are in App. E.

### 5.1 SYNTHETIC WORD-DOCUMENT DATA EXAMPLE

We generate a synthetic dataset with a latent feature hierarchy as follows. The generated dataset is designed to resemble a word-document data matrix $\mathbf{X}$, whose rows (samples) represent documents and columns (features) represent word presence. Our dataset consists of 100 documents and eight words of produce items: apple, orange, banana, carrot, beetroot, kale, spinach, and lettuce. These items are divided into categories and sub-categories as depicted in Fig. 2(a), giving rise to the latent feature hierarchy. The 100 documents are realizations of a random binary vector created according to the probabilistic hierarchical model in Fig. 2(a). For details of the data generation, see App. D.

We compare the proposed TWD with existing TWDs, including (i) Quadtree (Indyk & Thaper, 2003) that constructs a random tree by a recursive division of hypercubes, (ii) Flowtree (Backurs et al., 2020) that computes the optimal flow of Quadtree, (iii) tree-sliced Wasserstein distance (TSWD) (Le et al., 2019) that computes an average of TWDs by random sampling trees, where the numbers of sampling are set to 1, 5, and 10, (iv) UltraTree (Chen et al., 2024) that constructs an ultrametric tree (Johnson, 1967) by minimizing OT regression cost, (v) weighted cluster TWD (WCTWD) (Yamada et al., 2022), (vi) weighted Quadtree TWD (WQTWD) (Yamada et al., 2022), and their sliced variants (vii) SWCTWD and (viii) SWQTWD (Yamada et al., 2022). We also compare to TWDs where the trees are constructed by (i) MST (Prim, 1957), (ii) Tree Representation (TR) (Sonthalia & Gilbert, 2020) through a divide-and-conquer approach, (iii) gradient-based hierarchical clustering (HC) in hyperbolic space (gHHC) (Monath et al., 2019), (iv) gradient-based Ultrametric Fitting (UltraFit)

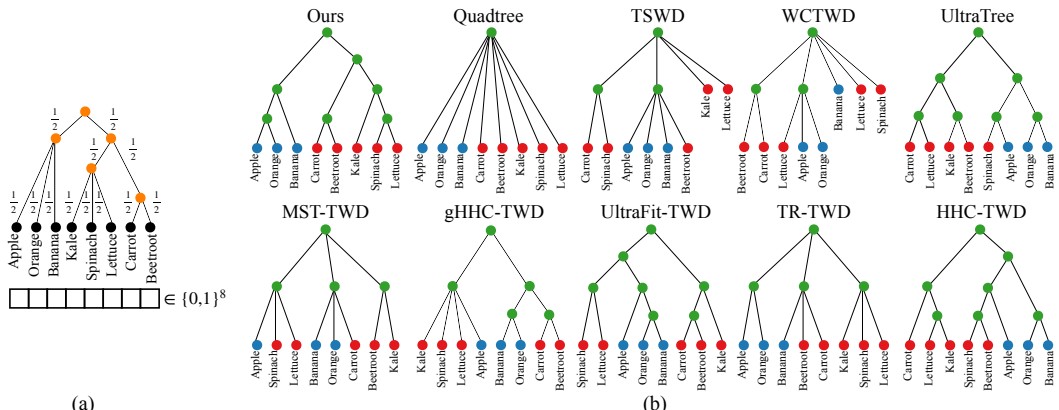

Figure 2: (a) Illustration of the probabilistic hierarchical model for generating synthetic samples consisting of 8 binary elements. The orange nodes represent (sub)categories, and the black nodes present produce items. The edge weights represent the probabilities. (b) Feature trees constructed by our TWD and competing baselines. Nodes corresponding to fruits are colored in blue, those representing vegetables are in red, and the internal nodes are in green.

(Chierchia & Perret, 2019), and (v) HC by hyperbolic Dasgupta's cost (HHC) (Chami et al., 2020). We refer to App. B for a detailed description of baselines and highlight the novel aspects of our TWD.

Fig. 2(b) presents the trees constructed using different methods. Our TWD constructs a binary tree that accurately represents the latent feature hierarchy, demonstrating a clear advantage over the trees obtained by the baselines. Further evaluation through a binary document classification task using $k$-nearest neighbors ($k$NN), where the documents are labeled by the presence of fruits (with probability $0.5$, creating two balanced classes), is presented in Tab. 1. The depicted classification accuracy is obtained by averaging over five runs, where in each run, the dataset is randomly split into $70\%$ training and $30\%$ testing sets. The same random splits are used across all comparisons. We see that our TWD demonstrates a significant improvement over the competing TWDs.

## 5.2 REAL-WORLD WORD-DOCUMENT DATA ANALYSIS

We demonstrate the power of the proposed TWD in real-world document classification tasks. The latent hierarchical structure of words (features) can be viewed as a node branching into related terms (Gollapalli & Caragea, 2014; Yurochkin et al., 2019; Chang & Blei, 2010), allowing for meaningful comparison between documents in OT distance (Kusner et al., 2015), as it measures how words and their associated meanings are transported from one document to another.

We test four word-document benchmarks in Kusner et al. (2015), including BBCSPORT, TWITTER, CLASSIC, and AMAZON. Detailed descriptions of these datasets are reported in App. D. We compare the proposed TWD to the same competing methods as in Sec. 5.1. In addition, we include the Word Mover's Distance (WMD) (Kusner et al., 2015) that computes the OT distance between documents using Word2Vec (Mikolov et al., 2013). A dissimilarity classification based on our TWD and the baselines using the $k$NN classifier is applied. We use cross-validation with five trials, where the dataset is randomly divided into $70\%$ training set and $30\%$ testing set, following previous work in word-document data analysis (Kusner et al., 2015; Huang et al., 2016).

Tab. 1 presents the document classification accuracy. The results of Quadtree, Flowtree, TSWD, and WMD are taken from Huang et al. (2016); Takezawa et al. (2021). Our method outperforms all the competing baselines on BBCSPORT, TWITTER, and AMAZON datasets, including the pre-trained WMD that is specifically designed for word-document data. On CLASSIC dataset, our proposed TWD yields the second-best classification accuracy and is competitive with WMD.

## 5.3 SINGLE-CELL GENE EXPRESSION DATA ANALYSIS

We further demonstrate the advantages of our TWD in cell classification tasks on scRNA-seq data. The hierarchical structure of genes in scRNA-seq data (Tanay & Regev, 2017) are essential for

Table 1: Document and cell classification accuracy (the highest in bold and the second highest underlined). Results marked with * are taken from Huang et al. (2016); Takezawa et al. (2021). The empirical comparisons to TWD methods demonstrate the advantages of our proposed approach, which effectively incorporates the latent feature hierarchy, leading to improved performance.

| | Synthetic | Real-World Word-Document | | | | Real-World scRNA-seq | |
| | | BBCSPORT | TWITTER | CLASSIC | AMAZON | Zeisel | CBMC |
|---|---|---|---|---|---|---|---|
| Quadtree | $\underline{98.2}_{\pm1.2}$ | $95.5_{\pm0.5}*$ | $69.6_{\pm0.8}*$ | $95.9_{\pm0.4}*$ | $89.3_{\pm0.3}*$ | $80.1_{\pm1.2}$ | $80.6_{\pm0.6}$ |
| Flowtree | $97.3_{\pm0.9}$ | $95.3_{\pm1.1}*$ | $70.2_{\pm0.9}*$ | $94.4_{\pm0.6}*$ | $90.1_{\pm0.3}*$ | $81.7_{\pm0.9}$ | $81.8_{\pm0.9}$ |
| WCTWD | $91.4_{\pm1.2}$ | $92.6_{\pm2.1}$ | $69.1_{\pm2.6}$ | $93.7_{\pm2.9}$ | $88.2_{\pm1.4}$ | $81.3_{\pm4.9}$ | $78.4_{\pm3.3}$ |
| WQTWD | $92.8_{\pm1.8}$ | $94.3_{\pm1.7}$ | $69.4_{\pm2.4}$ | $94.6_{\pm3.2}$ | $87.4_{\pm1.8}$ | $80.9_{\pm3.5}$ | $79.1_{\pm3.0}$ |
| UltraTree | $93.2_{\pm3.1}$ | $93.1_{\pm1.5}$ | $68.1_{\pm3.2}$ | $92.3_{\pm1.9}$ | $86.2_{\pm3.1}$ | $83.9_{\pm1.6}$ | $\underline{82.3}_{\pm2.6}$ |
| TSWD-1 | $90.9_{\pm1.3}$ | $87.6_{\pm1.9}*$ | $69.8_{\pm1.3}*$ | $94.5_{\pm0.5}*$ | $85.5_{\pm0.6}*$ | $79.6_{\pm1.8}$ | $72.6_{\pm1.8}$ |
| TSWD-5 | $92.4_{\pm1.1}$ | $88.1_{\pm1.3}*$ | $70.5_{\pm1.1}*$ | $95.9_{\pm0.4}*$ | $90.8_{\pm0.1}*$ | $81.3_{\pm1.4}$ | $74.9_{\pm1.1}$ |
| TSWD-10 | $94.2_{\pm0.9}$ | $88.6_{\pm0.9}*$ | $70.7_{\pm1.3}*$ | $95.9_{\pm0.6}*$ | $91.1_{\pm0.5}*$ | $83.2_{\pm0.8}$ | $76.5_{\pm0.7}$ |
| SWCTWD | $94.9_{\pm2.3}$ | $92.8_{\pm1.2}$ | $70.2_{\pm1.2}$ | $94.1_{\pm1.4}$ | $90.2_{\pm1.2}$ | $81.9_{\pm3.1}$ | $78.3_{\pm1.7}$ |
| SWQTWD | $95.1_{\pm1.8}$ | $94.5_{\pm1.0}$ | $70.6_{\pm1.9}$ | $95.4_{\pm2.0}$ | $89.8_{\pm1.1}$ | $80.7_{\pm2.5}$ | $79.8_{\pm2.5}$ |
| MST-TWD | $88.1_{\pm2.7}$ | $88.4_{\pm1.9}$ | $68.2_{\pm1.9}$ | $90.0_{\pm3.1}$ | $86.4_{\pm1.2}$ | $80.1_{\pm3.1}$ | $76.2_{\pm2.5}$ |
| TR-TWD | $93.9_{\pm0.7}$ | $89.2_{\pm0.9}$ | $70.2_{\pm0.7}$ | $92.9_{\pm0.8}$ | $88.7_{\pm1.1}$ | $80.3_{\pm0.7}$ | $78.4_{\pm1.2}$ |
| HHC-TWD | $94.1_{\pm1.7}$ | $85.3_{\pm1.8}$ | $70.4_{\pm0.4}$ | $93.4_{\pm0.8}$ | $88.5_{\pm0.7}$ | $82.3_{\pm0.7}$ | $77.3_{\pm1.1}$ |
| gHHC-TWD | $93.5_{\pm1.9}$ | $83.2_{\pm2.4}$ | $69.9_{\pm1.8}$ | $90.3_{\pm2.2}$ | $86.9_{\pm2.0}$ | $79.4_{\pm1.9}$ | $73.6_{\pm1.6}$ |
| UltraFit-TWD | $94.3_{\pm1.2}$ | $84.9_{\pm1.4}$ | $69.5_{\pm1.2}$ | $91.6_{\pm0.9}$ | $87.4_{\pm1.6}$ | $81.9_{\pm3.3}$ | $77.8_{\pm1.2}$ |
| WMD | - | $95.4_{\pm0.7}*$ | $\underline{71.3}_{\pm0.6}*$ | $\mathbf{97.2}_{\pm0.1}*$ | $92.6_{\pm0.3}*$ | - | - |
| GMD | - | - | - | - | - | $\underline{84.2}_{\pm0.7}$ | $81.4_{\pm0.7}$ |
| Ours | $\mathbf{99.8}_{\pm0.1}$ | $\mathbf{96.1}_{\pm0.4}$ | $\mathbf{73.4}_{\pm0.2}$ | $\underline{96.9}_{\pm0.2}$ | $\mathbf{93.1}_{\pm0.4}$ | $\mathbf{89.1}_{\pm0.4}$ | $\mathbf{84.3}_{\pm0.3}$ |

analyzing complex cellular processes (Ding & Regev, 2021), especially for computing OT distance (Bellazzi et al., 2021; Tran et al., 2021; Bunne et al., 2023). For instance, in developmental biology, genes associated with stem cell differentiation follow a hierarchical pattern (Kang et al., 2012). Revealing the latent gene hierarchy is essential for accurately computing the TWD between cells, as it captures the underlying biological relationships and functional similarities.

We test two scRNA-seq datasets in Dumitrascu et al. (2021), including the Zeisel and CBMC. Detailed descriptions of the datasets are reported in App. D. We compare the proposed TWD to the same baselines as in Sec. 5.2. Instead of WMD, we include the Gene Mover's Distance (GMD) (Bellazzi et al., 2021), a gene-based OT distance using Gene2Vec (Du et al., 2019). The cell classification tasks are performed in the same way as in Sec. 5.1 and Sec. 5.2. Tab. 1 shows the mean and the standard deviation of the cell classification accuracy. Our TWD outperforms the competing methods by a large margin. This suggests that the proposed TWD effectively transforms the latent tree-like relationships in gene space, improving the cell classification accuracy compared to random sampling trees and trees constructed by graph-based and HC methods. Notably, our TWD differentiates the cell types better than pre-trained GMD, which uses gene proximity, while ours is geometrically intrinsic.

## 6 CONCLUSIONS

We presented a novel distance between samples with a latent feature hierarchy. The computation of the distance is efficient and can accommodate comparisons of large datasets. We show experimental results on various applications, where our method outperforms competing baselines. Theoretically, the proposed TWD provably recovers the TWD for data samples with a latent feature hierarchy without prior knowledge about the hidden feature hierarchy and based solely on the observational data matrix. We note that the construction of the proposed distance consists of a new algorithm for decoding a binary tree based on hyperbolic embeddings, relying on the analogy between the exponential growth of the metric in high-dimensional hyperbolic spaces and the exponential expansion of trees. This tree decoding is general and can be used for decoding trees in broader contexts.

Our work is conceptually different from existing TWD works in two main aspects. First, we present an unconventional use of the TWD. In addition to speeding up the computation of the Wasserstein distance, we also use the tree to represent the feature hierarchy. Second, we consider and recover the latent feature hierarchy by a novel hyperbolic tree decoding algorithm, which differs from existing hierarchical (continuous) representation learning methods that consider (known) sample hierarchy.

## ETHICS STATEMENT

This paper presents new machine learning techniques using publicly available and commonly used benchmarking datasets. These datasets are fully anonymized and do not contain any personal or sensitive information. We respect the privacy and integrity of all data, and we do not foresee any negative societal impact from our methods. We have no conflicts of interest to disclose.

## REPRODUCIBILITY STATEMENT

We provide a detailed overview of our experimental setup, including the data splits, hyperparameter configurations, and the criteria for their selection. Additionally, we describe the type of classifier used and other relevant training details. For a complete explanation, please see App. D. Our code is available at `https://github.com/ya-wei-eileen-lin/TWDwithLatentFeatureHierarchy`.

## ACKNOWLEDGMENTS

We thank the anonymous reviewers for their insightful feedback. We thank Gal Maman for proofreading an earlier version of the paper. The work of YEL and RT was supported by the European Union's Horizon 2020 research and innovation programme under grant agreement No. 802735-ERC-DIFFOP. The work of RC was supported by the US Air Force Office of Scientific Research (AFOSR MURI FA9550-21-1-0317). The work of GM was supported by NSF CCF-2217058.

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

# A ADDITIONAL BACKGROUND

While we have covered the most relevant background in the main paper, here we present additional background on the Wasserstein distance, diffusion geometry, hyperbolic geometry, and trees.

## A.1 WASSERSTEIN DISTANCE

The application of the Wasserstein distance spans various fields, including enhancing generative models in machine learning (Arjovsky et al., 2017; Kolouri et al., 2018), aligning and comparing color distributions in computer graphics (Solomon et al., 2015; Lavenant et al., 2018), and serving as a robust similarity metric in data analysis (Bellazzi et al., 2021; Kusner et al., 2015; Yair et al., 2017).

Let $\mu_1$ and $\mu_2$ be two probability distributions on a measurable space $\Omega$ with a ground metric $d$, $\Pi(\mu_1, \mu_2)$ be the set of couplings $\pi$ on the product space $\Omega \times \Omega$ such that $\pi(B_1 \times \Omega) = \mu_1(B_1)$ and $\pi(\Omega \times B_2) = \mu_2(B_2)$ for any sets $B_1, B_2 \subset \Omega$. The Wasserstein distance (Villani, 2009) between $\mu_1$ and $\mu_2$ is defined by

$$\mathrm{OT}(\mu_1, \mu_2, d) := \inf_{\pi \in \Pi(\mu_1, \mu_2)} \int_{\Omega \times \Omega} d(x, y) \mathrm{d}\pi(x, y). \tag{4}$$

When replacing the inf in Eq. (4) with the $\arg\min$, the obtained solution $\pi$ is referred to as the Optimal Transport (OT) plan (Monge, 1781; Kantorovich, 1942). When computing the Wasserstein distance in Eq. (4) between two *discrete* probabilities with $m$ points, given by

$$\mathrm{OT}(\boldsymbol{\mu}_1, \boldsymbol{\mu}_2, \mathbf{C}) := \min_{\boldsymbol{\Phi} \in \mathbb{R}^{m \times m}} \langle \boldsymbol{\Phi}, \mathbf{C} \rangle \text{ s.t. } \begin{cases} \boldsymbol{\Phi} \mathbf{1}_m = \boldsymbol{\mu}_1, \\ \boldsymbol{\Phi}^\top \mathbf{1}_m = \boldsymbol{\mu}_2, \end{cases} \tag{5}$$

where $\mathbf{C} \in \mathbb{R}^{m \times m}$ is the ground pairwise distance matrix between the coordinates of the discrete probabilities, it can be computed in $O(m^3 \log m)$ (Peyré et al., 2019), preventing its use of OT for large-scale datasets (Bonneel et al., 2011). Therefore, existing works have been proposed to reduce this complexity. Among them, a notable work is the Sinkhorn algorithm (Cuturi, 2013; Chizat et al., 2020), which approximates the OT in quadratic time.

## A.2 DIFFUSION GEOMETRY

Diffusion geometry (Coifman & Lafon, 2006) is a mathematical framework designed to analyze high-dimensional data points $\{\mathbf{a}_j \in \mathbb{R}^n\}_{j=1}^m$ by effectively capturing their underlying geometric structures (Belkin & Niyogi, 2008). It focuses on analyzing the similarity between data points through the process of diffusion propagation. Specifically, the constructed diffusion operator $\mathbf{P} \in \mathbb{R}^{m \times m}$ derived from the observational data is associated with the heat kernel and the Laplacian of the underlying manifold (Belkin & Niyogi, 2008). Let $\mathbf{Q}(j, j') = \exp\left(-d^2(j, j')/\epsilon\right)$ be the $(j, j')$th element of the pairwise affinity matrix between the data points. In the limit $m \to \infty$ and $\epsilon \to 0$, the diffusion operator $\mathbf{P}$ converges to the heat kernel $\exp(-\Delta)$, where $\Delta$ is the Laplace–Beltrami operator on the manifold (Coifman & Lafon, 2006). By applying eigendecomposition to the diffusion operator $\mathbf{P}$, we obtain a sequence of eigenpairs $\{\nu_j, \boldsymbol{\varphi}_j\}_{j=1}^m$ with positive decreasing eigenvalues $1 = \nu_1 \geq \nu_2 \geq \ldots \geq \nu_m > 0$. The diffusion maps (DM) in $m' << m$ dimensions at the diffusion time $t$ embeds a point $\mathbf{a}_j$ into $\mathbb{R}^{m'}$ by

$$\boldsymbol{\Psi}_{m'} : \mathbf{a}_j \mapsto [\nu_2^t \boldsymbol{\varphi}_2, \nu_3^t \boldsymbol{\varphi}_3, \ldots, \nu_{m'+1}^t \boldsymbol{\varphi}_{m'+1}]^\top. \tag{6}$$

It is shown that the Euclidean distances defined in DM between any two points $\|\boldsymbol{\Psi}_{m'}(\mathbf{y}_j) - \boldsymbol{\Psi}_{m'}(\mathbf{y}_{j'})\|_2$ approximate the following diffusion distances

$$d_{\mathrm{DM},t}(j, j') = \left\| \mathbf{P}^t \mathbf{e}_j - \mathbf{P}^t \mathbf{e}_{j'} \right\|_2 \tag{7}$$

at the diffusion time $t$.

The diffusion operator has been shown to exhibit favorable convergence properties (Coifman & Lafon, 2006). In the limits, as the number of features $m$ increases toward infinity, the scaling parameter $\epsilon$ approaches zero, and the operator $\mathbf{P}^{t/\epsilon}$ converges pointwise to the Neumann heat kernel $\mathbf{H}_t = \exp(-t\Delta)$, associated with the underlying manifold, where $\Delta$ is the Laplace–Beltrami operator on the manifold. That is, the diffusion operator serves as a discrete approximation of the continuous heat kernel, thereby capturing the geometric structure of the manifold within a finite-dimensional framework (Coifman & Lafon, 2006; Singer, 2006; Belkin & Niyogi, 2008).

Note that the density-normalized affinity matrix was considered to mitigate the effect of non-uniform data sampling, which was proposed and shown theoretically in Coifman & Lafon (2006). In Section 3.1 in Coifman & Lafon (2006), the construction of a family of diffusion operators is introduced with a parameter $\alpha \in \mathbb{R}$ and the density $q$. In Section 3.4 in Coifman & Lafon (2006), it is shown that when $\alpha = 1$ (in our case $\widehat{\mathbf{Q}} = \mathbf{D}^{-1}\mathbf{Q}\mathbf{D}^{-1}$), the diffusion operator approximates the heat kernel, and this approximation is independent of the density $q$. This technique has been further explored and utilized in (Mishne et al., 2019; Katz et al., 2020; Shnitzer et al., 2022; 2024; Lin et al., 2025).

Diffusion geometry has mainly been utilized in manifold learning, leading to the development of multi-scale, low-dimensional representations and informative distances (Nadler et al., 2005; Coifman & Maggioni, 2006). Its utility has been demonstrated across various applications spanning various fields, including nonlinear stochastic dynamical systems (Shnitzer et al., 2020), multi-view dimensionality reduction (Lindenbaum et al., 2020b), shape recognition (Bronstein et al., 2010), wavelet analysis (Coifman & Maggioni, 2006), internal state values of long short-term memory (Kemeth et al., 2021), and supervised and semi-supervised learning tools (Mendelman & Talmon, 2025).

## A.3 HYPERBOLIC GEOMETRY

Hyperbolic geometry is a Riemannian manifold of a constant negative curvature (Lee, 2006). There are four commonly used models for hyperbolic space: the Poincaré disk model, the Lorentz model, the Poincaré half-space model, and the Beltrami-Klein model. These models are equivalent and isometric, representing the same geometry, and distances can be translated without distortion (Ratcliffe et al., 1994). In this work, we used the Poincaré half-space model to represent the continuous tree structure underlying the data because the exponential growth of the metric serves as a natural representation of the exponential propagation of the diffusion densities on a dyadic grid. Formally, the $n$-dimensional Poincaré half-space model $\mathbb{H}^n$ with constant negative curvature $-1$ is defined by $\mathbb{H}^n = \{\mathbf{x} \in \mathbb{R}^n | \mathbf{x}(n) > 0\}$ with the Riemannian metric tensor $ds^2 = (d\mathbf{x}^2(1) + d\mathbf{x}^2(2) + \ldots + d\mathbf{x}^2(n))/\mathbf{x}^2(n)$. The Riemannian distance for any $\boldsymbol{x}, \boldsymbol{y} \in \mathbb{H}^n$ is defined by $d_{\mathbb{H}^n}(\mathbf{x}, \mathbf{y}) = 2\sinh^{-1}(\|\mathbf{x} - \mathbf{y}\|_2 / (2\sqrt{\mathbf{x}(n)\mathbf{y}(n)}))$. The distance function $d_{\mathbb{H}^n}(\cdot, \cdot)$ in the hyperbolic space $\mathbb{H}^n$ gives rise to two distinct types of geodesic paths (Ratcliffe et al., 1994): straight lines that are parallel to the $n$-th axis, and semi-circles that are perpendicular to the boundary of the plane $\mathbb{R}^{n-1}$.

The Gromov product (Gromov, 1987) is used to define $\delta$-hyperbolic spaces in hyperbolic geometry.

**Definition A.1** ($\delta$-Hyperbolic Space). *A metric space $(\mathcal{X}, d_{\mathcal{X}})$ is $\delta$-hyperbolic space (Gromov, 1987) if there exists $\delta \geq 0$ such that for any four points $x_1, x_2, x_3, x_4 \in \mathcal{X}$, we have*

$$d_{\mathcal{X}}(x_4, x_1) + d_{\mathcal{X}}(x_2, x_3) \leq \max\{d_{\mathcal{X}}(x_1, x_2) + d_{\mathcal{X}}(x_3, x_4), d_{\mathcal{X}}(x_1, x_3) + d_{\mathcal{X}}(x_2, x_4)\} + 2\delta. \tag{8}$$

**Definition A.2** (Gromov Product (Gromov, 1987)). *Let $(\mathcal{X}, d_{\mathcal{X}})$ be a metric space. The Gromov product (Gromov, 1987) of any two points $x_1, x_2 \in \mathcal{X}$ w.r.t a reference point $x_3 \in \mathcal{X}$ is defined by*

$$\langle x_1, x_2 \rangle_{x_3} = \frac{1}{2}\left(d_{\mathcal{X}}(x_1, x_3) + d_{\mathcal{X}}(x_2, x_3) - d_{\mathcal{X}}(x_1, x_2)\right). \tag{9}$$

## A.4 PRELIMINARIES ON TREES

**Definition A.3.** *Consider an undirected weighted graph $G = (V, E, \mathbf{A})$, where $V = \{1, 2, \ldots, m\}$ is the node set, $E$ is the edge set, and $\mathbf{A} \in \mathbb{R}^{m \times m}$ is the non-negative edge weight matrix. The shortest path metric $d_G(j, j')$ of any two nodes $j, j' \in V$ is defined by the length of the shortest path on the graph $G$ from node $j$ to node $j'$.*

**Lemma A.1.** *The tree metric is $0$-hyperbolic.*

**Definition A.4.** *A metric $d : V \times V \to \mathbb{R}$ is a tree metric if there exists a weighted tree $T = (V, E, \mathbf{A})$ such that for every $j, j' \in V$, the metric $d(j, j')$ is equal to the shortest path metric $d_T(j, j')$.*

**Definition A.5.** *A balanced binary tree is a tree where each node has a degree of either 1, making it a leaf node, or 3, making it an internal node. This tree structure can be either empty or consist of a root node linked to two disjoint binary trees, the left and right subtrees. In a binary tree with $m$ leaves, there are exactly $m - 1$ internal nodes.*

**Definition A.6.** *A rooted and balanced binary tree is a specific type of binary tree in which one internal node, designated as the root, has a degree of exactly 2, while the other internal nodes have a degree of 3. The remaining nodes are leaf nodes with a degree of 1.*

**Definition A.7** (Ultrametric). *A metric $d_u : V \times V \to \mathbb{R}$ is ultrametric if the triangle inequality for any three points $j_1, j_2, j_3 \in V$ is strengthened by the ultrametric inequality $d_u(j_1, j_2) \leq \max\{d_u(j_1, j_3), d_u(j_2, j_3)\}$.*

**Remark A.1.** *If a tree is ultrametric, the distances of all the leaves to the root are identical.*

**Definition A.8** (LCA Relation (Wang & Wang, 2018).)**.** *Given a tree $T$, we say that the LCA relation $[j_2 \vee j_3 \sim j_1]_T$ holds in $T$, if the LCA of $j_2$ and $j_3$, denoted as $j_2 \vee j_3$, is a proper descendant of the LCA of $j_2$, $j_3$ and $j_1$, denoted as $j_2 \vee j_3 \vee j_1$.*

**Remark A.2.** *While very small or specific types (e.g., star-shaped) of trees may be isometrically embedded in Euclidean space, general trees or hierarchical structures cannot, and embedding them into Euclidean spaces results in larger distortion. For larger or more complex trees, hyperbolic space often serves as a more natural embedding space, preserving their hierarchical structure and distances.*

## B  ADDITIONAL RELATED WORK

Here, we provide more details about related methods and objectives.

**Wasserstein Distance on Hyperbolic Manifold.** Wasserstein distance learning in hyperbolic spaces builds upon adapting the Wasserstein distance (Villani, 2009) to the structure of hyperbolic geometry (Lee, 2006). This adaptation involves redefining the cost function in the optimization problem to incorporate the geodesic distance within the hyperbolic manifold. The resulting Wasserstein metric maintains its intuitive interpretation as a measure of the "effort" required to transform one distribution into another while under the manifold constraints of hyperbolic structure. It has proven valuable in various fields, for example, in shape analysis (Shi et al., 2016) and hierarchy matching (Alvarez-Melis et al., 2020; Hoyos-Idrobo, 2020). Recently, the work in Bonet et al. (2023) proposed extending the sliced-Wasserstein distance (Rabin et al., 2012) to hyperbolic representation, enabling efficient computation of the Wasserstein distance. Our work, however, diverges fundamentally from these approaches. We focus on observational data that inherently possess a hierarchy rather than probability distributions defined on hyperbolic spaces. Additionally, our focus is on the TWD, aiming for efficient computation while addressing the specific challenges posed by the hidden feature hierarchy.

**Existing TWDs.** The TWD is a popular method to reduce the computational complexity associated with calculating the Wasserstein distance on the Euclidean metric. This approach approximates the original Euclidean metric using a tree metric, as described in several studies (Indyk & Thaper, 2003; Le et al., 2019; Evans & Matsen, 2012; Leeb, 2018; Takezawa et al., 2021; Yamada et al., 2022). The resulting methods provide a coarser approximation of OT in an Euclidean space, facilitating more efficient computations from cubic complexity to linear complexity. One notable method is the Quadtree (Indyk & Thaper, 2003), which creates a random tree by recursively dividing hypercubes. This tree is then used to compute the TWD. Flowtree (Backurs et al., 2020) modifies the Quadtree method by focusing on the optimal flow and evaluating the cost of this flow within the ground metric. The Tree-Sliced Wasserstein Distance (TSWD) (Le et al., 2019) represents another TWD technique. It computes the average of TWDs by randomly sampling trees. The number of sampling trees is a hyperparameter, commonly set to one, five, or ten, resulting in the variants TSWD-1, TSWD-5, and TSWD-10. Further advancements are presented in Yamada et al. (2022), which introduce WQTWD and WCTWD. These methods employ a Quadtree or clustering-based tree to approximate the Wasserstein distances by optimizing the weights on the tree. Lastly, UltraTree (Chen et al., 2024) constructs an ultrametric tree (Johnson, 1967) by minimizing the OT regression cost, aiming to approximate the Wasserstein distance within the original metric space.

Our approach, however, departs significantly from these existing methods. While we also aim to accelerate the computation of the Wasserstein distance using a tree, we leverage the tree structure to introduce and recover the latent feature hierarchy. Specifically, our novel hyperbolic tree decoding algorithm is designed to recover hidden feature hierarchies. Our TWD approximates the Wasserstein distance incorporating the latent feature hierarchy, rather than relying on the original metric in Euclidean space.

**Tree Construction by Graph-Based and HC Methods.** Another line of related work involves constructing trees using existing graph-based algorithms, such as Minimum Spanning Tree (MST) (Alon et al., 1995), or hierarchical clustering (HC) methods, particularly those that leverage hyperbolic geometry. Generic graph-based algorithms, like those used in existing TWDs, aim to approximate the original metric with a tree metric. In the context of hierarchical clustering, various methods have been proposed. UltraFit (Chierchia & Perret, 2019) addresses an "ultrametric fitting" problem using Euclidean embeddings. gHHC (Monath et al., 2019) operates under the assumption that partial information about the optimal clustering, specifically the hyperbolic embeddings of the leaves, is known. HHC (Chami et al., 2020) decodes a tree from a two-dimensional hyperbolic embedding for hierarchical clustering by optimizing the continuous Dasgupta's cost (Dasgupta, 2016). While these methods may offer theoretical guarantees regarding clustering quality, they do not provide a theoretical guarantee for recovering the latent hierarchy underlying high-dimensional features, as our method does. Our approach uniquely focuses on revealing the latent feature hierarchy, ensuring a more accurate representation of the latent hierarchical structure underlying high-dimensional features.

**Fréchet Mean on Manifolds.** The Fréchet mean (Fréchet, 1948) is a powerful tool for data analysis on Riemannian manifolds (Lee, 2006), including hyperbolic, spherical, and various matrix manifolds. Its application extends across a diverse range of fields, including batch normalization (Brooks et al., 2019; Lou et al., 2020), batch effect removal (Lin et al., 2021; 2024), domain adaptation (Yair et al., 2019), shape analysis and object recognition (McLeod et al., 2012), to name but a few. In our work, we follow this line of work, using the Fréchet mean for constructing the HD-LCA in a product manifold of hyperbolic spaces (see Def. 1 and Def. 3). Then, this HD-LCA serves as a primary building block in our binary tree decoding (Alg. 1), which in turn, is used in the computation of the proposed TWD (Alg. 2).

## C    THEORETICAL ANALYSIS AND PROOFS

We note that the numbering of the statements corresponds to the numbering used in the paper, and we have also included several separately numbered propositions and lemmas that are used in supporting the proofs presented. We restate the claim of each statement for convenience.

### C.1    PROOF OF PROPOSITION 1

**Proposition 1.**    *The hyperbolic LCA $\mathbf{z}_j^k \vee \mathbf{z}_{j'}^k$ in Def. 1 has a closed-form solution, given by $\mathbf{z}_j^k \vee \mathbf{z}_{j'}^k =$*
$$\left[ \tfrac{1}{2} \left( \boldsymbol{\psi}_j^k + \boldsymbol{\psi}_{j'}^k \right)^\top, \mathtt{proj}(\mathbf{z}_j^k \vee \mathbf{z}_{j'}^k) \right]^\top, \textit{where } \mathtt{proj}(\mathbf{z}_j^k \vee \mathbf{z}_{j'}^k) = \left\| \left[ \tfrac{1}{2} \left( \boldsymbol{\psi}_j^k - \boldsymbol{\psi}_{j'}^k \right)^\top, 2^{k/2-2} \right]^\top \right\|_2.$$

*Proof.* Let $\gamma_{\mathbf{z}_j^k \rightsquigarrow \mathbf{z}_{j'}^k}(t)$ be the unique geodesic path from $\mathbf{z}_j^k$ to $\mathbf{z}_{j'}^k$ such that $\gamma_{\mathbf{z}_j^k \rightsquigarrow \mathbf{z}_{j'}^k}(0) = \mathbf{z}_j^k$ and $\gamma_{\mathbf{z}_j^k \rightsquigarrow \mathbf{z}_{j'}^k}(1) = \mathbf{z}_{j'}^k$. Note that the geodesic path $\gamma_{\mathbf{z}_j^k \rightsquigarrow \mathbf{z}_{j'}^k}$ is a semi-circle, and the last coordinate of the points $\mathbf{z}_j^k$ and $\mathbf{z}_{j'}^k$ are the same, i.e., $\mathbf{z}_j^k(m+1) = \mathbf{z}_{j'}^k(m+1) = 2^{k/2-2}$. Therefore, the center point of the semi-circle projected on the plane $\mathbb{R}^m$ is $\mathbf{c}_{\mathbf{z}_j^k \rightsquigarrow \mathbf{z}_{j'}^k} = \left[ \tfrac{1}{2} \left( \boldsymbol{\psi}_j^k + \boldsymbol{\psi}_{j'}^k \right)^\top, 0 \right]^\top$. Moreover, the radius of the geodesic path is given by

$$\mathtt{proj}(\mathbf{z}_j^k \vee \mathbf{z}_{j'}^k) = \left\| \mathbf{c}_{\mathbf{z}_j^k \rightsquigarrow \mathbf{z}_{j'}^k} - \mathbf{z}_j^k \right\|_2$$
$$= \left\| \mathbf{c}_{\mathbf{z}_j^k \rightsquigarrow \mathbf{z}_{j'}^k} - \mathbf{z}_{j'}^k \right\|_2$$
$$= \left\| \left[ \frac{1}{2} \left( \boldsymbol{\psi}_j^k - \boldsymbol{\psi}_{j'}^k \right)^\top, 2^{k/2-2} \right]^\top \right\|_2.$$

The Riemannian mean defined using the Fréchet mean (Fréchet, 1948) has a closed-form expression, and is located at the midpoint of the geodesic curve

$$\mathbf{z}_j^k \vee \mathbf{z}_{j'}^k = \left[ \frac{1}{2} \left( \boldsymbol{\psi}_j^k + \boldsymbol{\psi}_{j'}^k \right)^\top, \mathtt{proj}(\mathbf{z}_j^k \vee \mathbf{z}_{j'}^k) \right]^\top.$$

□

**Lemma C.1.** *The radius* $\mathtt{proj}(\mathbf{z}_j^k \vee \mathbf{z}_{j'}^k)$ *is lower-bounded by the diffusion distance at the diffusion time* $2^{-k}$, *given by*

$$d_{DM,2^{-k}}(j,j') \leq \mathtt{proj}(\mathbf{z}_j^k \vee \mathbf{z}_{j'}^k). \tag{10}$$

## C.2 PROOF OF PROPOSITION 2

**Proposition 2.** *For any* $k_1 \leq k_2$, $2^{-(k_2-k_1)} \simeq d_{\mathbb{H}^{m+1}}(\mathbf{z}_j^{k_2} \vee \mathbf{z}_{j'}^{k_2}, \mathbf{z}_j^{k_2})/d_{\mathbb{H}^{m+1}}(\mathbf{z}_j^{k_1} \vee \mathbf{z}_{j'}^{k_1}, \mathbf{z}_j^{k_1})$.

*Proof.* By Prop. 1, we have $\mathbf{z}_j^{k_1} \vee \mathbf{z}_{j'}^{k_1} = \left[\frac{1}{2}\left(\boldsymbol{\psi}_j^{k_1} + \boldsymbol{\psi}_{j'}^{k_1}\right)^\top, \mathtt{proj}(\mathbf{z}_j^{k_2} \vee \mathbf{z}_{j'}^{k_1})\right]^\top$ and $\mathbf{z}_j^{k_2} \vee \mathbf{z}_{j'}^{k_2} =$
$\left[\frac{1}{2}\left(\boldsymbol{\psi}_j^{k_2} + \boldsymbol{\psi}_{j'}^{k_2}\right)^\top, \mathtt{proj}(\mathbf{z}_j^{k_2} \vee \mathbf{z}_{j'}^{k_2})\right]^\top$, respectively. At the $k_1$-th level, the Riemannian distance between the $k_1$-th hyperbolic LCA $\mathbf{z}_j^{k_1} \vee \mathbf{z}_{j'}^{k_1}$ and $\mathbf{z}_j^{k_1}$ at the $k_1$-th level is given by

$$d_{\mathbb{H}^{m+1}}(\mathbf{z}_j^{k_1} \vee \mathbf{z}_{j'}^{k_1}, \mathbf{z}_j^{k_1}) = 2\sinh^{-1}\left(\left\|\mathbf{z}_j^{k_1} \vee \mathbf{z}_{j'}^{k_1} - \mathbf{z}_j^{k_1}\right\|_2 \Big/ \left(2\sqrt{\mathbf{z}_j^{k_1} \vee \mathbf{z}_{j'}^{k_1}(m+1)\mathbf{z}_j^{k_1}(m+1)}\right)\right).$$

Note that $\mathbf{z}_j^{k_1}(m+1) = 2^{k_1/2-2}$ and by Prop. 1, we have $\mathbf{z}_j^{k_1} \vee \mathbf{z}_{j'}^{k_1}(m+1) = \mathtt{proj}(\mathbf{z}_j^{k_1} \vee \mathbf{z}_{j'}^{k_1})$. We have the same relation at the $k_2$ level. Therefore, for any $k_1 \leq k_2$, we have

$$d_{\mathbb{H}^{m+1}}(\mathbf{z}_j^{k_2} \vee \mathbf{z}_{j'}^{k_2}, \mathbf{z}_j^{k_2})/d_{\mathbb{H}^{m+1}}(\mathbf{z}_j^{k_1} \vee \mathbf{z}_{j'}^{k_1}, \mathbf{z}_j^{k_1})$$

$$= \frac{\sinh^{-1}\left(\left\|\mathbf{z}_j^{k_2} \vee \mathbf{z}_{j'}^{k_2} - \mathbf{z}_j^{k_2}\right\|_2 \Big/ \left(2\sqrt{\mathbf{z}_j^{k_2} \vee \mathbf{z}_{j'}^{k_2}(m+1)\mathbf{z}_j^{k_2}(m+1)}\right)\right)}{\sinh^{-1}\left(\left\|\mathbf{z}_j^{k_1} \vee \mathbf{z}_{j'}^{k_1} - \mathbf{z}_j^{k_1}\right\|_2 \Big/ \left(2\sqrt{\mathbf{z}_j^{k_1} \vee \mathbf{z}_{j'}^{k_1}(m+1)\mathbf{z}_j^{k_1}(m+1)}\right)\right)}$$

$$= \sinh^{-1}\left(2^{-k_2/2+1}\left\|\boldsymbol{\psi}_j^{k_2} - \boldsymbol{\psi}_{j'}^{k_2}\right\|_2\right) \Big/ \sinh^{-1}\left(2^{-k_1/2+1}\left\|\boldsymbol{\psi}_j^{k_1} - \boldsymbol{\psi}_{j'}^{k_1}\right\|_2\right)$$

$$\overset{(1)}{\geq} \sinh^{-1}\left(2^{-k_2/2+1}\left\|\boldsymbol{\psi}_j^{k_2} - \boldsymbol{\psi}_{j'}^{k_2}\right\|_2\right) \Big/ \sinh^{-1}\left(2^{-k_1/2+1}\left\|\boldsymbol{\psi}_j^{k_2} - \boldsymbol{\psi}_{j'}^{k_2}\right\|_2\right)$$

$$= 2^{-(k_2-k_1)},$$

where the transition (1) is due to $\left\|\boldsymbol{\psi}_j^{k_1} - \boldsymbol{\psi}_{j'}^{k_1}\right\|_2 \leq \left\|\boldsymbol{\psi}_j^{k_2} - \boldsymbol{\psi}_{j'}^{k_2}\right\|_2$ for any $k_1 \leq k_2$. Similarly, we have

$$d_{\mathbb{H}^{m+1}}(\mathbf{z}_j^{k_2} \vee \mathbf{z}_{j'}^{k_2}, \mathbf{z}_j^{k_2})/d_{\mathbb{H}^{m+1}}(\mathbf{z}_j^{k_1} \vee \mathbf{z}_{j'}^{k_1}, \mathbf{z}_j^{k_1})$$

$$= \frac{\sinh^{-1}\left(\left\|\mathbf{z}_j^{k_2} \vee \mathbf{z}_{j'}^{k_2} - \mathbf{z}_j^{k_2}\right\|_2 \Big/ \left(2\sqrt{\mathbf{z}_j^{k_2} \vee \mathbf{z}_{j'}^{k_2}(m+1)\mathbf{z}_j^{k_2}(m+1)}\right)\right)}{\sinh^{-1}\left(\left\|\mathbf{z}_j^{k_1} \vee \mathbf{z}_{j'}^{k_1} - \mathbf{z}_j^{k_1}\right\|_2 \Big/ \left(2\sqrt{\mathbf{z}_j^{k_1} \vee \mathbf{z}_{j'}^{k_1}(m+1)\mathbf{z}_j^{k_1}(m+1)}\right)\right)}$$

$$= \sinh^{-1}\left(2^{-k_2/2+1}\left\|\boldsymbol{\psi}_j^{k_2} - \boldsymbol{\psi}_{j'}^{k_2}\right\|_2\right) \Big/ \sinh^{-1}\left(2^{-k_1/2+1}\left\|\boldsymbol{\psi}_j^{k_1} - \boldsymbol{\psi}_{j'}^{k_1}\right\|_2\right)$$

$$\overset{(1)}{\leq} 2 \cdot 2^{-k_1/2+1} \Big/ \frac{\eta}{2} \cdot 2^{-k_1/2+1}$$

where the transition (1) is because there is a lower bound $\eta$ of any Hellinger distance for $j \neq j'$. Therefore, we show that $2^{-(k_2-k_1)} \simeq d_{\mathbb{H}^{m+1}}(\mathbf{z}_j^{k_2} \vee \mathbf{z}_{j'}^{k_2}, \mathbf{z}_j^{k_2})/d_{\mathbb{H}^{m+1}}(\mathbf{z}_j^{k_1} \vee \mathbf{z}_{j'}^{k_1}, \mathbf{z}_j^{k_1})$. $\square$

## C.3 PROOF OF PROPOSITION 3

**Proposition 3.** *The HD-LCA* $\overline{\mathbf{h}}_{j,j'}$ *in Def. 3 has a closed-form solution, given by* $\overline{\mathbf{h}}_{j,j'} = [0,\ldots,0,a_{j,j'}]^\top$, *where* $a_{j,j'} = \sqrt[(K_c+1)]{\mathtt{proj}(\mathbf{z}_j^0 \vee \mathbf{z}_{j'}^0)\cdots\mathtt{proj}(\mathbf{z}_j^{K_c} \vee \mathbf{z}_{j'}^{K_c})}$.

*Proof.* The orthogonal projections $\{\mathbf{o}_{j,j'}^k\}_{k=0}^{K_c}$ lie on the unique geodesic path $\gamma_0$ that is on the $(m+1)$-axis in $\mathbb{H}^{m+1}$. Note that the geodesic distance joining any two points $\mathbf{o}_{j,j'}^{k_1}$ and $\mathbf{o}_{j,j'}^{k_2}$ is

given by $\left| \ln \left( \frac{\mathtt{proj}(\mathbf{z}_j^{k_1} \vee \mathbf{z}_{j'}^{k_1})}{\mathtt{proj}(\mathbf{z}_j^{k_2} \vee \mathbf{z}_{j'}^{k_2})} \right) \right|$ (Stahl, 1993; Udriste, 2013). The Fréchet mean (Fréchet, 1948) of $\{\mathbf{o}_{j,j'}^k\}_{k=0}^{K_c}$ is the point that minimizes the sum of squared distances to all points in the set. For our set of points, the Fréchet mean mean will have the form $[0, \dots, 0, a_{j,j'}]^\top$, where $a_{j,j'} \in \mathbb{R}_{>0}$. We want to find $a_{j,j'}$ that minimizes

$$\overline{\mathbf{h}}_{j,j'} \coloneqq \underset{\mathbf{h} \in \mathbb{H}^{m+1}}{\arg\min} \sum_{k=0}^{K_c} d_{\mathbb{H}^{m+1}}^2(\mathbf{h}, \mathbf{o}_{j,j'}^k)$$

$$= \underset{\mathbf{h}=[0,\dots,0,a]^\top \in \mathbb{H}^{m+1}}{\arg\min} \sum_{k=0}^{K_c} d_{\mathbb{H}^{m+1}}^2(\mathbf{h}, \mathbf{o}_{j,j'}^k)$$

$$= \underset{\mathbf{h}=[0,\dots,0,a]^\top \in \mathbb{H}^{m+1}}{\arg\min} \sum_{k=0}^{K_c} \left| \ln \left( \frac{\mathtt{proj}(\mathbf{z}_j^k \vee \mathbf{z}_{j'}^k)}{a} \right) \right|^2.$$

Therefore, the geometric mean $\overline{\mathbf{h}}_{j,j'}$ is obtained by setting $a_{j,j'} = {}^{(K_c+1)}\sqrt{\mathtt{proj}(\mathbf{z}_j^0 \vee \mathbf{z}_{j'}^0) \cdots \mathtt{proj}(\mathbf{z}_j^{K_c} \vee \mathbf{z}_{j'}^{K_c})}.$

$\square$

## C.4 PROOF OF PROPOSITION 4

**Proposition 4.** *Let $r$ be the root node of $B$ obtained in Alg. 1 and $\mathbf{z}_r$ be the corresponding embedding in $\mathcal{M}$. The HD-LCA depth is equivalent to the Gromov product (Gromov, 1987), i.e., $d_{\mathcal{M}}(\mathbf{z}_r, \overline{\mathbf{h}}_{j,j'}) \simeq \langle j, j' \rangle_r$, where $\langle j, j' \rangle_r = \frac{1}{2}(d_B(j,r) + d_B(j',r) - d_B(j,j'))$.*

*Proof.* Let $\mathbf{z}_j$ and $\mathbf{z}_{j'}$ be two points in $\mathcal{M}$. We denote $\gamma_{\mathbf{z}_j \rightsquigarrow \mathbf{z}_{j'}}(t)$ the unique geodesic path from $\mathbf{z}_j$ to $\mathbf{z}_{j'}$ such that $\gamma_{\mathbf{z}_j \rightsquigarrow \mathbf{z}_{j'}}(0) = \mathbf{z}_j$ and $\gamma_{\mathbf{z}_j \rightsquigarrow \mathbf{z}_{j'}}(1) = \mathbf{z}_{j'}$. Following Bowditch (2007), for any point $\mathbf{z}_{j''} \in \mathcal{M}$, we have $d_{\mathcal{M}}(\mathbf{z}_{j''}, \mathbf{z}) \geq \langle j, j' \rangle_{j''}$, where $\mathbf{z} \in \gamma_{\mathbf{z}_j \rightsquigarrow \mathbf{z}_{j'}}(t)$. Note that if $\mathbf{z}_{j''} \in \gamma_{\mathbf{z}_j \rightsquigarrow \mathbf{z}_{j'}}(t)$, then $\langle j, j' \rangle_{j''} = 0$. In addition, let $\gamma_{\mathbf{z}_j \rightsquigarrow \mathbf{z}_{j''}}(t)$ and $\gamma_{\mathbf{z}_{j'} \rightsquigarrow \mathbf{z}_{j''}}(t)$ be the unique geodesic path from $\mathbf{z}_j$ to $\mathbf{z}_{j''}$ and the unique geodesic path from $\mathbf{z}_{j'}$ to $\mathbf{z}_{j''}$, respectively. Let $\triangle_{j,j',j''} = (\gamma_{\mathbf{z}_j \rightsquigarrow \mathbf{z}_{j'}}(t), \gamma_{\mathbf{z}_j \rightsquigarrow \mathbf{z}_{j''}}(t), \gamma_{\mathbf{z}_{j'} \rightsquigarrow \mathbf{z}_{j''}}(t))$ be the triangle connected by these geodesic paths. As $\mathcal{M}$ is 0-hyperbolic, the triangle $\triangle_{j,j',j''}$ is 0-centre and by triangle inequality, we have

$$d_{\mathcal{M}}(\mathbf{z}_j, \mathbf{z}) + d_{\mathcal{M}}(\mathbf{z}_{j'}, \mathbf{z}) \leq d_{\mathcal{M}}(\mathbf{z}_j, \mathbf{z}_{j''})$$
$$d_{\mathcal{M}}(\mathbf{z}_{j'}, \mathbf{z}) + d_{\mathcal{M}}(\mathbf{z}_{j''}, \mathbf{z}) \leq d_{\mathcal{M}}(\mathbf{z}_j', \mathbf{z}_{j''})$$
$$d_{\mathcal{M}}(\mathbf{z}_j, \mathbf{z}) + d_{\mathcal{M}}(\mathbf{z}_{j'}, \mathbf{z}) = d_{\mathcal{M}}(\mathbf{z}_j, \mathbf{z}_{j'}).$$

Therefore, $d_{\mathcal{M}}(\mathbf{z}_{j''}, \mathbf{z}) \leq \langle j, j' \rangle_{j''}$. By the direct application of the aforementioned two inequality, we have $d_{\mathcal{M}}(\mathbf{z}_r, \overline{\mathbf{h}}_{j,j'}) \simeq \langle j, j' \rangle_r$. $\square$

## C.5 PROOF OF THEOREM 1

**Theorem 1.** *For sufficiently large $K_c$ and $m$, there exist some constants $C_1, C_2 > 0$ such that $C_1 d_T(j_1, j_2) \leq d_B(j_1, j_2) \leq C_2 d_T(j_1, j_2)$ for $j_1, j_2 \in [m]$, where $d_B$ is the decoded tree metric $d_B$ in Alg. 1, $T$ is the ground truth latent tree assumed from Sec. 4.1, and $d_T$ is the hidden tree metric between features defined as the length of the shortest path on $T$ (i.e., the geodesic distances).*

*Proof.* We adopt the equivalence proof from Lin et al. (2023), which stems from Coifman & Lafon (2006) and (Leeb & Coifman, 2016). Note that diffusion operator $\mathbf{P}$ is typically used for manifold learning, where it be viewed as a discrete proxy of the underlying manifold. It is shown that the diffusion operator recovers the underlying manifold (Coifman & Lafon, 2006), i.e., $\mathbf{P} \overset{m \to \infty, \epsilon \to 0}{\longrightarrow} \mathbf{H} = \exp(-\Delta)$, where $\mathbf{H}$ is the Neumann heat kernel of the underlying manifold and $\Delta$ is the Laplace–Beltrami operator on the manifold. In addition, the diffusion operator $\mathbf{P}$ is based on a finite set of points, which is a discrete approximation of $\mathbf{H}$.

For a family of the operator $\{A^t\}_{t \in \mathbb{R}}$, the multi-scale metric using the inverse hyperbolic sine function of the scaled Hellinger measure is defined by

$$d_{A,\mathcal{M}}(x, x') := \sum_{k \geq 0} 2 \sinh^{-1} \left( 2^{-\frac{k}{2}+1} \left\| \sqrt{A^{2^{-k}}(x, \cdot)} - \sqrt{A^{2^{-k}}(x', \cdot)} \right\|_2 \right),$$

where $A^t f(x) = \int_{\mathcal{X}} A^t(x, x') f(x') dx'$. Our goal is to show that $d_B(j, j') \simeq d_{\mathbf{H},\mathcal{M}}(j, j')$, and conclude with Thm. C.1 (Lin et al., 2023) which states the equivalence $d_{\mathbf{H},\mathcal{M}}(j, j') \simeq d_T(j, j')$.

Note that for small enough $\epsilon > 0$, we have $d_{\mathbf{P},\mathcal{M}}(j, j') \simeq d_{\mathbf{H},\mathcal{M}}(j, j')$. Specifically, following Coifman & Lafon (2006), we have $\|\mathbf{P} - \mathbf{H}\|_{L^2(\mathcal{M})} \to 0$ and $\|\mathbf{P} - \mathbf{H}\|_{L^1(\mathcal{M})} \to 0$ as $\epsilon \to 0$. We denote $d_{\mathcal{M}}(j, j') = d_{\mathbf{P},\mathcal{M}}(j, j') \simeq d_{\mathbf{H},\mathcal{M}}(j, j')$ for abbreviation.

For any three features $j_1, j_2, j_3 \in [m]$, assume the HD-LCA relation $[j_2 \vee j_3 \sim j_1]_{\mathcal{M}}$ holds. Let $j_2 \vee j_3$ and $j_1 \vee j_3$ be LCAs of $(j_2, j_3)$ and $(j_1, j_3)$ on $B$ obtained in Alg. 1, respectively. We denote $\mathbf{z}_{j_2} \vee \mathbf{z}_{j_3}$ and $\mathbf{z}_{j_2} \vee \mathbf{z}_{j_3}$ the corresponding embeddings. By Prop. 1, we have

$$\mathbf{z}_{j_2} \vee \mathbf{z}_{j_3} = \left[ \left[ \frac{1}{2} \left( \boldsymbol{\psi}_{j_2}^0 + \boldsymbol{\psi}_{j_3}^0 \right)^\top, \texttt{proj}(\mathbf{z}_{j_2}^0 \vee \mathbf{z}_{j_3}^0) \right]^\top, \ldots, \left[ \frac{1}{2} \left( \boldsymbol{\psi}_{j_2}^k + \boldsymbol{\psi}_{j_3}^k \right)^\top, \texttt{proj}(\mathbf{z}_{j_2}^{K_c} \vee \mathbf{z}_{j_3}^{K_c}) \right]^\top \right]^\top$$

and

$$\mathbf{z}_{j_1} \vee \mathbf{z}_{j_3} = \left[ \left[ \frac{1}{2} \left( \boldsymbol{\psi}_{j_1}^0 + \boldsymbol{\psi}_{j_3}^0 \right)^\top, \texttt{proj}(\mathbf{z}_{j_1}^0 \vee \mathbf{z}_{j_3}^0) \right]^\top, \ldots, \left[ \frac{1}{2} \left( \boldsymbol{\psi}_{j_1}^k + \boldsymbol{\psi}_{j_3}^k \right)^\top, \texttt{proj}(\mathbf{z}_{j_1}^{K_c} \vee \mathbf{z}_{j_3}^{K_c}) \right]^\top \right]^\top.$$

Therefore, the tree distance between $j_1$ and $j_3$ is given by $d_B(j_1, j_3) = d_{\mathcal{M}}(\mathbf{z}_{j_1}, \mathbf{z}_{j_1} \vee \mathbf{z}_{j_3}) + d_{\mathcal{M}}(\mathbf{z}_{j_1} \vee \mathbf{z}_{j_3}, \mathbf{z}_{j_2} \vee \mathbf{z}_{j_3}) + d_{\mathcal{M}}(\mathbf{z}_{j_2} \vee \mathbf{z}_{j_3}, \mathbf{z}_{j_3})$. By triangle inequality, we have $d_B(j_1, j_3) \leq d_{\mathcal{M}}(\mathbf{z}_{j_1}, \mathbf{z}_{j_3})$. We denote $d_{\mathcal{M}}(j_1, j_2) = d_{\mathcal{M}}(\mathbf{z}_{j_1}, \mathbf{z}_{j_3})$ for simplicity. By Prop. 2 and Prop. 4, we have $c \cdot d_{\mathcal{M}}(j_1, j_3) \leq d_B(j_1, j_3)$. Moreover, since the HD-LCA relation $[j_2 \vee j_3 \sim j_1]_{\mathcal{M}}$ holds, the tree distance between $j_2, j_3 \in [m]$ on $B$ can be directly obtained by $d_B(j_2, j_3) = d_{\mathcal{M}}(j_2, j_3)$. As only one of the HD-LCA relations $[j_2 \vee j_3 \sim j_1]_{\mathcal{M}}, [j_1 \vee j_3 \sim j_2]_{\mathcal{M}}$, or $[j_1 \vee j_2 \sim j_3]_{\mathcal{M}}$ holds in $B$, the proof can be applied to any three points $j_1, j_2, j_3 \in [m]$ of which HD-LCA relation holds. Using Monte-Carlo integration, the convergence is with probability one as $K_c \to \infty$. For a finite set, we have a high probability bound on the convergence. Therefore, $d_B \simeq d_{\mathcal{M}} \simeq d_T$.

$\square$

**Theorem C.1** (Theorem 1 in Lin et al. (2023)). *For $\alpha \to \frac{1}{2}$ and sufficiently large $K_c$, there exists some constants $\widetilde{C}_1, \widetilde{C}_2 > 0$ such that $\widetilde{C}_1 d_T^{2\alpha} \leq d_{\mathcal{M}} \leq \widetilde{C}_2 d_T^{2\alpha}$.*

### C.6 PROOF OF THEOREM 2

**Theorem 2.** *For sufficiently large $K_c$ and $m$, there exist some constant $\widetilde{C}_1, \widetilde{C}_2 > 0$ such that $\widetilde{C}_1 \texttt{TW}(\mathbf{x}_i, \mathbf{x}_{i'}, T) \leq \texttt{TW}(\mathbf{x}_i, \mathbf{x}_{i'}, B) \leq \widetilde{C}_2 \texttt{TW}(\mathbf{x}_i, \mathbf{x}_{i'}, T)$ for all samples $i, i' \in [n]$.*

*Proof.* Let $\mu_i$ and $\mu_{i'}$ be two probability distributions supported on $T$ with a ground metric $d_T$, $\Pi(\mu_i, \mu_{i'})$ be the set of couplings $\pi$ on the product space $T \times T$ such that $\pi(B_1 \times T) = \mu_i(B_1)$ and $\pi(T \times B_2) = \mu_{i'}(B_2)$ for any sets $B_1, B_2 \subset T$. The Wasserstein distance (Villani, 2009) between $\mu_i$ and $\mu_{i'}$ is defined by

$$\texttt{OT}(\mu_i, \mu_{i'}, d_T) := \inf_{\pi \in \Pi(\mu_i, \mu_{i'})} \int_{T \times T} d(x, y) \mathrm{d}\pi(x, y). \tag{11}$$

Let $\mathcal{F}_{d_T}$ and $\|\cdot\|_{L_{d_T}}$ denote the Lipschitz functions w.r.t. $d_T$ and Lipschitz norm w.r.t. $d_T$, respectively. The Kantorovich-Rubinstein dual (Villani, 2009) of Eq. (11) is given by

$$\texttt{OT}(\mu_i, \mu_{i'}, d_T) = \sup_{\|f\|_{L_{d_T}} \leq 1} \int_T f(x) \mu_i(dx) - \int_T f(y) \mu_{i'}(dy).$$

By Evans & Matsen (2012), the witness function $f$ can be represented by a Borel function $g : T \to [-1, 1]$, i.e.,

$$\int_T f(x) \mu_i(dx) = \int_T g(z) \lambda(dz) \mu_i(\Gamma_T(z)),$$

where $\lambda$ is the unique Borel measure on $T$. Therefore, following Le et al. (2019), the Wasserstein distance $\mathtt{OT}(\mu_i, \mu_{i'}, d_T)$ can be written as

$$
\begin{aligned}
\mathtt{OT}(\mu_i, \mu_{i'}, d_T) &= \sup \int_T \left( \mu_i(\Gamma_T(z)) - \mu_{i'}(\Gamma_T(z)) \right) g(z) \lambda(dz) \\
&= \int_T \left| \mu_i(\Gamma_T(z)) - \mu_{i'}(\Gamma_T(z)) \right| \lambda(dz) \\
&= \sum_{v \in V} \omega_v \left| \sum_{u \in \Gamma_T(v)} \left( \mu_i(u) - \mu_{i'}(u) \right) \right| \\
&= \mathtt{TW}(\mu_i, \mu_{i'}, T).
\end{aligned}
$$

Note that based on Thm. 1, the decoded tree metric $d_B(\cdot, \cdot)$ and the underlying tree metric $d_T(\cdot, \cdot)$ are bilipschitz equivalent. Additionally, the TWD using the tree $T$ is the Wasserstein distance with ground pairwise distance $d_T$ (Evans & Matsen, 2012), i.e., $\mathtt{TW}(\mu_1, \mu_2, T) = \mathtt{OT}(\mu_1, \mu_2, d_T)$. Therefore, for any two discrete histograms $\mathbf{x}_i$ and $\mathbf{x}_{i'}$ supported on $T$, we have $\mathtt{TW}(\mathbf{x}_i, \mathbf{x}_{i'}, B) \simeq \mathtt{TW}(\mathbf{x}_i, \mathbf{x}_{i'}, T)$. $\quad\square$

## C.7 PROOF OF PROPOSITION 5

**Proposition 5.** *The TWD in Eq. (3) is bilipschitz equivalent to the sum of the multi-scale hyperbolic TWDs* $\mathtt{TW}(\mathbf{x}_i, \mathbf{x}_{i'}, B^k)$ *for* $k = 0, 1, \ldots, K_c$, *i.e.,* $\mathtt{TW}(\mathbf{x}_i, \mathbf{x}_{i'}, B) \simeq \sum_{k=0}^{K_c} \mathtt{TW}(\mathbf{x}_i, \mathbf{x}_{i'}, B^k)$.

*Proof.* By Alg. 3, the $k$-th rooted binary tree $B^k = (\widehat{V}_c^k, \widehat{E}_c^k, \widehat{\mathbf{A}}_c^k)$ is decoded based on the $k$-th scale hyperbolic LCA $\{\mathbf{z}_j^k \vee \mathbf{z}_{j'}^k\}_{j,j'}$, and the length of the edge is determined by the hyperbolic distance $d_{\mathbb{H}^{m+1}}$. By Prop. 2, for all $k \in \{0, 1, \ldots, K_c\}$, the tree metric of the $k$-th rooted binary tree between the features $j, j' \in [m]$ is bounded by

$$
\beta_k' d_{\widehat{B}_c^0}(j, j') \le d_{B^k}(j, j') \le \beta_k d_{\widehat{B}_c^0}(j, j'),
$$

where $\beta_k, \beta_k' > 0$, and $d_{\widehat{B}_c^0}(\cdot, \cdot)$ is the tree metric of $\widehat{B}_c^0$. Since the embedding distance is induced by the $\ell_1$ distance on the product of $K_c + 1$ hyperbolic spaces, the tree metric $d_B$ is the $\ell_1$ distance on the product of $K_c + 1$ rooted binary tree $\{B^k\}_{k=0}^{K_c}$, i.e., for any two features $j, j' \in [m]$, we have

$$
d_B(j, j') = \sum_{k=0}^{K_c} d_{B^k}(j, j').
$$

Incorporating the above equation with Prop. 2, the tree metric of the decoded tree $d_B$ is bounded by the tree metric $d_{\widehat{B}_c^0}$, given by

$$
d_{\widehat{B}_c^0}(j, j') \sum_{k=0}^{K_c} \beta_k' \le d_B(j, j') \le d_{\widehat{B}_c^0}(j, j') \sum_{k=0}^{K_c} \beta_k.
$$

Under the condition of uniform cost scaling in OT, we have

$$
\mathtt{OT}(\mathbf{x}_i, \mathbf{x}_{i'}, d_B) \le \mathtt{OT}(\mathbf{x}_i, \mathbf{x}_{i'}, d_{\widehat{B}_c^0}) \sum_{k=0}^{K_c} \beta_k \overset{(1)}{\le} \sum_{k=0}^{K_c} \mathtt{OT}(\mathbf{x}_i, \mathbf{x}_{i'}, d_{B^k}),
$$

where transition (1) is based on that $\mathtt{OT}(\mathbf{x}_i, \mathbf{x}_{i'}, d_{B^k})$ is at most $\beta_k \cdot \mathtt{OT}(\mathbf{x}_i, \mathbf{x}_{i'}, d_{\widehat{B}_c^0})$. The lower bound can be obtained in a similar way. Therefore, as $\mathtt{TW}(\mu_1, \mu_2, T) = \mathtt{OT}(\mu_1, \mu_2, d_T)$, the proposed TWD is bilipschitz equivalent to the hyperbolic tree-sliced Wasserstein distance

$$
\mathtt{TW}(\mathbf{x}_i, \mathbf{x}_{i'}, B) \simeq \sum_{k=0}^{K_c} \mathtt{TW}(\mathbf{x}_i, \mathbf{x}_{i'}, B^k).
$$

$\square$

Table 2: Statistics on real-world word-document and scRNA-seq benchmarks.

| Dataset | # Samples ($n$) | # Classes | # Features ($m$) |
|---------|-----------------|-----------|------------------|
| BBCSPORT | 517 documents | 5 | 13243 BOW |
| TWITTER | 2176 documents | 3 | 6344 BOW |
| CLASSIC | 4965 documents | 4 | 24277 BOW |
| AMAZON | 5600 documents | 4 | 42063 BOW |
| Zeisel | 3005 cells | 47 | 4000 genes |
| CBMC | 8617 cells | 56 | 500 genes |

# D    ADDITIONAL DETAILS ON EXPERIMENTAL STUDY

We describe the setups and additional details of our experiments in Sec. 5. The experiments are performed on NVIDIA DGX A100.

## D.1    IMPLEMENTATION DETAILS

**Setup.** In our experiments, the datasets are divided into a 70% training set and a 30% testing set. The split of the word-document datasets in Sec. 5.2 follows previous work (Kusner et al., 2015; Huang et al., 2016). This split was applied to both learning of the underlying feature tree and to the kNN model. Therefore, there is no train data leakage to the test data. The feature tree constructed from the training data can be reused to process new samples. For each evaluation, we employ a $k$NN classifier with varying $k \in \{1, 3, 5, 7, 9, 11, 13, 15, 17, 19\}$. The evaluation is repeated five times, and we report the best result, averaged over these five runs.

**Datasets.** In real-world word-document datasets, we test four word-document benchmarks in Kusner et al. (2015): (i) the BBCSPORT consisting of 13243 bags of words (BOW) and 517 articles categorized into five sports types, (ii) the TWITTER comprising 6344 BOW and 2176 tweets in three types of sentiment, (iii) the CLASSIC, including 24277 BOW and 4965 academic papers from four publishers, and (iv) the AMAZON containing 42063 BOW and 5600 reviews of four products. The Word2Vec embedding (Mikolov et al., 2013) is used as the word embedding vectors that are trained on the Google News[1] dataset. The types of documents are used as labels in classification tasks. In scRNA-seq data analysis, we test two scRNA-seq datasets in Dumitrascu et al. (2021), where the datasets are available at the link[2]. The first dataset, the Zeisel dataset, focuses on the mouse cortex and hippocampus (Zeisel et al., 2015). It comprises 4000 gene markers and 3005 single cells. The second dataset, the CBMC dataset, is derived from a cord blood mononuclear cell study (Stoeckius et al., 2017). It comprises 500 gene markers and 8617 single cells. We apply the divisive biclustering method from Zeisel et al. (2015) to obtain 47 subclasses for the Zeisel dataset and 56 subclasses for the CBMC dataset. The cell subclasses are used as labels in cell classification tasks. The Gene2Vec embeddings (Du et al., 2019) are used as the gene embedding vectors following the study of Gene Mover's Distance (Bellazzi et al., 2021).

We report the statistics of these benchmarks in Tab. 2.

**Baselines.**   We compare the proposed TWD with existing TWDs, including (i) Quadtree (Indyk & Thaper, 2003) that constructs a random tree by a recursive division of hypercubes, (ii) Flowtree (Backurs et al., 2020) that computes the optimal flow of Quadtree, (iii) tree-sliced Wasserstein distance (TSWD) (Le et al., 2019) that computes an average of TWDs by random sampling trees, where the numbers of sampling are set to 1, 5, and 10, (iv) UltraTree (Chen et al., 2024) that constructs an ultrametric tree (Johnson, 1967) by minimizing OT regression cost, (v) weighted cluster TWD (WCTWD) (Yamada et al., 2022), (vi) weighted Quadtree TWD (WQTWD) (Yamada et al., 2022), and their sliced variants (vii) SWCTWD and (viii) SWQTWD (Yamada et al., 2022). We also compare to TWDs where the trees are constructed by (i) MST (Prim, 1957), (ii) Tree Representation (TR)

---

[1] https://code.google.com/archive/p/word2vec/
[2] https://github.com/solevillar/scGeneFit-python/tree/62f88ef0765b3883f592031ca593ec79679a52b4/scGeneFit/data_files

(Sonthalia & Gilbert, 2020) through a divide-and-conquer approach, (iii) gradient-based hierarchical clustering (HC) in hyperbolic space (gHHC) (Monath et al., 2019), (iv) gradient-based Ultrametric Fitting (UltraFit) (Chierchia & Perret, 2019), and (v) HC by hyperbolic Dasgupta's cost (HHC) (Chami et al., 2020). We refer to App. B for an extended description of these baselines. For non-TWD baselines, we in addition include the Word Mover's Distance (WMD) (Kusner et al., 2015) for word-document data and we include the Gene Mover's Distance (GMD) (Bellazzi et al., 2021) for scRNA-seq data. In the experiments in Sec. 5, the Euclidean distance $\widetilde{d}$ is used as the ground metric for WMD, GMD, and the TWD baselines, which aligns with existing studies (Kusner et al., 2015; Bellazzi et al., 2021; Indyk & Thaper, 2003; Backurs et al., 2020; Le et al., 2019; Chen et al., 2024; Yamada et al., 2022). For the ablation study that uses cosine distance as the ground metric and directly performs classification tasks in the feature space using Euclidean distance and cosine distance, we refer to App. E.

**Kernel Type, Scale, and Initial Distance Metric.** When constructing the Gaussian kernel in Sec. 3, the initial distance is derived from the cosine similarity calculated in the ambient space, and the kernel scale is set to be $\{0.1, 1, 2, 5, 10\} \times \sigma$, where $\sigma$ is the median of the pairwise distances.

The diffusion operator's scale parameter $\epsilon$ controls information propagation across data points. Smaller $\epsilon$ values preserve local structures, beneficial for distinct clusters but may cause overfitting and noise sensitivity. Larger $\epsilon$ values capture broader relationships, useful for overlapping clusters but can blur distinctions. To balance these effects, we set $\epsilon$ within a standard range that is commonly-used in kernel and manifold learning.

The maximal scale $K_c$ determines the range of scales over which the hyperbolic manifold operates. We follow (Lin et al., 2023) and suggest setting $K_c \in \{0, 1, \ldots, 19\}$. Our empirical study, consistent with (Lin et al., 2023), shows that increasing the maximal scale does not significantly impact classification accuracy. This robustness simplifies the parameter selection process by indicating that the method performs reliably across a broad range of $K_c$ values.

As these parameters are task-specific and do not have a one-size-fits-all solution, we use Optuna (Akiba et al., 2019) to efficiently explore the parameter space and identify optimal settings, where $(\epsilon, K_c)$ are set $(10, 12)$, $(5, 7)$, $(1, 15)$, $(10, 14)$, $(2, 7)$, and $(5, 9)$ for BBCSPORT, TWITTER, CLASSIC, AMAZON, Zeisel, and CBMC, respectively. We note that the selection of kernel type, scale, and the initial distance metric in the kernel influences the effectiveness of our approach. Such choices are not only crucial but also task-specific, which is in line with the common practice in kernel and manifold learning methods. These elements, which do not have a one-size-fits-all solution, continue to be of wide interest in ongoing research (Lindenbaum et al., 2020a).

In our study, we employed the Gaussian kernel with a distance measure based on cosine similarity following prior research (Jaskowiak et al., 2014; Kenter & De Rijke, 2015), which is standard and commonly used in classification tasks and manifold learning. In our empirical analysis, we observed that employing the Euclidean distance in the ambient space tends to be highly sensitive and underperforms compared to using a distance based on the cosine similarity in the tested datasets, as detailed in the App. E. Our primary objective is to highlight the unique aspects of our method rather than kernel selection, which, while crucial, is a common consideration across all kernel and manifold learning methods.

### D.2 PROBABILISTIC HIERARCHICAL MODEL FOR SYNTHETIC DATA

To generate the synthetic dataset in Sec. 5.1, we employ a probabilistic model (see Fig. 2(a)) designed for generating 8-element vectors $\mathbf{x}_i \in \{0, 1\}^8$, where each element indicates the presence of one produce item. The generation of each vector is as follows. First, each of the two, fruits and vegetables, is selected with an independent $50\%$ probability. If fruits are selected, then each fruit (Apple, Orange, and Banana) is selected with a $50\%$ probability independently. If vegetables are selected, then each of the subcategories, green leaf, and root vegetable, is selected with a $50\%$ probability independently. If root vegetable is selected, then the vegetables Carrot and Beetroot are selected with a $50\%$ probability, independently. If the green leaf is selected, then the vegetables Kale, Spinach, and Lettuce are selected with a $50\%$ probability, independently. By generating a set of 100 such vectors $\{\mathbf{x}_i\}_{i=1}^{100}$, we create a data matrix with the hidden hierarchy of features (words).

### D.3 Accommodating Large Datasets

In our work, we note that constructing the diffusion kernels and their eigenvectors by applying the eigendecomposition to them is the most computationally intensive step. However, recent methods in diffusion geometry have proposed various techniques to significantly reduce the run time and space complexity of diffusion operators. Here, we consider the diffusion landmark (Shen & Wu, 2022) for scalability improvements which reduces the complexity to $O(n^{1+2\tau})$ from the much larger $O(n^3)$, where $\tau < 1$ is the proportion of the landmark set. We briefly describe the diffusion landmark approach below.

Consider a set of points $\mathcal{A} = \{\mathbf{a}_j\}_{j=1}^m \subseteq \mathbb{R}^n$, let $\mathcal{A}' = \{\mathbf{a}'_j\}_{j=1}^{m'} \subseteq \mathbb{R}^n$ be a subset of $\mathcal{A}$, where $1 > \tau = \log_m(m')$. A landmark-set affinity matrix is constructed by $\mathbf{K}(j, j') = \exp(-d^2(j, j')/\epsilon)$, where $d(j, j')$ denotes a suitable distance between the data points $\mathbf{a}_j \in \mathcal{A}$ and $\mathbf{a}'_j \in \mathcal{A}'$. Let $\widetilde{\mathbf{D}}$ be a diagonal matrix, where the diagonal elements are $\widetilde{\mathbf{D}}(j, j) = \mathbf{e}_j^\top \mathbf{K}\mathbf{K}^\top \mathbf{1}_m$. The eigenstructure of an SPD matrix $\mathbf{D}^{-1/2}\widetilde{\mathbf{Q}}\mathbf{D}^{-1/2}$ that is similar to the diffusion operator (Coifman & Hirn, 2014; Katz et al., 2020) is recovered by applying SVD to the matrix $\widetilde{\mathbf{D}}^{-1/2}\mathbf{K} = \widetilde{\mathbf{U}}\widetilde{\mathbf{\Lambda}}\widetilde{\mathbf{V}}$, where $\widetilde{\mathbf{Q}} = \mathbf{K}\mathbf{K}^\top \in \mathbb{R}^{m \times m}$ is the landmark-affinity matrix. The technique of constructing the diffusion on $\mathcal{A}$ by going through the subset $\mathcal{A}'$ and their eigenvectors can be seamlessly integrated into our TWD, enabling the analysis of datasets larger than ten thousand data points, e.g., the word-document data in Sec. 5.2.

## E Additional Experimental Results

### E.1 Toy Problem

Consider a four-level balanced binary tree $T = (V, E, \mathbf{A})$ rooted at node $v_1$, where $V = \{v_1, \ldots, v_{15}\}$ is the vertex set organized from the root to the leaves of the tree, $E$ is the edge set, and $\mathbf{A} \in \mathbb{R}^{15 \times 15}$ is the non-negative edge weights. Consider a set of vectors $\{\mathbf{x}_i \in \mathbb{R}^{15}\}_{i=1}^n$ such that $\mathbf{x}_i$ is a non-negative realization from a normal distribution $\mathcal{N}(\boldsymbol{\mu}, \mathbf{L}^\dagger)$ (Rue & Held, 2005), where $\boldsymbol{\mu} = 5 \cdot \mathbf{1}_{15} \in \mathbb{R}^{15}$ and $\mathbf{L}$ is the graph Laplacian of $T$[3].

The set of vectors $\{\mathbf{x}_i\}_{i=1}^n$ were generated five times, and each time with a different number of samples $n \in \{3, 10, 32, 10^2, 316, 10^3, 3162, 10^4\}$. Fig. 3 illustrates the normalized Frobenius norm of the discrepancy between the pairwise TWD matrix obtained by Alg. 2 and the pairwise TWD matrix $\mathtt{TW}(\cdot, \cdot, T)$. We see that the proposed TWD provides an accurate approximation of the TWD with the tree $T$, even based on a limited sample size. In addition, as the number of samples increases, the proposed TWD approaches closer to the true TWD, highlighting its efficacy in capturing the hidden feature hierarchy.

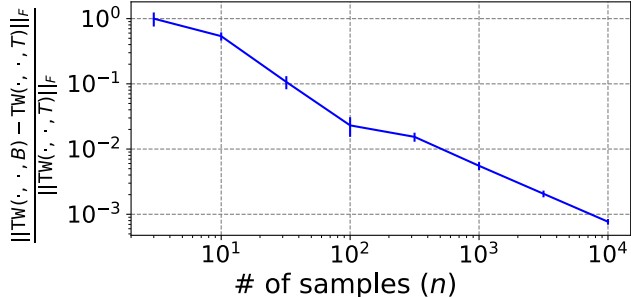

Figure 3: The normalized Frobenius norm of the difference between the proposed TWD and ground truth TWD with different number of samples $n$.

---

[3]The mean is large compared to the covariance matrix, and the realizations are non-negative with high probability. Realizations with negative elements are rejected.

### E.2 RUNTIME ANALYSIS

In Fig. 4, we present the runtime performance of our TWD in comparison to other TWDs and Wasserstein distances when applied to scRNA-seq and word-document datasets. Although our TWD is not the fastest method, it consistently ranks as the fourth or fifth quickest. It outperforms deep-based methods like HHC-TWD and gHHC-TWD, as well as pre-trained OT baselines such as GMD for single-cell RNA sequencing datasets and WMD for word-document datasets. The fastest methods identified are TSWD-1, TR-TWD, and MST-TWD. The work in (Le et al., 2019) reports that the runtime of TSWD is influenced by the number of tree slices used in sampling. Similarly, slice variants SWCTWD and SWQTWD are also affected by the number of slices. Despite being slightly slower compared to TSWD-1, TR-TWD, and MST-TWD, our TWD demonstrates significant advantages in terms of classification accuracy, as shown in Tab. 1. We also report the runtime of the OT distance using the embedding distance between features as a ground distance, denoted as OT-ED. We note that this computational expense is $O(mn^3 + m^3 \log m)$. In contrast, our TWD offers a substantially reduced runtime of $O(m^{1.2})$, thereby demonstrating a significant advantage when handling large datasets. Note that the scalability of the proposed TWD is enhanced by utilizing diffusion landmarks (Shen & Wu, 2022), as detailed in App. D.3. This enhancement allows our TWD to achieve computational performance comparable to TSWD-1, TR-TWD, and MST-TWD, particularly in datasets with a large number of features, such as the word-document datasets shown in Fig. 4.

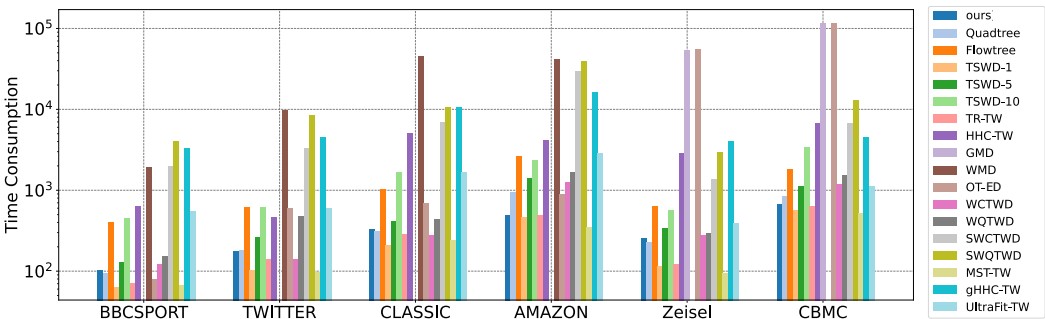

Figure 4: Run time of the proposed TWD and competing TWD and OT distances on scRNA-seq datasets and word-document datasets.

### E.3 ABLATION STUDIES

We note that our method consists of several non-trivial steps. We provide theoretical justification for our method in Sec. 4. Specifically, in Thm. 2, we show that the TWD $\mathtt{TW}(\cdot, \cdot, T)$ based on the hidden feature tree $T$ can be approximated by our TWD using the decoded tree $B$, where the proposed TWD and $B$ are obtained solely from the data matrix, without access to ground truth $T$. In addition, in Thm. 1, we show that the tree metric of the decoded tree $B$ is bilipschitz equivalent to the tree metric of the hidden tree $T$. These results stem directly from the particular steps of the proposed method and might not exist otherwise. In the following, we performed a comprehensive empirical ablation study, demonstrating the importance of each component in our method.

#### E.3.1 EUCLIDEAN INITIAL METRIC

Fig. 5 presents the classification performance on the scRNA-seq and word-document datasets using the Euclidean distance as an input to Alg. 2 instead of using the distance based on cosine similarity. The Euclidean distance leads to inferior classification compared to the cosine similarity. Indeed, as remarked in App. D, the selection of the distance metric influences the performance of our method. In future work, we plan to comprehensively investigate the use of different metrics.

#### E.3.2 TWD BY MST AND AT

Fig. 6 shows the classification performance of the scRNA-seq and word-document datasets using the TWD with the tree constructed using Minimum Spanning Trees (MST) (Prim, 1957) and Approximat-

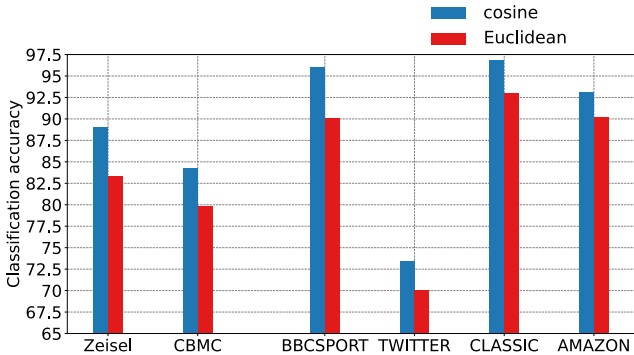

Figure 5: The classification accuracy of the proposed TWD using the distance based on cosine similarity and Euclidean distance for scRNA-seq and word-document datasets.

ing Tree (AT) (Chepoi et al., 2008). To ensure that the tree nodes are contained in a subtree, which is required when computing the TWD in Eq. (1), we set the minimal leaf number as the number of features in MST for computing the TWD between samples. We see that the TWD approximated by either the MST or the AT is less effective than our TWD, implying that MST and AT do not capture well the underlying tree structure between features compared to our binary tree construction.

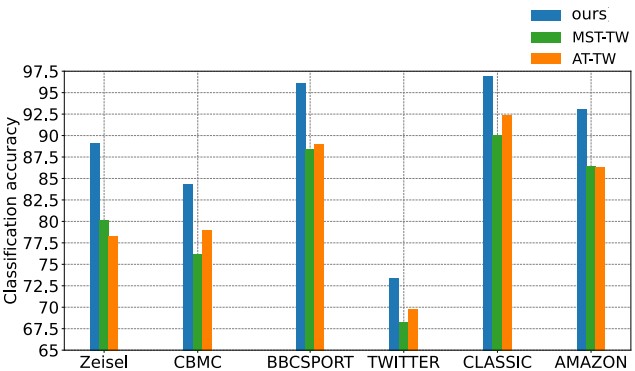

Figure 6: The classification accuracy of our TWD and TWDs computed with MST and AT for scRNA-seq and word-document datasets.

### E.3.3 TREE CONSTRUCTION BY EMBEDDING DISTANCE

We note that the embedding distance (Lin et al., 2023) could be simply used for tree construction in existing TWDs (Indyk & Thaper, 2003; Le et al., 2019; Yamada et al., 2022), graph-based algorithms (Alon et al., 1995), or linkage methods (Murtagh & Contreras, 2012) instead of the HD-LCA in the tree decoding (Alg. 1). However, these generic algorithms, while providing upper bounds on the distortion of the constructed tree distances, do not build a geometrically meaningful tree when given a tree metric (Sonthalia & Gilbert, 2020), because they do not exploit the hyperbolic geometry of the embedded space or do not have theoretical guarantees to recover the hidden feature hierarchy.

Tab. 3 shows the classification accuracy on the word-document and scRNA-seq datasets obtained by the proposed TWD and TWDs using embedding distance as input to construct the feature tree. We see that our TWD leads to the best empirical performance among all the tested methods. This result indicates that using HD-LCA, where the information is encoded in the multi-scale hyperbolic LCAs, is important for revealing the hidden feature hierarchy, and therefore, our TWD is more effective and meaningful.

Table 3: Document and single-cell classification accuracy of the proposed TWD and TWDs using embedding distance (ED).

| | Real-World Word-Document | | | | Real-World scRNA-seq | |
|---|---|---|---|---|---|---|
| | BBCSPORT | TWITTER | CLASSIC | AMAZON | Zeisel | CBMC |
| Quadtree-ED | $94.7_{\pm0.3}$ | $70.3_{\pm1.0}$ | $94.7_{\pm0.9}$ | $88.2_{\pm0.7}$ | $81.4_{\pm1.7}$ | $80.9_{\pm1.3}$ |
| Flowtree-ED | $95.0_{\pm1.6}$ | $71.2_{\pm1.2}$ | $93.2_{\pm0.9}$ | $90.6_{\pm1.1}$ | $81.0_{\pm1.2}$ | $81.0_{\pm1.3}$ |
| WCTWD-ED | $93.1_{\pm2.7}$ | $68.7_{\pm2.4}$ | $92.4_{\pm3.1}$ | $87.9_{\pm1.7}$ | $81.9_{\pm5.2}$ | $78.9_{\pm2.7}$ |
| WQTWD-ED | $94.0_{\pm2.1}$ | $67.5_{\pm2.9}$ | $93.5_{\pm3.9}$ | $86.5_{\pm2.1}$ | $80.0_{\pm3.7}$ | $79.7_{\pm3.6}$ |
| UltraTree-ED | $94.0_{\pm2.2}$ | $69.3_{\pm3.0}$ | $91.4_{\pm2.5}$ | $87.9_{\pm3.4}$ | $83.1_{\pm2.3}$ | $80.6_{\pm2.2}$ |
| TSWD-1-ED | $86.4_{\pm2.1}$ | $67.3_{\pm1.8}$ | $92.9_{\pm1.4}$ | $86.2_{\pm1.3}$ | $79.4_{\pm1.2}$ | $74.8_{\pm2.2}$ |
| TSWD-5-ED | $87.3_{\pm1.9}$ | $69.2_{\pm1.4}$ | $93.7_{\pm1.4}$ | $89.8_{\pm0.7}$ | $80.8_{\pm1.2}$ | $75.4_{\pm1.9}$ |
| TSWD-10-ED | $88.0_{\pm1.3}$ | $70.0_{\pm1.1}$ | $94.2_{\pm0.9}$ | $90.7_{\pm0.3}$ | $81.9_{\pm1.3}$ | $76.0_{\pm1.3}$ |
| SWCTWD-ED | $91.9_{\pm1.5}$ | $70.7_{\pm1.0}$ | $94.0_{\pm1.3}$ | $91.0_{\pm1.5}$ | $81.4_{\pm2.5}$ | $76.2_{\pm1.5}$ |
| SWQTWD-ED | $93.8_{\pm1.4}$ | $71.2_{\pm1.4}$ | $94.2_{\pm2.6}$ | $90.3_{\pm1.5}$ | $82.6_{\pm2.7}$ | $80.4_{\pm2.9}$ |
| MST-TWD-ED | $89.2_{\pm4.4}$ | $69.1_{\pm1.4}$ | $89.2_{\pm2.7}$ | $87.8_{\pm2.1}$ | $81.3_{\pm2.2}$ | $77.4_{\pm2.7}$ |
| TR-TWD-ED | $90.6_{\pm1.3}$ | $71.8_{\pm1.3}$ | $94.5_{\pm1.5}$ | $90.6_{\pm1.5}$ | $84.2_{\pm1.9}$ | $80.5_{\pm1.0}$ |
| HHC-TWD-ED | $84.1_{\pm1.3}$ | $71.2_{\pm2.2}$ | $91.8_{\pm3.3}$ | $89.6_{\pm1.5}$ | $80.7_{\pm1.2}$ | $79.5_{\pm0.9}$ |
| gHHC-TWD-ED | $82.5_{\pm2.3}$ | $70.8_{\pm1.4}$ | $91.0_{\pm1.8}$ | $87.4_{\pm3.1}$ | $80.1_{\pm0.8}$ | $74.6_{\pm1.7}$ |
| UltraFit-TWD-ED | $84.5_{\pm1.2}$ | $70.6_{\pm1.5}$ | $93.0_{\pm1.6}$ | $89.3_{\pm2.7}$ | $80.6_{\pm2.9}$ | $78.1_{\pm1.3}$ |
| Ours | $96.1_{\pm0.4}$ | $73.4_{\pm0.2}$ | $96.9_{\pm0.2}$ | $93.1_{\pm0.4}$ | $89.1_{\pm0.4}$ | $84.3_{\pm0.3}$ |

### E.3.4 Exploring Other Hyperbolic Embedding and the Impact of Curvature

Hyperbolic embedding methods (Chamberlain et al., 2017; Nickel & Kiela, 2017; 2018) typically assume a fully known or partially known graph of $T$, which in turn is embedded into hyperbolic space. In Sonthalia & Gilbert (2020); Lin et al. (2023); Sala et al. (2018); Weber et al. (2024), these methods were shown to inadequately capture the intrinsic hierarchical relation underlying high-dimensional observational data $\mathbf{X}$ (i.e., the node attributes). Therefore, we opted to use hyperbolic embedding (Lin et al., 2023), a recent hyperbolic embedding method with theoretical guarantees. This embedding and its accompanying theory enabled us to introduce our theoretical results in Sec. 4.

It is important to note that we could indeed apply traditional hyperbolic embedding methods directly to the affinity matrix $\mathbf{Q}$. From a graph perspective, this is the affinity of a graph, but this graph is not usually a tree. These traditional methods, due to the quantity they are minimizing, will estimate a hyperbolic distance that approximates as best as possible the shortest path distance on this non-tree graph induced by $\mathbf{Q}$. As such, they do not, in general, recover a hyperbolic distance related to the hyperbolic distance of the hidden tree $T$. In contrast, the hyperbolic diffusion embedding approach (Lin et al., 2023) is designed such that the recovered hyperbolic metric is close to that of the hidden tree $T$, guaranteeing it will be bilipschitz equivalent to it. One of the reasons for the success of this method is that the obtained embedding is not provided by the minimization of an objective function controlled by $\mathbf{Q}$. Instead, Lin (Lin et al., 2023) designed a non-trivial process that is neither iterative nor based on optimization that computes the embedding directly with theoretical guarantees. We build on the theoretical guarantees of this embedding in our own theoretical contribution in Thm. 1.

We here give an intuition as to the reason for the success of the approach from Lin et al. (2023). After normalizing the affinity matrix $\mathbf{Q}$, the diffusion operator $\mathbf{P}$ is computed. This operator was proven to reveal the underlying manifold. This property is what allows an embedding approach based on the diffusion operator to approximately recover the underlying tree metric $d_T$ rather than fit the metric of a non-tree graph constructed from the affinity matrix $\mathbf{Q}$. The diffusion operator reveals the underlying manifold in the following sense (Coifman & Lafon, 2006). In the limit of the infinite number of features $m \to \infty$ and small scale $\epsilon \to 0$, the operator $\mathbf{P}^{t/\epsilon}$ pointwise converges to the Neumann heat kernel of the underlying manifold $\mathbf{H}_t = \exp(-t\Delta)$, where $\Delta$ is the Laplace–Beltrami operator on the manifold. In other words, the diffusion operator is a discrete approximation of the heat kernel on the manifold. Intuitively, $\mathbf{P}^t$ is designed to reveal the local connectivity at diffusion time scale $t$.

Considering multiple timescales on a dyadic grid $\{2^{-k}\}_{k=0}^{K_c}$ allows for the association of different diffusion timescales with neighborhoods of varying sizes. The multi-scale embedding of the associated diffusion operators in hyperbolic space generates distances that are exponentially scaled, aligning with the structure of a tree distance. This scaling naturally introduces a hierarchical relationship between the diffusion timescales, enabling the estimation of a latent hierarchical metric. From a theoretical standpoint, it was shown that the $\ell_1$ distance in this multi-scale embedding induced by the metric of the product of hyperbolic spaces can estimate the distance $d_T$ in the hidden hierarchical space that underlies the observations even when $d_T$ is not given or when we do not have access to the explicit $T$ (Lin et al., 2023). It is stated under the assumption that $(\mathbf{P}^t)_{t \in (0,1]}$ is a pointwise approximation of the heat kernel. Such an approximation, mainly in the limit $m \to \infty$ and $\epsilon \to 0$, was shown and studied in Coifman & Lafon (2006); Singer (2006); Belkin & Niyogi (2008). In addition, three strong regularity conditions are required: an upper bound on the operator elements, a lower bound on the operator elements, and a Hölder continuity condition. Importantly, the heat kernel satisfies these conditions (Lin et al., 2023).

In our work, we use the hyperbolic diffusion embedding from Lin et al. (2023) to present a novel data-driven tree decoding method that operates in high-dimensional hyperbolic spaces. Our method is based on a new definition of the counterpart of the LCA in hyperbolic spaces. Our Thm. 1 is built on the multi-scale hyperbolic embedding from Lin et al. (2023), where we use the theoretical property (Lin et al., 2023) that the $\ell_1$ distance in this embedding can estimate the hidden tree metric $d_T$. The difference between Thm. 1 and the work from Lin et al. (2023) is that we construct an explicit tree $B$ whose tree metric approximates the hidden tree metric $d_T$. In Lin et al. (2023), no tree is recovered, and only the hidden tree metric is approximated by the hyperbolic embedding distances. Our Thm. 2 is established based on Thm. 1, where the proposed TWD can estimate the TWD associated with the latent feature tree $T$ without access to $T$. The uniqueness of our approach lies in its ability to work without the need for a pre-known hierarchical graph.

Among the existing hyperbolic models, we chose to work with the Poincaré half-space model because (i) it is natural to represent diffusion times on a dyadic grid in this model, following previous work (Lin et al., 2023), and (ii) there exists a closed-form solution of the hyperbolic LCA and the HD-LCA, which are the key components in our tree decoding algorithm in Alg. 1. Since the four standard models of hyperbolic space (Bowditch, 2007): the Poincaré disk, Lorentz, Poincaré half-space, and Beltrami-Klein, are equivalent and isometric, it is possible to use the closed-form solution of LCA in the alternative models, e.g., the Lorentz model for numerical stability. However, we did not encounter any numerical instabilities in our empirical studies using our implementation in the Poincaré half-space model.

We consider hyperbolic space with negative constant curvature of $-1$ because it aligns with the common practice in hyperbolic representation (Nickel & Kiela, 2017; 2018; Sala et al., 2018; Chami et al., 2020). Any negative constant curvature can indeed be used and considered as a hyperparameter, e.g., (Chami et al., 2019). The modifications that it will require from our algorithm are minimal, and most of the steps can be applied seamlessly in such a space. We conducted experiments on scRNA-seq data to investigate the role of curvature in our method. Specifically, we vary the curvature and explore the effects of constant negative curvature $-1/\kappa$ where $\kappa > 0$. Fig. 7 presents the classification accuracy of the Zeisel and CBMC datasets as a function of the curvature. We see that decreasing the curvature slightly improves the cell classification accuracy. This suggests that while the choice of curvature does have a large impact, our method remains robust across a range of curvatures. In addition, fine-tuning the curvature can enhance performance, and starting with the default choice of $-1/\kappa = -1$ is a reasonable and commonly accepted practice.

We note that other TWD baselines could use other two-dimensional Poincaré representations (e.g., Poincaré embedding (PE) (Nickel & Kiela, 2017) or hyperbolic VAE (Nagano et al., 2019)) and decode the hyperbolic embedding into a tree by HHC (Chami et al., 2020). We denoted these baselines by PE-HHC-TWD and HVAE-HHC-TWD, respectively, and present their results in Tab. 4. We see in the table that HHC (Chami et al., 2020) applied to a PE (Nickel & Kiela, 2017) or hyperbolic VAE (Nagano et al., 2019) is less effective than our TWD by a large margin. This discrepancy is attributed to the sensitivity of the recovery of the hidden feature hierarchy using the competing methods, a finding that is consistent with previous works (Sala et al., 2018; Sonthalia & Gilbert, 2020; Lin et al., 2023).

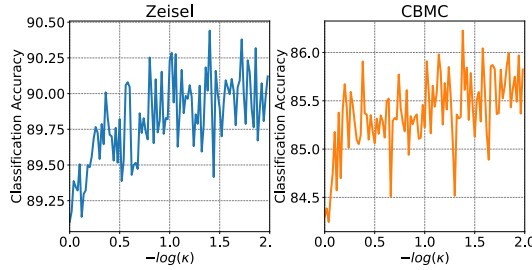

Figure 7: The classification accuracy for the Zeisel and CBMC datasets under different curvatures $-1/\kappa$. Decreasing curvature could slightly improve the cell classification accuracy.

Table 4: Document and single-cell classification accuracy of our TWD and TWDs constructed by other hyperbolic embeddings.

|  | Real-World Word-Document | | | | Real-World scRNA-seq | |
|---|---|---|---|---|---|---|
|  | BBCSPORT | TWITTER | CLASSIC | AMAZON | Zeisel | CBMC |
| PE-HHC-TWD | $87.2_{\pm1.2}$ | $69.7_{\pm0.9}$ | $93.2_{\pm1.1}$ | $86.2_{\pm0.9}$ | $81.6_{\pm0.9}$ | $74.9_{\pm1.0}$ |
| HVAE-HHC-TWD | $88.4_{\pm2.5}$ | $64.3_{\pm1.7}$ | $90.8_{\pm1.5}$ | $87.3_{\pm1.7}$ | $80.2_{\pm0.9}$ | $75.1_{\pm1.0}$ |
| Ours | $96.1_{\pm0.4}$ | $73.4_{\pm0.2}$ | $96.9_{\pm0.2}$ | $93.1_{\pm0.4}$ | $89.1_{\pm0.4}$ | $84.3_{\pm0.3}$ |

### E.3.5 ROBUSTNESS ANALYSIS UNDER GAUSSIAN NOISE

To demonstrate the robustness of our method, we conducted experiments on scRNA-seq data to evaluate the resilience of the proposed TWD to the presence of noise. Specifically, we added Gaussian noise to the measurements and examined the effect of varying the noise variance on classification performance.

Fig. 8 presents the cell classification accuracy on Zeisel and CBMC datasets under Gaussian noise perturbations. We see that while adding noise to the measurements hinders the extraction of hidden feature hierarchy and is directly reflected in degraded cell classification performance, our method remains more robust than the competing methods by a large margin. This demonstrates that our approach can provide reliable recovery guarantees even when the ground truth hierarchical features are obscured by noise.

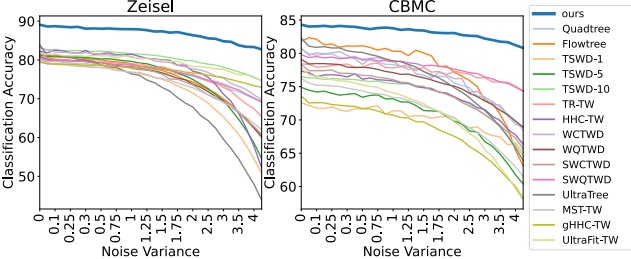

Figure 8: Assessing the robustness of cell classification under Gaussian noise perturbations. While adding noise to the measurements hinders the extraction of hidden feature hierarchy and is directly reflected in the cell classification performance, our method remains more robust than the competing baselines by a large margin.

### E.3.6 COMPARISON TO NON-TWD BASELINES

For non-TWD baselines, we also compare our method to Word Mover's Distance (WMD) (Kusner et al., 2015) for word-document datasets and Gene Mover's Distance (GMD) (Bellazzi et al., 2021) for scRNA-seq datasets in Sec. 5. Note that WMD (Kusner et al., 2015) computes the Wasserstein

Table 5: Document and single-cell classification accuracy of WMD and GMD using cosine distance as the ground metric.

| | Real-World Word-Document | | | | Real-World scRNA-seq | |
| | BBCSPORT | TWITTER | CLASSIC | AMAZON | Zeisel | CBMC |
| --- | --- | --- | --- | --- | --- | --- |
| WMD | $91.2_{\pm0.4}$ | $68.4_{\pm0.2}$ | $93.5_{\pm0.3}$ | $89.4_{\pm0.2}$ | - | - |
| GMD | - | - | - | - | $86.9_{\pm0.5}$ | $82.7_{\pm0.9}$ |
| Ours | $96.1_{\pm0.4}$ | $73.4_{\pm0.2}$ | $96.9_{\pm0.2}$ | $93.1_{\pm0.4}$ | $89.1_{\pm0.4}$ | $84.3_{\pm0.3}$ |

Table 6: Document and cell classification accuracy in the feature space using the Euclidean distance and cosine distance.

| | Real-World Word-Document | | | | Real-World scRNA-seq | |
| | BBCSPORT | TWITTER | CLASSIC | AMAZON | Zeisel | CBMC |
| --- | --- | --- | --- | --- | --- | --- |
| Euclidean | $78.2_{\pm1.5}$ | $57.3_{\pm0.6}$ | $64.0_{\pm0.5}$ | $71.3_{\pm0.4}$ | $67.2_{\pm1.3}$ | $62.5_{\pm0.9}$ |
| Cosine | $91.5_{\pm0.9}$ | $68.1_{\pm0.6}$ | $90.3_{\pm0.8}$ | $86.4_{\pm0.7}$ | $73.5_{\pm1.0}$ | $72.7_{\pm1.1}$ |
| Ours | $96.1_{\pm0.4}$ | $73.4_{\pm0.2}$ | $96.9_{\pm0.2}$ | $93.1_{\pm0.4}$ | $89.1_{\pm0.4}$ | $84.3_{\pm0.3}$ |

distance using the Euclidean distance $\widetilde{d}$ between precomputed Word2Vec embeddings (Mikolov et al., 2013) as the ground cost, while GMD (Bellazzi et al., 2021) is defined similarly, using the Euclidean distance $\widetilde{d}$ between precomputed Gene2Vec embeddings as the ground cost. In the experiments in Sec. 5, the Euclidean distance $\widetilde{d}$ is used as the ground metric for WMD, GMD, and the TWD baselines, which aligns with existing studies (Kusner et al., 2015; Huang et al., 2016; Le et al., 2019; Takezawa et al., 2021; Yamada et al., 2022; Chen et al., 2024; Bellazzi et al., 2021).

We conduct additional experiments evaluating WMD on word-document datasets and GMD on scRNA-seq datasets, both using cosine distance as the ground metric. Tab. 5 presents the document and cell classification accuracy of WMD and GMD using cosine distance as the ground metric. The results show that WMD performs worse with cosine distance as the ground metric compared to Euclidean distance, while GMD performs better with cosine distance than Euclidean distance. However, regardless of the ground metric (cosine or Euclidean), WMD and GMD exhibit less effective performance compared to our method. These findings emphasize the value of inferring latent feature hierarchies, as our approach outperforms the baselines in these tasks.

It is important to note that using the ground metric $\widetilde{d}$ (e.g., Euclidean) in WMD, GMD, or existing TWD methods results in the (approximated) Wasserstein distance being based on the predefined metric $\widetilde{d}$. In contrast, our method employs an initial distance $d$, computed using the cosine similarity in the experiments, to compute an affinity matrix within the diffusion operator, which is central to kernel and manifold learning methods. This process leads to a Wasserstein distance based on a latent hierarchical metric induced by $d$. As noted in App. D.1, the kernel type, scale parameter, and initial distance metric selection in our approach significantly influence its effectiveness. These choices are task-specific and are critical in achieving optimal performance (Lindenbaum et al., 2020a).

In addition, we conduct additional tests evaluating classification performance directly in the feature space, without using Wasserstein distance or TWD. Tab. 6 presents the document and cell classification accuracy in the feature space using the Euclidean distance and cosine distance. We see that the classification performance relying on generic metrics (Euclidean or cosine) is much less effective than WMD and GMD, which aligns with the findings reported in WMD (Kusner et al., 2015) and GMD (Bellazzi et al., 2021). This highlights that incorporating the feature relationship into a distance between samples improves the effectiveness of distance metric learning. Furthermore, we see that our TWD method performs better than the distance relying on generic metrics, WMD and GMD. This suggests that inferring explicitly the feature hierarchies is important and leads to the effectiveness of distance metric learning for data with a hierarchical structure.

### E.4 EVALUATING OUR TWD ON NON-HIERARCHICAL DATA

We focus on the problem of data with a hidden feature hierarchy as it is of broad interest and prevalent across a wide range of applications. Therefore, the existence of a hidden feature hierarchy is an important assumption in our method.

We tested our TWD on MNIST datasets, where the images are column-stacked. We compared the proposed TWD with the same competing methods as in the paper. The digit classification was performed in the same way as in Sec. 5. Fig. 9 presents the resulting digit classification accuracy. As expected, and in contrast to the word-document and scRNA-seq data, the classification results do not match the state of the art since there are no definitive latent hierarchical structures in this case. Yet, our method still demonstrates good results and outperforms all the other TWD-based methods.

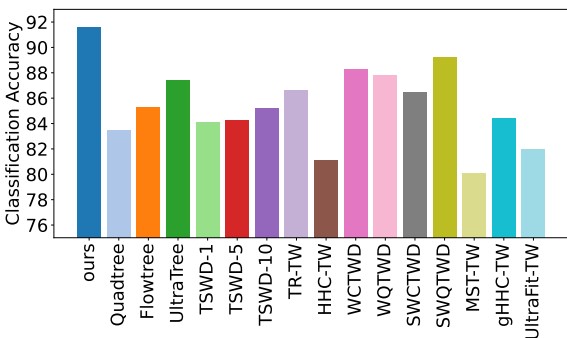

Figure 9: Digit classification accuracy on MNIST dataset. In contrast to the word-document and scRNA-seq data, in this case, there are no definitive latent hierarchical structures. Yet, our method still demonstrates good results and outperforms the competing TWD methods.

We remark that the state-of-the-art methods of MNIST do not tackle the problem of finding meaningful distances between data samples that incorporate the latent hierarchical structure of the features. Moreover, they usually train on labeled data, whereas the distance we compute is unsupervised. Therefore, comparing the results obtained by our method (and the results obtained by the other TWD-based methods) to the state of the art is not entirely fair.

## F ADDITIONAL REMARKS

### F.1 SINGLE-SCALE HYPERBOLIC DIFFUSION BINARY TREE DECODING

In Alg. 1, the tree decoding is based on the multi-scale embeddings and the corresponding HD-LCAs. In Alg. 3 we present a similar algorithm based on the single scale hyperbolic embedding $\mathbf{z}_j^k$ and the corresponding $k$-th hyperbolic LCAs $\mathbf{z}_j^k \vee \mathbf{z}_{j'}^k$ in Def. 1, giving rise to a (single-scale) binary tree $B^k$. Specifically, we merge the pair of features exhibiting the highest similarity at the $k$-th hyperbolic LCA, determined by the smallest value of $\mathtt{proj}(\mathbf{z}_j^k \vee \mathbf{z}_{j'}^k)$ in Prop. 1. Following Def. 2, only one of the hyperbolic LCA relations $[j_2 \vee j_3 \sim j_1]_{\mathbb{H}^{m+1}}^k$, $[j_1 \vee j_3 \sim j_2]_{\mathbb{H}^{m+1}}^k$, or $[j_1 \vee j_2 \sim j_3]_{\mathbb{H}^{m+1}}^k$, holds in $B^k$. The merged nodes are then connected by edges whose weights are assigned by $d_{\mathbb{H}^{m+1}}(\cdot, \cdot)$ between the corresponding points. It is worth noting that considering $(K_c + 1)$ trees $\{B^k\}_{k=0}^{K_c}$ generated by Alg. 3 gives rise to a tree-sliced Wasserstein distance as defined in Prop. 5. In Prop. 5, we show that our TWD efficiently captures the tree-sliced information encoded within these multi-scale trees, circumventing the need for their explicit construction. In addition, we posit that our high-dimensional tree decoding in Alg. 3 for single-scale hyperbolic embedding can be readily applied to other hyperbolic embeddings, e.g., (Nickel & Kiela, 2017; 2018; Nagano et al., 2019), extending its applicability and utility. While this extension is technically feasible, it does not carry the theoretical guarantees as those established in Thm. 1 and Thm. 2.

---

**Algorithm 3** Single-Scale Hyperbolic Binary Tree Decoding

---

**Input:** A single scale Poincaré half-space representation at the $k$-th level $\{\{\mathbf{z}_1^k\}, \{\mathbf{z}_2^k\}, \ldots, \{\mathbf{z}_m^k\}\}$

**Output:** The $k$-th rooted binary tree $B^k = (\widehat{V}_c^k, \widehat{E}_c^k, \widehat{\mathbf{A}}_c^k)$ with $m$ leaf nodes

**function** H_BinaryTree($\{\{\mathbf{z}_1^k\}, \{\mathbf{z}_2^k\}, \ldots, \{\mathbf{z}_m^k\}\}$)
    $B^k \leftarrow$ leaves($\{j\} : j \in \{1, 2, \ldots, m\}$)
    **for** $j, j' \in \{1, 2, \ldots, m\}$ **do**
        $\mathbf{z}_j^k \vee \mathbf{z}_{j'}^k \leftarrow \underset{\mathbf{z} \in \mathbb{H}^{m+1}}{\arg\min} \sum_{l=j,j'} d_{\mathbb{H}^{m+1}}^2(\mathbf{z}, \mathbf{z}_l^k)$
    **end for**
    $\mathcal{S} = \{(j, j') |$ pairs sorted by $\mathbf{z}_j^k \vee \mathbf{z}_{j'}^k (m+1)\}$
    **for** $(j, j') \in \mathcal{S}$ **do**
      **if** nodes $j$ and $j'$ are not in the same subtree in $B^k$
        $\mathcal{I}_j \leftarrow$ internal node consisting node $j$
        $\mathcal{I}_{j'} \leftarrow$ internal node consisting node $j'$
        add a new internal node consisting $\mathcal{I}_j$ and $\mathcal{I}_{j'}$
        assign the edge weight by their geodesic distance
      **end if**
    **end for**
    **return** $B^k$

---

### F.2 PUTTING THE FEATURES ON THE LEAVES OF THE TREE

The ground truth latent tree $T$ does not assume that features are positioned at the leaves. In addition, note that the hyperbolic embedding (Lin et al., 2023), which is consistent with other common hyperbolic embedding methods (Chamberlain et al., 2017; Nickel & Kiela, 2017; 2018; Sala et al., 2018), does not restrict all the features to the leaves. Assigning the features to the leaf nodes is only carried out in the tree decoding (Alg. 1), which is in line with other tree decoding algorithms (Sarkar, 2011; Chami et al., 2020; Indyk & Thaper, 2003). Note that our tree decoding algorithm generates a weighted binary tree. We assert that this does not pose a restriction because any tree can be effectively transformed into a binary tree (Bowditch, 2007; Cormen et al., 2022). For instance, consider the flexible tree in Fig. 10 (a): [sport[soccer,football,basketball]] (where "sports" is a parent node with three children: "basketball", "soccer", and "football"). This tree can be accurately modeled as a (weighted) binary tree as in Fig. 10 (b): [c[b[a[soccer, football], basketball]], sport] (where "sports", "basketball", "soccer", and "football" are all leaves, $a, b, c$ are internal nodes, and the edges encode the same relationships as the flexible tree above).

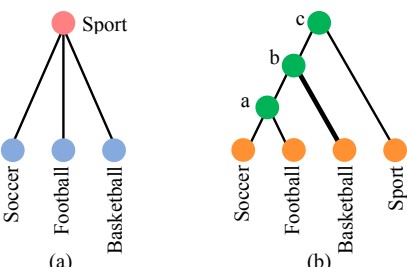

(a)              (b)

Figure 10: (a) In a flexible tree, "sports" is a parent node with three children: "basketball", "soccer", and "football". (b) In a binary tree, "sports", "basketball", "soccer", and "football" are all leaves, $a, b, c$ are internal nodes.

### F.3 DIFFERENTIABILITY OF OUR TWD

The proposed TWD can be expressed in a matrix form (Takezawa et al., 2021) as

$$
\mathrm{TW}(\mathbf{x}_i, \mathbf{x}_{i'}, B) = \sum_{v \in \widetilde{V}} \alpha_v \left| \sum_{u \in \Gamma_B(v)} (\mathbf{x}_i(u) - \mathbf{x}_{i'}(u)) \right| = \|\mathrm{diag}(\boldsymbol{\alpha}) \mathbf{R}_B (\mathbf{x}_i - \mathbf{x}_{i'})\|_1, \qquad (12)
$$

where $\boldsymbol{\alpha} \in \mathbb{R}^M$ is a weights vector with $\boldsymbol{\alpha}(v) = \alpha_v$, $\mathbf{R}_B \in \{0, 1\}^{M \times m}$ is a parent-child relation matrix of the decoded tree $B$, where $\mathbf{R}_B(v', j) = 1$ if node $v'$ is the ancestor of the leaf node $j$, and $M = 2m + 1$ is the total number of the nodes in $B$.

Note that the hyperbolic embedding in Lin et al. (2023) can be recast as a function mapping from the feature diffusion operator $\mathbf{P}$ to the product manifold of the hyperbolic spaces $\mathcal{M}$, denoted by $f_1 : \mathbf{P} \to \mathcal{M}$. The weight $\alpha_v$ in the our TWD can then be expressed by

$$
g_1 : \mathcal{M} \times \mathcal{M} \to \mathbb{R},
$$

and the weights vector $\boldsymbol{\alpha} \in \mathbb{R}^M$ in Eq. (12) can be written as $g_2 : f_1(\mathbf{P}) \times f_1(\mathbf{P}) \to \mathbb{R}^M$ by applying the function $g_1$ $M$ times. Let $h_1 : \mathcal{M} \to B$ be the hyperbolic diffusion binary tree decoding function in Alg. 1. The parent-child relation matrix $\mathbf{R}_B$ can be obtained by applying the function

$$
h_2 : B \to \{0, 1\}^{M \times m}
$$
$$
\Rightarrow \quad h_2 : h_1(\mathcal{M}) \to \{0, 1\}^{M \times m}
$$
$$
\Rightarrow \quad h_2 : (h_1 \circ f_1)(\mathbf{P}) \to \{0, 1\}^{M \times m},
$$

where $\circ$ denotes function composition. Therefore, the gradient of the our TWD w.r.t. $j$ can be written as

$$
\nabla \|\mathrm{diag}(f_1(\mathbf{P}) \times f_1(\mathbf{P}))((h_1 \circ f_1)(\mathbf{P})) (\mathbf{x}_i - \mathbf{x}_{i'})\|_1.
$$

This implies that the gradient of our TWD depends on the feature diffusion operator $\mathbf{P}$. Notably, since the gradient of the feature diffusion operator hinges on the gradient of the Gaussian kernel $\mathbf{Q}_c$ (as detailed in Sec. 3), the gradient of the proposed TWD can be expressed only in terms of the gradient of $\mathbf{Q}_c$. Such formulation could be highly beneficial in future applications that require rapid computation of the gradients of the proposed TWD.

### F.4 THE DIFFERENCES BETWEEN OUR TWD AND THE EMBEDDING DISTANCE (LIN ET AL., 2023)

Our TWD is a TWD between *samples* that incorporates the hidden hierarchical structure of the *features*, while the embedding distance (Lin et al., 2023) is a hierarchical distance between the samples. Namely, in our TWD, the features are assumed to have hidden hierarchies, whereas in Lin et al. (2023), the samples are assumed to have hidden hierarchies. The work in Lin et al. (2023) does not involve TWD or OT.

More concretely, in comparison to the work in Lin et al. (2023), our work presents a new algorithm for tree decoding (Alg. 1) and a corresponding new TWD (Alg. 2). Our work also includes theoretical analysis of these two algorithms:

1. Our TWD recovers the TWD associated with the true hidden feature tree, as stated in Thm. 2.

2. Our decoded tree metric represents a form of unsupervised ground metric learning derived from the latent tree underlying the features, as presented in Thm. 1.

3. Our TWD captures the tree-sliced information encoded in the multi-scale trees efficiently, as stated in Prop. 5.

4. Our TWD can be computed in linear time, enhancing computational efficiency in OT framework, as detailed in Sec. 4 and App. E.

5. Our TWD is differentiable (details in App. F.3), which is an important property that is lacking in existing TWD baselines.

In addition, we present favorable empirical results in Sec. 5 compared to several TWDs and Wasserstein distances.

Table 7: Document and single-cell classification accuracy of our TWD, embedding distance (ED), and OT distance using embedding distance (OT-ED).

|  | Real-World Word-Document | | | | Real-World scRNA-seq | |
|---|---|---|---|---|---|---|
|  | BBCSPORT | TWITTER | CLASSIC | AMAZON | Zeisel | CBMC |
| ED | $87.2_{\pm1.2}$ | $69.7_{\pm0.9}$ | $93.2_{\pm1.1}$ | $86.2_{\pm0.9}$ | $84.5_{\pm1.7}$ | $81.4_{\pm2.8}$ |
| OT-ED | $95.9_{\pm0.3}$ | $73.5_{\pm0.3}$ | $96.7_{\pm0.4}$ | $93.2_{\pm0.2}$ | $89.0_{\pm0.3}$ | $84.1_{\pm0.5}$ |
| Ours | $96.1_{\pm0.4}$ | $73.4_{\pm0.2}$ | $96.9_{\pm0.2}$ | $93.1_{\pm0.4}$ | $89.1_{\pm0.4}$ | $84.3_{\pm0.3}$ |

Seemingly, from a metric learning viewpoint, both our TWD and the embedding distance (Lin et al., 2023) generate distances between the samples. Since the two are fundamentally different, the comparison can be carried out in two ways: (i) directly compute the embedding distance between the samples (denoted as ED), and (ii) use the embedding distance between the features as the ground distance for the computation of the Wasserstein distance between the samples (denoted as OT-ED). We present the classification accuracy of the word-document and scRNA-seq datasets of these two baselines in Tab. 7. We see a clear empirical advantage of our TWD in classifying documents and scRNA-seq data compared to embedding distance (ED), as our TWD learns the latent hierarchy of the features explicitly by constructing a tree, and then, incorporating it in a meaningful distance between the samples. We note that while OT-ED performs similarly to our TWD in terms of accuracy (empirically corroborating Thm. 2), the computational expense is $O(mn^3 + m^3 \log m)$. In contrast, our proposed TWD offers a substantially reduced runtime of $O(m^{1.2})$, thereby demonstrating a significant advantage when handling large datasets. This computational advantage is empirically validated and illustrated in Fig. 4.

### F.5 INTUITION BEHIND SINGLE-LINKAGE TREE CONSTRUCTION IN HYPERBOLIC EMBEDDINGS

The single-linkage tree aligns well with hyperbolic embeddings because it inherently emphasizes hierarchical relationships by connecting points based on the measure of similarity determined by their HD-LCAs, where there is the "treelike" nature of a hyperbolic space (Bowditch, 2007). Specifically, Prop. 4 shows that the LCA depth closely approximates the Gromov product, which corresponds to the tree depth in 0-hyperbolic metrics.

In the context of tree-Wasserstein distance (TWD), many tree construction methods aim to approximate a given ground metric $\widetilde{d}$ (often Euclidean) using tree metrics, enabling efficient computation of optimal transport distances. The quality of TWD methods is typically assessed by the mean relative error between the proposed TWD and the Wasserstein distance with the ground metric $\widetilde{d}$. For instance, an ultrametric incurs a $\log m$ distortion with respect to Euclidean distance (Fakcharoenphol et al., 2003), as utilized in UltraTree (Chen et al., 2024). Hence, the ultrametric Wasserstein distance (Chen et al., 2024) incurs a $\log m$ distortion compared to the original Euclidean Wasserstein distance.

Our approach differs fundamentally from existing TWD methods. Instead of approximating the Wasserstein distance with a given ground metric $\widetilde{d}$ (e.g., Euclidean), we aim to approximate the Wasserstein distance with a latent underlying hierarchical distance via TWD, where we use the tree to represent the latent feature hierarchy.

