# OpenReview forum: "Tree-Wasserstein Distance for High Dimensional Data with a Latent Feature Hierarchy"
_ICLR.cc/2025/Conference — ICLR 2025 Poster_

### Official Review · Reviewer_sSsQ · 2024-10-19

**Soundness:** 3
**Presentation:** 3
**Contribution:** 3
**Rating:** 8
**Confidence:** 3

**Summary:**

This paper presents an approach for learning distances between samples when (a) features are known to exhibit hierarchical structure, and (b) data points are represented as distributions over features. The paper leverages previous work on multi-scale diffusion embeddings to obtain hyperbolic representations of features (not samples) and uses single-linkage clustering to build a tree of said features. The tree-Wasserstein distance is used to compute distances between samples, which leverages (a) the fact that data is represented as a distribution over features and (b) the learned feature hierarchy.

**Strengths:**

Originality: The idea of learning an explicit hierarchy of features (instead of samples) is a clever one, and it lends itself to applying the TWD when data points are represented as distributions over features.

Quality: The method appears to beat existing approaches as measured by tree reconstruction quality on synthetic data and nearest neighbor classification accuracy on real data.

Clarity: The paper is well-written and clear and uses consistent notation.

Significance: Metric learning for data where features exhibit hierarchical structure is important, as this is a common phenomenon. So any effort to (a) recover the feature hierarchy, and (b) extract a metric that reflects that hierarchy is significant.

**Weaknesses:**

1. The method assumes features are necessarily leaves in the ground-truth feature hierarchy. I can imagine simple situations (in language data) where this may not hold - for example, if 'fruit' was included as a word in the synthetic example provided in the paper, this assumption would not hold, it seems.

2. The construction of the feature diffusion operator could be explained in more detail. Primarily, the computation of the distance between features d(i,j) (which I assume to be the Euclidean distance between X[:, i] and X[:, j]) was not explicitly stated from what I can see. Can the authors add an explicit description of how these distances are computed?

3. On line 150 the LCA relaxation was referenced but never directly defined. Please add a restatement of the definition from Wang et. al in the body of the paper or add a description to the appendix.

4. The method doesn't address cases where the underlying feature tree cannot be isometrically embedded in Euclidean space. Can discussion be added to address this? For example, how would this affect the performance of the proposed approach?

5. No intuition is given for why a single-linkage tree construction approach performs better than other tree constructions given the hyperbolic embeddings. Could some intuition or theoretical justification be added?

**Questions:**

1. Can you clarify the construction of the diffusion operator as referred to above?
2. Does the method require assuming the feature tree can be isometrically embedded in Euclidean space?

---

> ### Author Response · Authors · 2024-11-21
>
> This is comment 1 out of 2.
>
> We thank the reviewer for the insightful comments and questions. In addition, we thank the reviewer for appreciating the idea of learning an explicit hierarchy of features and its relevance to distance metric learning and TWD for data represented as distributions over features.
>
> > **Assumption on Features Being Leaves in the Ground Truth Feature Hierarchy**
>
> We would like to clarify that we do not assume features are leaves in $T = ([m], E, \mathbf{A}, \mathbf{X}^\top)$. In this context, the vertex set $V = [m] = \\{1, \ldots, m\\}$ represents the nodes on the tree $T$.
>
> In our decoded tree $B$, however, we assign all $m$ features to be leaf nodes in $B$, where the tree $B$ consists of $2m-1$ nodes.
>
> To further emphasize that we do not assume features are leaves in $T$, we have extended the discussion in Appendix F.2, where we state that neither the underlying tree $T$ nor the hyperbolic embedding requires all features to be assigned to leaves. The assignment of features to leaf nodes is performed only during the tree decoding process.
>
> > **Clarification of Feature Diffusion Operator Construction**
>
> To clarify the construction of the diffusion operator, we have incorporated the reviewer's suggestion and specified in the revised version in Section 4.2 that we computed the initial distance using the cosine similarity in the ambient space, defined as $d(j,j') = 1- \frac{\mathbf{X}\_{:, j} \cdot \mathbf{X}\_{:, j'}}{\left\lVert\mathbf{X}\_{:, j}\right\rVert_2 \left\lVert\mathbf{X}\_{:, j'}\right\rVert_2 }$, where $\cdot$ is the dot product. We also referred to Appendix D for further discussion on the initial distance metric, the scale parameter $\epsilon$, and the kernel type.
>
> In the appendix, we highlighted that these elements, which do not have a one-size-fits-all solution, continue to be of wide interest in ongoing research [1]. Our primary objective is to highlight the unique aspects of our method rather than kernel selection, which, while crucial, is a common consideration across all kernel and manifold learning methods.
>
> [1] Lindenbaum et al., 2020. Gaussian bandwidth selection
> for manifold learning and classification.
>
> > **Definition of LCA Relation**
>
> In the revised version, we have incorporated the reviewer’s suggestion by including the definition of the LCA relation from [2] in Appendix A.4.
>
> [2] Wang \& Wang (2018).  An improved cost function for hierarchical cluster trees.
>
> > **Discussion on Non-Isometric Euclidean Embedding of Feature Trees**
>
> We would like to clarify that we do not assume the feature tree can be isometrically embedded in Euclidean space. While very small or specific types of trees (e.g., star-shaped trees) may allow for isometric embeddings in Euclidean space, general trees or hierarchical structures typically cannot. Embedding such structures into Euclidean space often introduces distortion [3]. For larger or more complex trees, hyperbolic space often serves as a more natural embedding space, preserving their hierarchical structure and distances [3-6].
>
> To clarify that we do not require feature trees to be isometrically embedded in Euclidean space, we have included this remark in Appendix A.4.
>
> [3] Nickel \& Kiela (2017). Poincaré embeddings for learning hierarchical representations
>
> [4] Nickel \& Kiela (2018). Learning continuous hierarchies in the Lorentz model of hyperbolic geometry
>
> [5] Chami et al. (2019). Hyperbolic graph convolutional neural
> networks.
>
> [6] Monath el al. (2019). Gradient-based hierarchical clustering using continuous representations of trees in hyperbolic space.

---

> ### Author Response · Authors · 2024-11-21
>
> This is comment 2 out of 2.
>
> > **Intuition Behind Single-Linkage Tree Construction in Hyperbolic Embeddings**
>
> The single-linkage tree aligns well with hyperbolic embeddings because it inherently emphasizes hierarchical relationships by connecting points based on the measure of similarity determined by their HD-LCAs, where there is the “treelike” nature of a hyperbolic space [7]. Specifically, Proposition 4 shows that the LCA depth closely approximates the Gromov product, which corresponds to the tree depth in 0-hyperbolic metrics.
>
> In the context of tree-Wasserstein distance (TWD), many tree construction methods aim to approximate a given ground metric $\widetilde{d}$ (often Euclidean) using tree metrics, enabling efficient computation of optimal transport distances.
> The quality of TWD methods is typically assessed by the mean relative error between the proposed TWD and the Wasserstein distance with the ground metric $\widetilde{d}$. For instance, an ultrametric incurs a $\log m$ distortion with respect to Euclidean distance [8], as utilized in UltraTree [9]. Hence, the ultrametric Wasserstein distance [9] incurs a $\log m$ distortion compared to the original Euclidean Wasserstein distance.
>
> Our approach differs fundamentally from existing TWD methods. Instead of approximating the Wasserstein distance with a given ground metric $\widetilde{d}$ (e.g., Euclidean), we aim to approximate the Wasserstein distance with a latent underlying hierarchical distance via TWD, where we use the tree to represent the latent feature hierarchy.
>
> To clarify why our single-linkage tree construction approach performs better than other tree constructions given the hyperbolic embeddings, we have included the above discussion in Appendix F.5.
>
> Thank you again for the valuable comment.
>
> [7] Bowditch (2007). A course on geometric group theory.
>
> [8] Fakcharoenphol et al. (2003). A tight bound on approximating arbitrary metrics by tree metrics.
>
> [9] Chen et al. (2024). Learning ultrametric trees for optimal transport regression.

---

> > ### Comment · Reviewer_sSsQ · 2024-11-22
> > **Response to rebuttal**
> >
> > Thank you for updating the manuscript and responding to comments! I still have a few things I would like to clarify.
> >
> > *To further emphasize that we do not assume features are leaves in T, we have extended the discussion in Appendix F.2, where we state that neither the underlying tree nor the hyperbolic embedding requires all features to be assigned to leaves. The assignment of features to leaf nodes is performed only during the tree decoding process.*
> > I think I see now. One can construct a decoded tree such that the hierarchy structure is preserved even if all features are assigned leaves. Is this inference correct? This is my interpretation derived from Figure 10.
> >
> > *We would like to clarify that we do not assume the feature tree can be isometrically embedded in Euclidean space. While very small or specific types of trees (e.g., star-shaped trees) may allow for isometric embeddings in Euclidean space, general trees or hierarchical structures typically cannot. Embedding such structures into Euclidean space often introduces distortion [3]. For larger or more complex trees, hyperbolic space often serves as a more natural embedding space, preserving their hierarchical structure and distances [3-6].*
> > Perhaps my question wasn't clear here. My concern is when the underlying ground truth hierarchy of features is structured such that it cannot be embedded in low-dimensional (or any dimensional) Euclidean space. This would mean the affinity matrix inferred from the cosine similarity of features (across all samples) would not faithfully represent the ground-truth feature hierarchy.

---

> ### Author Response · Authors · 2024-11-23
>
> We sincerely thank the reviewer for taking the time to carefully evaluate our responses and for their insightful questions.
>
> * Regarding the assignment of features to leaves: Yes, your interpretation is correct.
> * Regarding embedding trees in Euclidean space and the affinity matrix:
>
>     Thank you for clarifying your concern.
>
>     We agree with the reviewer that *global* distances, e.g., Euclidean or cosine distances, do not capture the hierarchical information or the underlying manifold structure of the features. However, we would like to point out that the affinity matrix is inherently *local* due to the use of the Gaussian kernel. It is not because local structures are computed with Euclidean components that the global geometry has to be Euclidean or that we are embedding in a Euclidean space. For instance, curved Riemannian manifolds are locally Euclidean, and the geodesic distance between nearby points can be approximated by the Euclidean distance, yet the geodesic distance between distant points is far from Euclidean.
>
>     In our method, the Gaussian kernel enables the affinity matrix to capture local relationships. This approach, related to heat diffusion on a curved non-Euclidean manifold, does not mean that we are embedding the features in a Euclidean space, as global distances are not Euclidean. This kernel enables the affinity matrix to capture local relationships, where affinities between features that are close in the ambient space embody information on the hidden hierarchical structure. Specifically, the discrete diffusion operator, constructed from these local Gaussian affinities in the ambient space, asymptotically converges pointwise to the heat kernel of the underlying hierarchical metric space in the limit of $m\rightarrow \infty$ and $\epsilon \rightarrow 0$. This is a standard practice in the studies of data manifolds and manifold learning to approximate and represent data on non-Euclidean manifolds [1, 2].
>
>     [1] Coifman \& Lafon (2006). Diffusion maps.
>
>     [2] Belkin et al. (2003). Laplacian eigenmaps for dimensionality reduction and data representation

---

> > ### Comment · Reviewer_sSsQ · 2024-11-24
> > **Response to comment**
> >
> > *We agree with the reviewer that global distances, e.g., Euclidean or cosine distances, do not capture the hierarchical information or the underlying manifold structure of the features. However, we would like to point out that the affinity matrix is inherently local due to the use of the Gaussian kernel.*
> > Thank you for the detailed response! My concern was not in fact about hierarchy preservation with respect to global distances. My concern was whether or not the metric structure of the feature hierarchy could be embedded isometrically into euclidean space such that *local* distances are preserved (and thus the affinity matrix could even capture information about the true hierarchy). For example, one cannot embed most trees into euclidean space such that all points connected by an edge (not all pairwise distances) exhibit the same distance as the original tree metric. Perhaps I am misunderstanding part of the construction of the method, but I don't believe I am.
> >
> > Regardless, the more I think about it, the more I believe this concern is outside of the scope of this paper. It seems reasonable to assume that (given other contributions within this work) local distances between features of the original data contain metric structure aligned with the original hierarchy (though I think there are many hierarchies in real datasets where this might not be true, as an isometric embedding into low dimensional Euclidean space is not possible).
> >
> > Given the authors detailed responses to my review and the reviews of others, I'm happy to bump my score. I think this is a well executed and interesting paper built on a clever insight.

---

### Official Review · Reviewer_7P2E · 2024-10-21

**Soundness:** 2
**Presentation:** 3
**Contribution:** 3
**Rating:** 6
**Confidence:** 3

**Summary:**

In their paper "Tree-Wasserstein distance for high dimensional data with a latent feature hierarchy", the authors consider the problem of constructing a NxN pairwise distance matrix between samples, given a data matrix X of NxM size and under assumption that the M features show some unknown hierarchical organization. The authors suggest a three-stage approach: (1) embed the M features into a hyperbolic space using diffusion eigenvectors; (2) construct a binary tree of the M features from the embedding; (3) use tree-Wasserstein distance between samples given the feature tree. In experiments, the authors argue that resulting distance yields superior kNN classification accuracy in bag-of-words and scRNA-seq datasets.

**Strengths:**

The paper considers an interesting problem setting, and suggests what seems to be a sensible algorithm. Technically the paper is well done, with high-quality figures and reasonably clear mathematical exposition.

**Weaknesses:**

The experimental evaluation is missing naive baselines, and in general lacks details.

Overall, I am giving a borderline score which I would be willing to revise upwards (or downwards) based on authors' clarifications regarding the experiments.

**Questions:**

MAJOR COMMENTS

* All comparisons in Table 1 are with other TWD distances. However, I would like to see a comparison to kNN classification accuracy directly in the feature space (R^M), as a baseline. The main point of the article is that inferring latent feature hierarchy is beneficial, so I'd like to see a comparison to a non-TWD approach. Please use the same distance function as elsewhere (I think you used cosine distance?).

* There are A LOT of TWD methods used for comparison in Table 1. Here I got a bit confused (because I am not familiar with this literature). From the introduction to this paper, my impression was that there are no existing TWD methods that estimate the latent feature tree. So what do all these TWD methods in Table 1 do instead? The description in section 5.1 suggests that you estimate the tree in many different ways and then use TWD with that. Why do all these methods of tree estimation work so much worse than yours? Maybe you need a part in the Introduction / Related work that would introduce these methods and explain what you are doing differently.

* The performance of kNN classification on the X matrix directly for RNAseq or text data will likely strongly depend on the normalization/transformation of the data. In RNAseq literature, row-normalized counts are usually log1p-transformed. In the NLP literature, words counts are usually transformed into TF-IDF with log-scaling. Operating on raw row-normalized counts without any transformation should perform much worse. It wasn't clear to me what data representation you use as input to your TWD benchmarking.


MINOR COMMENTS

* line 39: "due to its exponential growth" -- unclear formulation, whose growth?

* line 113: What is V? This notation wasn't introduced.

* line 126: Usually normalized Laplacian is defined with D^{-1/2} on both sides instead of D^{-1}. So I would expect to see Q hat defined as D^{-1/2} Q D^{-1/2}. Is this a typo? If not, why did you use D^{-1}? You say "following Coifman & Lafon 2006", but I am pretty sure they used D^{-1/2}. Furthermore, I would expect P to be defined directly as D^{-1} Q, instead of \hat D^{-1} \hat Q. Can you clarify?

---

> ### Author Response · Authors · 2024-11-21
>
> This is comment 1 out of 4.
>
> We thank the reviewer for the insightful comments and questions.
> In addition, we thank the reviewer for their appreciation of the technical quality, clarity of the mathematical exposition, and high-quality figures in our paper.
>
> >**Clarification of Data Representation for TWD Benchmarking**
>
> Regarding the data representation, we follow common methodologies in the Wasserstein-related fields. For word-document datasets, we use the benchmarks in [1],  which uses normalized bag-of-words and the Word2Vec embedding [2] trained on the Google News dataset as word embedding vectors. These benchmark datasets are studied in the context of Word Mover’s Distance [1, 3] and Tree-Wasserstein Distance [4–7]. For the scRNA-seq application, we use the datasets in [8], where the datasets are provided at the [link](https://github.com/solevillar/scGeneFit-python/tree/62f88ef0765b3883f592031ca593ec79679a52b4/scGeneFit/data_files), and there the CBMC are sparse and normalized by $\log_2(1+X)$. Here, we use the Gene2Vec embeddings [9] as the gene embedding vector following the study in Gene Mover's Distance [10].
>
> To clarify the data representations used in our experiments, we have updated the beginning of Section 5 and included the detailed information outlined above in Appendix D.1.
>
> >**Rationale Behind the Empirical Comparison to TWD Methods in Table 1**
>
> To address "Why do all these methods of tree estimation work so much worse than yours?": for data with a hidden hierarchy, explicitly inferring the geometry of the data, i.e., the hierarchy underlying the features, is crucial and contributes to the effectiveness of the distance learning between samples in downstream tasks on datasets with a hierarchical feature structure, e.g., word-document and scRNA-seq data. Our method constructs the tree to represent the underlying hierarchical structure and incorporates it in the TWD framework. In contrast, existing TWD methods also use a tree, but they do not use it to estimate the latent feature hierarchy. Instead, they only use the tree to approximate a given ground metric $\widetilde{d}$ (often Euclidean) and approximate the Wasserstein distance on the ground metric.
>
> The results in Table 1 demonstrate the importance of accounting for the feature hierarchy as our method yields consistently superior results to the many existing TWD-based methods, none of which accounts for that hierarchy. We have expanded the caption in Table 1 to highlight that the empirical comparisons to TWD methods demonstrate the advantages of our proposed approach.
>
> It is important to note that using the ground metric $\widetilde{d}$ (e.g., Euclidean) in WMD, GMD, or existing TWD methods, results in the (approximated) Wasserstein distance being based on the predefined metric $\widetilde{d}$. In contrast, our method employs an initial distance $d$ to calculate local affinities, computed using the cosine similarity in the experiments. This initial distance is used to compute an affinity matrix within the diffusion operator, which is central to kernel and manifold learning methods. This process leads to a Wasserstein distance based on a latent hierarchical metric induced by $d$. As noted in Appendix D.1, the kernel type, scale parameter, and initial distance metric selection in our approach significantly influence its effectiveness. These choices are task-specific and are critical in achieving optimal performance [12]. We have emphasized this point more clearly in Appendix F.5 in our revised version.

---

> ### Author Response · Authors · 2024-11-21
>
> This is comment 2 out of 4.
>
> >**Clarification of the Novelty of Our Approach Compared to Existing TWD Methods**
>
> Thank you for your insightful comment.
>
> In Appendix B, we discussed further details of existing Tree-Wasserstein Distance (TWD) methods and related work and their objectives.
>
> You are correct that, as outlined in the introduction, existing Tree-Wasserstein Distance (TWD) methods do not explicitly estimate the latent feature tree. Instead, these methods aim to approximate a given ground metric $\widetilde{d}$ (often Euclidean) using tree metrics. This allows for efficient computation of optimal transport distances (Wasserstein distances), as the computational complexity of TWD is linear, whereas that of standard optimal transport distances is cubic. For example, the Quadtree [18] is a widely used tree approximation method, that is used to define TWD in, e.g. [18, 5, 6, 7]. To construct the tree metric, the Quadtree method starts by defining a randomly shifted hypercube that contains all feature embedding vectors, and then recursively divides the hypercube into smaller hypercubes with half the side length, until each hypercube contains only a single feature embedding vector. Each hypercube corresponds to a node in the tree. The quality of TWD methods is typically assessed by the mean relative error between the proposed TWD and the Wasserstein distance with the ground metric $\widetilde{d}$. For instance, an ultrametric incurs a $\log m$ distortion with respect to Euclidean distance [11], as utilized in UltraTree [7]. Hence, the ultrametric Wasserstein distance [7] incurs a $\log m$ distortion compared to the original Euclidean Wasserstein distance.
>
> Our approach differs fundamentally from existing TWD methods. Instead of approximating the Wasserstein distance with a given ground metric $\widetilde{d}$ (e.g., Euclidean), we aim to approximate the Wasserstein distance with a latent underlying *hierarchical distance* via TWD, where we use the tree to represent the latent feature hierarchy. The key difference in our method lies in how we construct the latent feature tree. Our approach explicitly incorporates the hierarchical structure into the tree estimation process, tailored for the data's underlying geometry.
>
>
> Following the reviewer's comment, we have updated the text in Section 2 to emphasize that "We assume features lie in a latent hierarchical space, infer a tree that represents this space, and compute a TWD based on the inferred tree to define a meaningful distance between the samples, incorporating the *latent feature hierarchy*". In addition, we have added the reference in the Related Work section (Section 2) and highlighted it in Section 5.1, pointing to Appendix B, where we introduce the existing methods used for comparison and highlight the novel aspects of our approach.

---

> ### Author Response · Authors · 2024-11-21
>
> This is comment 3 out of 4.
>
> >**Comparison to Non-TWD Baselines**
>
> Thank you for your comment.
>
> * **New Naive Baselines in Feature Spaces.** Following the reviewer's request, we have conducted additional tests evaluating classification performance directly in the feature space. In the revised manuscript, the new Table 6 in Appendix E presents the document and cell classification accuracy in the feature space using the Euclidean distance and cosine distance. We see that the classification performance relying on generic metrics (Euclidean or cosine)  is less effective than our TWD method. This suggests that inferring explicitly the feature hierarchies is important and leads to the effectiveness of distance metric learning for data with a hierarchical structure.
>
> * **Additional Non-TWD baselines.** We would like to clarify that in addition to comparisons with TWD methods, we also evaluated our method with non-TWD methods that are specifically designed for each type of data, which are considered strong baselines in their respective domains.
>
>     We included Word Mover's Distance (WMD) [1] for word-document datasets and Gene Mover's Distance (GMD) [10] for scRNA-seq datasets. WMD computes the Wasserstein distance using the Euclidean distance $\widetilde{d}$ between precomputed Word2Vec embeddings as the ground metric, while GMD is defined similarly, using the Euclidean distance $\widetilde{d}$ between precomputed Gene2Vec embeddings as the ground metric.
>     Each of these methods was designed specifically for a certain type of data (word-document, gene expression, etc.), while our approach is general and infers the latent hierarchy. We present favorable empirical results, demonstrating that our method, while generic and not tailored to specific data types, is highly effective for datasets with hidden hierarchies underlying the features since we infer and utilize the hierarchical structure. We have emphasized this point more clearly in Table 1, Section 5.2, and Appendix D.1 in our revised version.
>
>
>      Motivated by the reviewer's comment, we have expanded our experiments to include additional evaluations of WMD on word-document datasets and GMD on scRNA-seq datasets, both using cosine distance as the ground metric (the new Table 5 in Appendix E in the revised manuscript).
>     The results show that WMD performs worse with cosine distance as the ground metric compared to Euclidean ground metric, while GMD performs better with ground metric using cosine distance than Euclidean distance.  However, regardless of the ground metric (cosine or Euclidean), our method outperforms WMD and GMD.
>      Specifically, when using cosine distance as the ground metric in WMD and GMD, our method achieves higher classification accuracy, with improvements of 4.9\% for BBCSPORT, 5\% for TWITTER, 3.4\% for CLASSIC, 3.7\% for AMAZON, 2.2\% for Zeisel, and 1.6\% for CBMC. These findings emphasize the value of inferring latent feature hierarchies in our method.
>
>
>
>     In response to the reviewer's comment on the non-TWD comparison, we have included the above discussion and the additional experiments in the revised version (Table 1, Section 5.2, Appendix D.1, Appendix E.3.6, and Appendix F.5). Thank you again for the valuable comment.

---

> ### Author Response · Authors · 2024-11-21
>
> This is comment 4 out of 4.
>
> > **Clarification of "Exponential Growth"**
>
> We have revised the text for clarity as follows: "Hyperbolic geometry has gained prominence in hierarchical representation learning because the lengths of geodesic paths in hyperbolic spaces grow exponentially with the radius, a property that naturally mirrors the exponential growth of the number of nodes in hierarchical structures as the depth increases".
>
> > **Typo $V$**
>
> Thank you for pointing this out. We have updated $V$ to $[m]$ in the revised version.
>
> > **Clarification of the Construction of the Diffusion Operator**
>
> We would like to clarify that the density-normalized affinity matrix was considered to mitigate the effect of non-uniform data sampling, which was proposed and shown theoretically in  Coifman \& Lafon (2006) [13]. In Section 3.1 of  Coifman \& Lafon (2006) [13], the construction of a family of diffusion operators is introduced with a parameter $\alpha \in \mathbb{R}$ and the density $q$. In Section 3.4 of  Coifman \& Lafon (2006) [13], it is shown that when $\alpha = 1$ (in our case $\widehat{\mathbf{Q}} = \mathbf{D}^{-1}\mathbf{Q}\mathbf{D}^{-1}$), the diffusion operator approximates the heat kernel, and this approximation is independent of the density $q$. This technique has been further explored and utilized in [14-17].
>
> In addition, we consider the column-stochastic version of the diffusion operator due to the propagation of the distributions on the graph. That is, given a probability density $\mathbf{w}$, the operation $\mathbf{Pw}$ keeps the resulting distribution normalized.
>
> To clarify the affinity matrix normalization and the diffusion operator construction, we have extended the discussion on diffusion geometry in Appendix A.2.
>
> ---
>
> **Reference**
>
> [1] Kusner et al., (2015). From word embeddings to document distances.
>
> [2] Mikolov et al., (2013). Distributed representations of words and phrases and their compositionally.
>
> [3] Huang et al. (2016). Supervised word mover's distance.
>
> [4] Le et al. (2019). Tree-sliced variants of Wasserstein
> distances.
>
> [5] Takezawa et al. (2021). Supervised tree-Wasserstein distance.
>
> [6] Yamada et al. (2022). Approximating 1-Wasserstein distance with trees.
>
> [7] Chen et al. (2024). Learning ultrametric trees for optimal transport regression.
>
> [8] Dumitrascu et al. (2021). Optimal marker
> gene selection for cell type discrimination in single cell analyses.
>
> [9] Du et al. (2019). Gene2vec: distributed representation of genes based on co-expression.
>
> [10] Bellazzi et al., (2021).  The gene mover’s distance: Single-cell similarity via optimal transport.
>
> [11] Fakcharoenphol et al. (2003). A tight bound on approximating arbitrary metrics by tree metrics.
>
> [12] Lindenbaum et al., 2020. Gaussian bandwidth selection
> for manifold learning and classification.
>
> [13]  Coifman \& Lafon (2006). Diffusion maps.
>
> [14] Shnitzer et al. (2022). Log-Euclidean Signatures for Intrinsic Distances Between Unaligned Datasets.
>
> [15] Shnitzer et al. (2024). Spatiotemporal Analysis Using Riemannian Composition of Diffusion Operators.
>
> [16] Katz et al. (2020). Spectral Flow on the Manifold of SPD Matrices for Multimodal Data Processing.
>
> [17] Mishne et al. (2019). Diffusion nets.
>
> [18] Indyk \& Thaper, (2003). Fast image retrieval via embeddings.

---

> > ### Comment · Reviewer_7P2E · 2024-11-21
> >
> > I thank the authors for their very detailed responses. Many things that you wrote sound overall reasonable, but I am concerned about the "naive baselines".
> >
> > For example, on the Zeisel dataset you report kNN classification accuracy 89.1% with your method and 67.2% / 73.5% in the feature space with Euclidean/cosine metrics. These numbers  seem very low... The Zeisel dataset (Table 2) has only 7 classes corresponding to major cell types; they should be very well separated in the log-expression space. The original paper contains a t-SNE that suggests that these 7 classes are separated very well, and t-SNE operates on the kNN graph of the data. So I would expect kNN accuracy in the feature space to be high.
> >
> > Actually, after I wrote the above paragraph, I went ahead and computed the kNN accuracy myself. I downloaded the Zeisel files from the link provided in the paper. Here is my entire code:
> >
> > ```
> > from scipy import io
> >
> > f = io.loadmat("zeisel_data.mat")
> > X = f['zeisel_data'].T
> >
> > f = io.loadmat("zeisel_labels1.mat")
> > y = f['zeisel_labels1'].ravel()
> >
> > from sklearn.neighbors import KNeighborsClassifier
> > from sklearn.model_selection import train_test_split
> >
> > X_train, X_test, y_train, y_test = train_test_split(X, y, test_size=0.3, random_state=1)
> >
> > KNeighborsClassifier().fit(X_train, y_train).score(X_test, y_test)
> > ```
> >
> > This results in 95.9% classification accuracy. What am I doing differently?

---

> > > ### Author Response · Authors · 2024-11-22
> > >
> > > We sincerely thank the reviewer for taking the time to carefully evaluate our responses and for their thorough feedback regarding the cell classification performance in the feature space.
> > >
> > > In our experiments, to show the advantage of incorporating feature hierarchy in distance metric learning for data with a hierarchical structure, we actually consider the more intricate task using subclasses as labels for cell classification, rather than the major classes.
> > >
> > > In the original Zeisel dataset [1], 47 subclasses are obtained using the divisive biclustering method from 9 classes, and we use these subclasses as labels in our experiments, and not from [2], which is a typo. Classification of the subclasses is significantly more challenging than that of the major classes, which explains why our reported performance of the naive baselines is lower than what the reviewer points out.
> > > We agree with the reviewer that considering the major class classification is a simple task, as the naive generic metric already yields strong performance. For example, when using the alternative 7 major classes of [2], the generic metrics perform well (Euclidean 95.7\% $\pm$ 0.1 and Cosine 88.8\% $\pm$ 0.2), yet our method has marginal improvements (96.4\% $\pm$ 0.2).
> > >
> > >
> > >
> > > Following the reviewer's comment, we also tested the alternative 48 subclasses from [2], where our method attains the classification accuracy 82.6\% $\pm$ 0.2, outperforming the generic metrics  (Euclidean 62.4\% $\pm$ 0.1 and Cosine 52.6\% $\pm$ 0.2). These results are consistent with our findings on subclass classification using labels obtained from [1], demonstrating that incorporating feature hierarchy is more effective for finer-grained labels in subclass classification.
> > >
> > > We have fixed the typo in the revised manuscript, detailing that we consider the fine-grained labels and not the major classes. The fix corresponds to an update of Appendix D.1 and Table 2 to clarify the label information used for the scRNA-seq data in our experiments.
> > >
> > > In the final version, we will provide our code for reproducing our results.
> > >
> > > Thank you again for pointing this out.
> > >
> > >
> > > [1] Zeisel et al. (2015). Cell types in the mouse cortex and hippocampus revealed by single-cell RNA-seq
> > >
> > > [2] Dumitrascu et al. (2021). Optimal marker gene selection for cell type discrimination in single cell analyses.

---

> > > > ### Comment · Reviewer_7P2E · 2024-11-22
> > > >
> > > > Oh I see, this explains it. Using `zeisel_labels2.mat`, I can get to 69% if I use PCA preprocessing to 50 components first.
> > > >
> > > > ```
> > > > Z = PCA(n_components=50).fit_transform(X)
> > > > X_train, X_test, y_train, y_test = train_test_split(Z, y, test_size=0.3, random_state=1)
> > > > KNeighborsClassifier(n_neighbors=10).fit(X_train, y_train).score(X_test, y_test)
> > > > ```
> > > >
> > > > This is much lower than 82% that you obtained. Very impressive. I have to say that I still don't fully understand this result... All these subclasses are not ground truth, but in fact were derived by some sort of clustering in the original feature space. So it's surprising to me that the classification accuracy is higher with some other metric, like your TWD, given that TWD was *not* used to derive the class labels in the original publications.
> > > >
> > > > In any case, I am increasing my score to recommend acceptance.

---

### Official Review · Reviewer_EDMh · 2024-10-31

**Soundness:** 3
**Presentation:** 3
**Contribution:** 3
**Rating:** 8
**Confidence:** 3

**Summary:**

The authors propose an algorithm for constructing a binary tree from data points in the Poincaré half-space. They furthermore propose a method for learning a distance between histograms, assuming a latent hierarchical relation between the bins of the histograms. The method starts with embedding the columns of the data matrix (where each sample/row is a histogram) into Poincaré half-spaces using a combination of diffusion embeddings at multiple scales. In a second step, the tree construction algorithm is applied to these embeddings, resulting in a tree on the bins of the histograms. Finally, they plug this tree into the definition of the Tree Wasserstein distance, resulting in the desired distance between histograms.

For the tree construction algorithm the authors show that, asymptotically in the number of dimensions and scales, it recovers a tree defined by the hyperbolic lowest common ancestor metric on the Poincaré half-space. For the complete method, it is shown that (asymptotically in the number of dimensions and scales) the resulting distance is equivalent within a factor 1/sqrt(2) to TWD using a “ground latent tree” (which is unfortunately left undefined).

The authors validate the theoretical results using experiments with a synthetic dataset (where “ground latent tree” is a probabilistic graphical model), a text dataset as well as single-cell gene expression dataset.

**Strengths:**

The proposed method is well-motivated and combines an innovative application of previous results (on hyperbolic diffusion embeddings) with novel contributions (tree construction algorithm based on hyperbolic LCAs).

The paper clearly highlights the contributions. The related work and preliminaries sections are clearly written and contain necessary background material together with the appropriate references. The exposition of the method is at an appropriate level of detail and easy to follow once the preliminaries have been digested. The appendix contains further material related to the efficient implementation of the method.

The problem of computing distances between histograms based on a latent tree is well-motivated by practical applications to single-cell sequencing data.

**Weaknesses:**

The paper never gives a definition of the "ground latent tree" used in the statements of Theorems 1 and 2. Theorem 1 references Section 4.1, which however does not seem to contain any formal definitions. The proof of Theorem 1 (via Theorem C.1) references Lin et al. (2023) Theorem 1 for the convergence of the manifold metric to the "hidden tree metric". Without a definition of the "hidden tree" or "ground latent tree", the Theorems and the paper as a whole are incomplete.

At first reading, it was unclear to me what is meant by "features" and "feature hierarchy" and which are the measures to which the Wasserstein distance is eventually applied. The paper states for the first time that it "focuses" on data points that are essentially histograms on page 4, l.172 and only confirms that the method is limited to histograms on page 7, line 372. It would greatly improve the presentation of the paper if the setting was made clear from the start.

I set my rating to weak reject because the paper is missing the definition of the "ground latent tree" used in Theorems 1 and 2. If that definition is added (and consistent with the experiments), I am willing to raise my rating to accept.

**Questions:**

* l. 113 V is not defined, should this be [m]?

---

> ### Author Response · Authors · 2024-11-21
>
> We thank the reviewer for the insightful comments and questions, and their appreciation of the motivation behind our method, the tree decoding algorithm in hyperbolic spaces, and the clarity of our exposition.
>
> >**Definition of the Ground Truth Latent Tree**
>
> Thank you for this important comment.
>
> In response to the reviewer’s comment, we have extended the description in Section 4.1 to further clarify the definition of the ground truth latent tree. Specifically, we assume the features $\\{\mathbf{X}\_{:,1}, \ldots, \mathbf{X}\_{:,m}\\} \subseteq \mathcal{H} \subset \mathbb{R}^n$ lie on an underlying manifold $\mathcal{H}$, which is a complete and simply connected Riemannian manifold with negative curvature embedded in a high-dimensional ambient space $\mathbb{R}^n$ with geodesic distance $d_\mathcal{H}$. We view $(\mathcal{H}, d_\mathcal{H})$ as a hidden hierarchical metric space and consider a weighted tree $T = ([m], E, \mathbf{A}, \mathbf{X}^\top)$ as its discrete approximation in the following sense. The tree node $j \in [m]$ is associated with the $j$-th feature, $E$ is the edge set, and $\mathbf{A}\in \mathbb{R}^{m \times m}$ is the weight matrix of the tree edges, defined such that the tree distance $d_T(j, j')$ between two nodes $j$ and $j'$, i.e., the length of the shortest path on $T$, coincides with the geodesic distance $d_\mathcal{H}(j, j')$ between the features $j$ and $j'$. Therefore, we refer to the hidden tree distance $d_T$ as the ground truth distance. Note that the assumption of an underlying manifold embedded in an ambient space $\mathbb{R}^n$ is widely used in studies of data manifolds and manifold learning [1-5], where the manifold typically underlies *samples* and graphs are viewed as discretizations of the underlying Riemannian manifold. In contrast, we consider the manifold underlying *features*. Specifically, we focus on a specification of the manifold to a hierarchical space $\mathcal{H}$ of features, and accordingly, a specification of the graph to a tree $T$. Our aim is to construct a tree $B$ from the data $\mathbf{X}$, such that the TWD between samples using our learned tree $B$ is (bilipschitz) equivalent to the Wasserstein distance using the ground truth latent tree distance $d_T$. To allow this construction of $B$, we assume that local affinities between (node) features $\\{\mathbf{X}\_{:,j}\\}\_{j=1}^m$ that are close in $\mathbb{R}^n$ embody information on the hidden hierarchical structure. More specifically, we assume that the discrete diffusion operator $\mathbf{P}$ constructed from the local Gaussian affinities computed on the features in the ambient space, in the limit of $m\rightarrow \infty$ and $\epsilon \rightarrow 0$, converges pointwise to the heat kernel of the underlying hierarchical metric space $(\mathcal{H},d_{\mathcal{H}})$.
>
>
>
> [1] Tong et al. (2021). Diffusion Earth Mover’s Distance and Distribution Embeddings.
>
> [2] Shnitzer et al. (2022). Log-Euclidean Signatures for Intrinsic Distances Between Unaligned Datasets.
>
> [3] Huguet et al. (2022). Manifold Interpolating Optimal-Transport Flows for Trajectory Inference.
>
> [4] Kapusniak et al. (2024). Metric Flow Matching for Smooth Interpolations on the Data Manifold.
>
> [5] Kruiff et al.  (2024). Pullback Flow Matching on Data Manifolds.
>
> >**Improving Clarity on Features, Feature Hierarchy, and Method Setting**
>
> To improve the clarity of the problem setup, we have incorporated the reviewer's suggestions into the Introduction section. Specifically, to improve the clarity of "feature" and "feature hierarchy",  we have included word-document data in the following illustrative example of high-dimensional datasets with a latent feature hierarchy. In word-document data, samples correspond to documents, and features correspond to words. The hierarchy underlying features (i.e., words) is often assumed because words are naturally organized as nodes in a tree, each connected to others through various hierarchical linguistic relationships. In addition, to improve the clarity of "the measures to which the Wasserstein distance is applied", we have highlighted that we model the data samples as distributions supported on a latent hierarchical structure.
>
> >**Typo $V$**
>
> Thank you for pointing this out. We have updated $V$ to $[m]$ in the revised version.

---

> > ### Comment · Reviewer_EDMh · 2024-11-22
> >
> > Thank you, I increased my score to accept.

---

### Official Review · Reviewer_89iu · 2024-11-11

**Soundness:** 2
**Presentation:** 3
**Contribution:** 3
**Rating:** 6
**Confidence:** 4

**Summary:**

The paper presents a new Tree-Wasserstein Distance that is specifically designed for a hierarchical data and presents a novel flow matching method. This method can be applied to high dimensional data measured at non-equidistant time points without reducing dimensionality, so that the dynamics of the data are not oversimplified. The method is validated on several datasets and showed the improvement compared to existing methods.

**Strengths:**

I think this paper is a nice combination of three ideas, 1) from finite data construct a hyperbolic embedding, 2) from points in hyperbolic space construct a tree 3) compute tree wasserstein distance on that tree. The part 1) is borrowed from Lin et al. (2023) and part 3) is the usual tree wasserstein distance, but in 2) algorithms and theoretical works seems to be their own and nice combinations from hyperbolic geometry. I felt this originality and signifcance should be appreciated. Although, my research is not in tree wasserstein distance or hyperbolic geometry with machine learning, so I cannot correctly evaluate originality or significance. I think the paper is is clearly written. I think the paper is in general in good quality, though some theoretical results are a bit skeptical: see weaknesses below.

**Weaknesses:**

In Theorem 1, the latent tree distance $d_{T}$ and the decided tree metric $d_{B}$ are claimed to be bilipschitz with constants $1/\sqrt{2}$ and $1$. However, I am not sure if the bilipschitz constants are correct. The authors just said that "Specifically, by taking $\alpha\rightarrow 1/2$ and $\beta =2$ in
Section 3 in Leeb & Coifman (2016) and Appendix C in Lin et al. (2023), the bilipschitz constants $c$
and $C$ can be determined by $1/\sqrt{2}$ and $1$ in $d_{H} \simeq d_{T}$, respectively." and did not provide detailed proof for why the constant can be $1/\sqrt{2}$ and $1$. However, proofs of Section 3 in Leeb & Coifman (2016) and Appendix C in Lin et al. (2023) are "all up to constants" and did not trace exact constants.

For example, for the Lipschitz constant $1$, i.e., $d_{B} \geq d_{T}$, it is somewhat necessary to argue that $d_{\mathcal{M}} \leq d_{T}$, where $d_{\mathcal{M}}$ is the distance in Lin et al. (2023). In Lin et al. (2023), the corresponding proof is Proposition C.2, and here the constant $1$ contains several constants as $A$ and $\epsilon$ (different from $epsilon$ in the authors' paper). In particular, the $\epsilon$ is from Lemma 3 in Leeb & Coifman (2016), which is $C_{1}\Psi(1) - C_{2}\Phi(A-1)$ in the corresponding proof. It does not seem possible to simplify everything and conclude that we can take the constant $1$.

I think authors should provide proof for how the bilipschitz constants $1/\sqrt{2}$ and $1$, or they should change the statement of Theorem 1 should be weakened to "there exists some constants $C_1$ and $C_2$ so that $C_1 d_B \leq d_T \leq C_2 d_B$".

Another thing is about the hyperparameter tuning. The suggested algorithm is based on the diffusion operator P in Equation (2) to provide similarity between data points. To compute Equation (2), the user needs to determine the scale parameter $epsilon>0$. Hence, I think providing a guideline for choosing $epsilon$ should be helpful to readers, for e.g., Lin et al. (2023) suggested the median of distances between data points multiplied by a constant.

**Questions:**

I put the suggestions in the weakness.

---

> ### Author Response · Authors · 2024-11-21
>
> We thank the reviewer for the insightful and thorough review, and their appreciation of our hyperbolic tree decoding algorithm and its integration with TWD.
>
> >**Bilipschitz Constants in Theorem 1 and 2**
>
> We thank the reviewer for the detailed review and insightful comments.
>
> In our proof, to show the equivalence between the ground truth distance $d_T$ and the inferred distance $d_B$, we used an auxiliary distance on the manifold $d_\mathcal{M}$, and show the equivalence in two steps, leading to $d_B \simeq d_\mathcal{M} \simeq d_T$.
>
> In our initial submission, due to the haste in preparing the manuscript, we mistakenly took the constant from the intermediate equivalence $d_B$ and $d_\mathcal{M}$, while the constants between $d_\mathcal{M}$ and $d_T$ are more involved [1,2].
>
> In the revised version, we have incorporated the reviewer’s suggestion and corrected the statement in Theorem 1: for sufficiently large $K_c$ and $m$, there exist constants $C_1, C_2 > 0$ such that $C_1 d_T \leq d_B\leq C_2 d_T$, and we have revised Theorem 2 accordingly.
>
> Thank you for this important comment.
>
>
> [1] Leeb \& Coifman (2016). Hölder--Lipschitz norms and their duals on spaces with semigroups, with applications to earth mover's distance.
>
> [2] Lin et al. (2023). Hyperbolic diffusion embedding and distance for hierarchical representation learning.
>
> >**Hyperparameter Tuning**
>
> In the revised version, we have incorporated the reviewer’s comment in Section 2 regarding the scale parameter, noting that it is often chosen as the median pairwise distance among data points scaled by a constant factor. We also refer to Appendix D for additional details on hyperparameter tuning.

---

### Meta-Review · Area_Chair_Rh1b · 2024-12-19

**Metareview:**

The authors focus on estimating the hierarchical structure (i.e., tree structure) given observations from latent hierarchical features, and leverage the tree-Wasserstein for a fast computation for measures with hierarchical structure on supports. The authors propose to embed features into a multi-scale hyperbolic space and algorithmic approach to estimate the tree structure from embedded features. The authors show advantages of estimating tree structure from observed data (with a latent hierarchical structure) for tree Wasserstein on document and single-cell RNA sequencing datasets. All the Reviewers agree that it is a good submission for ICLR'2025. However, it is better to clarify whether the proposed approach can be applied for general applications which may or may not have latent hierarchical structure as in standard tree-sliced-Wasserstein setting (Le et al., 2019) (i.e., whether the proposed approach can be applied for applications without latent hierarchical structure? and/or how to know whether a given dataset in practice has prerequisite latent hierarchical structure or not?). We urge the Authors to incorporate the Reviewers' comments and discussions in the rebuttal into the updated version.

**Additional Comments On Reviewer Discussion:**

The authors clarify an important missing piece on the assumption of latent hierarchical structure of observed data, and the importance of estimating that hierarchical structure to construct tree for tree-Wasserstein in applications.

---

### Decision · Program_Chairs · 2025-01-22

Accept (Poster)